# High Probability Generalization Bounds with Fast Rates for Minimax Problems

**Shaojie Li[1,2], Yong Liu[1,2],***
[1]Gaoling School of Artificial Intelligence, Renmin University of China, Beijing, China
[2]Beijing Key Laboratory of Big Data Management and Analysis Methods, Beijing, China
`2020000277@ruc.edu.cn, liuyonggsai@ruc.edu.cn`

## Abstract

Minimax problems are receiving an increasing amount of attention in a wide range of applications in machine learning (ML), for instance, reinforcement learning, robust optimization, adversarial learning, and distributed computing, to mention but a few. Current studies focus on the fundamental understanding of general minimax problems with an emphasis on convergence behavior. As a comparison, there is far less work to study the generalization performance. Additionally, existing generalization bounds are almost all derived in expectation, and the high probability bounds are all presented in the slow order $\mathcal{O}(1/\sqrt{n})$, where $n$ is the sample size. In this paper, we provide improved generalization analyses and obtain sharper high probability generalization bounds for most existing generalization measures of minimax problems. We then use the improved learning bounds to establish high probability generalization bounds with fast rates for classical empirical saddle point (ESP) solution and several popular gradient-based optimization algorithms, including gradient descent ascent (GDA), stochastic gradient descent ascent (SGDA), proximal point method (PPM), extra-gradient (EG), and optimistic gradient descent ascent (OGDA). In summary, we provide a systematical analysis of sharper generalization bounds of minimax problems.

## 1 Introduction

Minimax learning problems have achieved great success over a broad range of learning tasks in machine learning, with examples including reinforcement learning (Du et al., 2017; Dai et al., 2018), robust optimization (Chen et al., 2017; Namkoong & Duchi, 2017), adversarial learning (Goodfellow et al., 2014), distributed computing (Razaviyayn et al., 2020; Shamma, 2008; Mateos et al., 2010), and AUC maximization (Lei & Ying, 2021b), to just name a few. This framework is formulated as a zero-sum game characterized as two groups of decision variables, one for minimization and one for maximization. The coupling of the two groups of variables makes analysis of minimax problems more complex than the standard statistical learning theory setting, with only one minimization operator (Liu et al., 2021b; Yin et al., 2020; Li & Liu, 2021; Li et al., 2018; Liu et al., 2020; Li & Liu, 2021). Researchers have designed various optimization algorithms, for instance, gradient descent ascent (GDA), stochastic gradient descent ascent (SGDA), proximal point method (PPM), extra-gradient (EG), and optimistic gradient descent ascent (OGDA), to solve the minimax optimization problem (Farnia & Ozdaglar, 2021). Current theoretical research in ML literature is mainly devoted to the convergence rate and optimality of these minimax optimization algorithms in different setting, such as convex-concave settings (Nemirovski et al., 2008), nonconvex-concave setting (Rafique et al., 2018), strongly convex-strongly-concave setting (Balamurugan & Bach, 2016), and nonconvex-nonconcave setting (Liu et al., 2021a; Yang et al., 2020). In contrast, there is far less work on the generalization performance analysis, which is an important measure to indicate the performance of the learned model based on training samples when generalized to the test data.

To the best of our knowledge, there is only three work on the generalization bounds of minimax optimization algorithms (Zhang et al., 2021a; Farnia & Ozdaglar, 2021; Lei et al., 2021). Among them,

---

(Zhang et al., 2021a) studies the generalization bounds for ESP solution to minimax problems, (Farnia & Ozdaglar, 2021) analyzes the generalization properties of several gradient-based optimization algorithms: GDA, SGDA, GDmax and PPM, and (Lei et al., 2021) provides a systematical generalization analysis of SGDA. However, in the above-mentioned papers, almost all generalization bounds are derived in expectation. Only two high probability bounds exist, proposed in (Lei et al., 2021). Unfortunately, they are of the slow order $\mathcal{O}\left(1/\sqrt{n}\right)$.

It is known that the high probability bound is beneficial to understand the robustness of optimization algorithms (Bousquet et al., 2020; Klochkov & Zhivotovskiy, 2021) and is much more challenging to be derived (Bousquet et al., 2020; Lei et al., 2021; Lv et al., 2021). In this paper, our goal is to provide the sharper high probability generalization bounds for minimax learning problems. We leverage the lens of algorithmic stability, which is also served as an important tool in (Zhang et al., 2021a; Farnia & Ozdaglar, 2021; Lei et al., 2021). Our contributions are summarized below.

1. In view of the coupling construction between the minimization variable and the maximization variable, minimax learning problems have many generalization measures (Lei et al., 2021; Farnia & Ozdaglar, 2021; Zhang et al., 2021a). In this paper, we provide improved stability analyses for almost all existing generalization measures, based on which we establish sharper high probability generalization bounds. These developed learning bounds can be employed to derive generalization bounds with fast rates for stable minimax learning algorithms.

2. The generalization performance of the ESP solution and gradient-based optimization algorithms stands a central place in the learning theory of minimax problems (Lei et al., 2021). In this paper, we develop high probability generalization bounds with fast rates for ESP solution and several popular gradient-based optimization algorithms: GDA, SGDA, PPM, EG, and OGDA. Overall, we provide a systematical analysis of sharper generalization bounds for minimax learning problems.

## 2 RELATED WORK

**Algorithmic stability.** Algorithmic stability is a fundamental concept in learning theory (Bousquet & Elisseeff, 2002), which has a deep connection with learnability (Rakhlin et al., 2005; Shalev-Shwartz & Ben-David, 2014; Shalev-Shwartz et al., 2010). A training algorithm is stable if small changes in the training set lead to small differences in the output predictions of the trained model. Different algorithmic stability measures have been developed, including uniform stability (Bousquet & Elisseeff, 2002; Feldman & Vondrak, 2018; 2019; Klochkov & Zhivotovskiy, 2021; Hardt et al., 2016; Lei et al., 2020), uniform argument stability (Liu et al., 2017; Bassily et al., 2020), hypothesis stability (Bousquet & Elisseeff, 2002; Charles & Papailiopoulos, 2018), hypothesis set stability (Foster et al., 2019), on average stability (Shalev-Shwartz et al., 2010; Lei & Ying, 2020; Kuzborskij & Lampert, 2018; Zhang et al., 2021b; Lei & Ying, 2021a), locally elastic stability (Deng et al., 2021), collective stability (London et al., 2016), and PAC-Bayesian stability (Li et al., 2020). These stability measures have been extensively studied in the generalization analysis of the standard statistical learning theory setting (Chen et al., 2018; Zhang et al., 2021b). Several stability measures have also been extended to minimax learning problems, for instance, weak stability, argument stability, and uniform stability (Farnia & Ozdaglar, 2021; Zhang et al., 2021a; Lei et al., 2021). In related work (Farnia & Ozdaglar, 2021; Zhang et al., 2021a; Lei et al., 2021), they mostly focus on the expectation form of these stability measures since they are to derive bounds in expectation. In this paper, we will focus on the last two measures, which are often used when establishing high probability generalization bounds (Feldman & Vondrak, 2018; 2019; Klochkov & Zhivotovskiy, 2021).

**Convergence analysis.** Convergence analysis has been widely studied in different settings, including convex-concave learning (Nemirovski, 2005; Nedic & Ozdaglar, 2009; Mokhtari et al., 2020; Cherukuri et al., 2017; Mokhtari et al., 2019; Balamurugan & Bach, 2016; Hsieh et al., 2019; Yan et al., 2020; Lin et al., 2020b; Wang & Li, 2020; Yoon & Ryu, 2021), nonconvex-concave learning (Rafique et al., 2018; Kong & Monteiro, 2019; Luo et al., 2020; Grnarova et al., 2017; Thekumparampil et al., 2019; Lu et al., 2020; Namkoong & Duchi, 2016; Sanjabi et al., 2018; Nouiehed et al., 2019; Lin et al., 2020a; Sinha et al., 2017; Chen et al., 2021), and nonconvex-nonconcave learning (Heusel et al., 2017; Balduzzi et al., 2018; Daskalakis & Panageas, 2018; Mertikopoulos et al., 2019; Loizou et al., 2020; Yang et al., 2020; Liu et al., 2021a; Lin et al., 2018; Diakonikolas et al., 2021; Wang et al., 2020; Loizou et al., 2021; Fiez & Ratliff, 2021). There are so many studies on convergence. Thus, considering the length limit, the references listed here are not complete. Please refer

to the related references concerning the above work. We investigate the generalization performance of minimax problems instead of the convergence behavior. Note that the convergence analysis also plays an essential role in this paper, formalized as strong PD empirical risk (please refer to Definition 1), which is defined on the function value difference and referred to as optimization error or primal-dual gap in some convergence literature (Lei et al., 2021; Nemirovski, 2005; Mokhtari et al., 2019; 2020).

## 3 PRELIMINARIES

Let $\mathcal{X}$ and $\mathcal{Y}$ be two parameter spaces in $\mathbb{R}^d$. Let $\mathbb{P}$ be a probability measure defined on a sample space $\mathcal{Z}$. We define $f : \mathcal{X} \times \mathcal{Y} \times \mathcal{Z} \mapsto \mathbb{R}$ and consider the following minimax optimization problem

$$\min_{\mathbf{x} \in \mathcal{X}} \max_{\mathbf{y} \in \mathcal{Y}} F(\mathbf{x}, \mathbf{y}) := \mathbb{E}_{z \sim \mathbb{P}}[f(\mathbf{x}, \mathbf{y}; z)]. \tag{1}$$

The above minimax objective represents an expectation of a cost function $f(\mathbf{x}, \mathbf{y}; z)$ for minimization variable $\mathbf{x}$, maximization variable $\mathbf{y}$ and data variable $z$. Unfortunately, we typically are not available to the underlying distribution $\mathbb{P}$. In practice, $F$ is approximated by the corresponding empirical risk. Let $S = \{z_1, ..., z_n\}$ be a dataset whose samples are independent drawn according to $\mathbb{P}$, the empirical risk is defined as

$$F_S(\mathbf{x}, \mathbf{y}) = \frac{1}{n} \sum_{i=1}^{n} f(\mathbf{x}, \mathbf{y}; z_i). \tag{2}$$

Let the output of a (randomized) algorithm $A$ on a dataset $S$ be $A(S) := (A_{\mathbf{x}}(S), A_{\mathbf{y}}(S)) \in \mathcal{X} \times \mathcal{Y}$. Since $A(S)$ is just an empirical approximated solution of the true minimax optimization problem, we are interested in studying how well $A(S)$ generalizes to the unseen data. As claimed in (Farnia & Ozdaglar, 2021; Lei et al., 2021), the coupling between the minimization variable and the maximization variable in (1) makes minimax problems have many different generalization performance measures. These measures are collected in (Lei et al., 2021). For better readability, we use their symbols. Let $\mathbb{E}$ be the expectation with respect to (w.r.t.) the randomness of algorithm $A$ and the dataset $S$. These generalization measures are listed below.

**Definition 1.** *(Lei et al., 2021) There are four groups of generalization measures.*

*1 (Primal Measures.) The primal population risk of a model $\mathbf{x}$ is defined as $R(\mathbf{x}) = \sup_{\mathbf{y} \in \mathcal{Y}} F(\mathbf{x}, \mathbf{y})$, and the corresponding primal empirical risk is defined as $R_S(\mathbf{x}) = \sup_{\mathbf{y} \in \mathcal{Y}} F_S(\mathbf{x}, \mathbf{y})$. Then, when using empirical risk $R_S(\mathbf{x})$ to bound $R(\mathbf{x})$, we call this error of the model $\mathbf{x}$ the primal generalization error. While using optimal $\inf_{\mathbf{x} \in \mathcal{X}} R(\mathbf{x})$ to bound $R(\mathbf{x})$, we call this error of the model $\mathbf{x}$ the excess primal population risk.*

*2 (Plain Measure.) When using $F_S(\mathbf{x}, \mathbf{y})$ to bound $F(\mathbf{x}, \mathbf{y})$, we call the this error of a model $(\mathbf{x}, \mathbf{y})$ the plain generalization error.*

*3 (Strong Measures.) The strong primal-dual (PD) population risk of a model $(\mathbf{x}, \mathbf{y})$ is defined as*

$$\triangle^s(\mathbf{x}, \mathbf{y}) = \sup_{\mathbf{y}' \in \mathcal{Y}} F(\mathbf{x}, \mathbf{y}') - \inf_{\mathbf{x}' \in \mathcal{X}} F(\mathbf{x}', \mathbf{y}),$$

*and the corresponding strong PD empirical risk is defined as*

$$\triangle_S^s(\mathbf{x}, \mathbf{y}) = \sup_{\mathbf{y}' \in \mathcal{Y}} F_S(\mathbf{x}, \mathbf{y}') - \inf_{\mathbf{x}' \in \mathcal{X}} F_S(\mathbf{x}', \mathbf{y}).$$

*Then, the strong PD generalization error $\triangle^s(\mathbf{x}, \mathbf{y}) - \triangle_S^s(\mathbf{x}, \mathbf{y})$ of the model $(\mathbf{x}, \mathbf{y})$ is defined as*

$$\left( \sup_{\mathbf{y}' \in \mathcal{Y}} F(\mathbf{x}, \mathbf{y}') - \sup_{\mathbf{y}' \in \mathcal{Y}} F_S(\mathbf{x}, \mathbf{y}') \right) + \left( \inf_{\mathbf{x}' \in \mathcal{X}} F_S(\mathbf{x}', \mathbf{y}) - \inf_{\mathbf{x}' \in \mathcal{X}} F(\mathbf{x}', \mathbf{y}) \right).$$

*4 (Weak Measures.) The weak PD population risk of a (randomized) model $(\mathbf{x}, \mathbf{y})$ is defined as*

$$\triangle^w(\mathbf{x}, \mathbf{y}) = \sup_{\mathbf{y}' \in \mathcal{Y}} \mathbb{E}[F(\mathbf{x}, \mathbf{y}')] - \inf_{\mathbf{x}' \in \mathcal{X}} \mathbb{E}[F(\mathbf{x}', \mathbf{y})],$$

*and the corresponding weak PD empirical risk is defined as*

$$\triangle_S^w(\mathbf{x}, \mathbf{y}) = \sup_{\mathbf{y}' \in \mathcal{Y}} \mathbb{E}[F_S(\mathbf{x}, \mathbf{y}')] - \inf_{\mathbf{x}' \in \mathcal{X}} \mathbb{E}[F_S(\mathbf{x}', \mathbf{y})].$$

*Then, the weak PD generalization error $\triangle^w(\mathbf{x}, \mathbf{y}) - \triangle_S^w(\mathbf{x}, \mathbf{y})$ of the model $(\mathbf{x}, \mathbf{y})$ is defined as*

$$\left( \sup_{\mathbf{y}' \in \mathcal{Y}} \mathbb{E}[F(\mathbf{x}, \mathbf{y}')] - \sup_{\mathbf{y}' \in \mathcal{Y}} \mathbb{E}[F_S(\mathbf{x}, \mathbf{y}')] \right) + \left( \inf_{\mathbf{x}' \in \mathcal{X}} \mathbb{E}[F_S(\mathbf{x}', \mathbf{y})] - \inf_{\mathbf{x}' \in \mathcal{X}} \mathbb{E}[F(\mathbf{x}', \mathbf{y})] \right).$$

**Remark 1.** We provide some discussions for the four groups of measures. (1. Primal Measures:) In the context of GANs, the primal population risk $R(\mathbf{x})$ represents a divergence measure between the learned and true distributions, and in the context of adversarial training it represents the learner's risk under adversarial perturbations (Farnia & Ozdaglar, 2021). One would be interested in the relationship between $R(\mathbf{x})$ and its corresponding empirical risk $R_S(\mathbf{x})$, and the relationship between $R(\mathbf{x})$ and its infimum $\inf_{\mathbf{x}' \in \mathcal{X}} R(\mathbf{x}')$. (2. Plain Measure:) This generalization measure is a direct extension of the standard generalization error in the minimization optimization. (3. Strong Measures:) $\triangle_S^s(\mathbf{x}, \mathbf{y})$ is referred to as the primal-dual gap in the optimization literature. $\triangle^s(\mathbf{x}, \mathbf{y})$ is the primal-dual gap of the population risk. $\triangle^s(\mathbf{x}, \mathbf{y}) - \triangle_S^s(\mathbf{x}, \mathbf{y})$ studies the difference between the population primal-dual gap and its empirical counterpart. (4. Weak Measures:) The difference between the strong and weak measures is that weak measures take the expectation over the randomness of the dataset and the algorithm, for instance, $\sup_{\mathbf{y}' \in \mathcal{Y}} F(\mathbf{x}, \mathbf{y}') - \inf_{\mathbf{x}' \in \mathcal{X}} F(\mathbf{x}', \mathbf{y})$ in the strong measures and $\sup_{\mathbf{y}' \in \mathcal{Y}} \mathbb{E}[F(\mathbf{x}, \mathbf{y}')] - \inf_{\mathbf{x}' \in \mathcal{X}} \mathbb{E}[F(\mathbf{x}', \mathbf{y})]$ in the weak measures. Therefore, the upper bounds of weak measures hold in expectation, while the upper bounds of strong measures hold uniformly for any dataset.

Denote the $L_p$ norm of a random variable $Z$ as $\|Z\|_p = (\mathbb{E}_Z |Z|^p)^{1/p}$. Let $\|\cdot\|$ be the Euclidean norm and $\langle \cdot, \cdot \rangle$ be the inner product. A differentiable function $g : \mathcal{W} \mapsto \mathbb{R}$ is called $\mu$-strongly-convex in $\mathbf{w}$ if the following inequality holds for every $\mathbf{w}_1, \mathbf{w}_2$:

$$g(\mathbf{w}_1) - g(\mathbf{w}_2) \geq \langle \nabla g(\mathbf{w}_2), \mathbf{w}_1 - \mathbf{w}_2 \rangle + \frac{\mu}{2} \|\mathbf{w}_1 - \mathbf{w}_1\|^2,$$

where $\nabla$ is the gradient operator. We say $g$ is $\mu$-strongly-concave if $-g$ is $\mu$-strongly-convex.

**Definition 2.** *Let $g : \mathcal{X} \times \mathcal{Y} \mapsto \mathbb{R}$. Assume that $\mathcal{X}$ and $\mathcal{Y}$ are convex feasible sets. Then*

*1. $g$ is $\mu$-strongly-convex-strongly-concave ($\mu$-SC-SC) if $g(\cdot, \mathbf{y})$ is $\mu$-strongly-convex for any $\mathbf{y} \in \mathcal{Y}$ and $g(\mathbf{x}, \cdot)$ is $\mu$-strongly-concave for any $\mathbf{x} \in \mathcal{X}$.*

*2. $g$ is convex-concave (C-C) if $g$ is 0-SC-SC.*

We then introduce the definition of algorithmic stability this paper used. Algorithmic stability plays an important role in studying the generalization behavior of a learning algorithm. Intuitively, an algorithm $A : \mathcal{Z}^n \mapsto (\mathcal{X}, \mathcal{Y})$ is said to be stable if the output model $(A_\mathbf{x}(S), A_\mathbf{y}(S))$ is insensitive to perturbations. Let $S'$ be a neighboring dataset that differs at most one single example to $S$.

**Definition 3** (Algorithmic Stability). *Let $A$ be a learning algorithm and $\epsilon > 0$.*

*1. We say $A$ is $\epsilon$-uniformly-stable if for any training datasets $S, S' \in \mathcal{Z}^n$ we have*

$$\sup_z [f(A_\mathbf{x}(S), A_\mathbf{y}(S); z) - f(A_\mathbf{x}(S'), A_\mathbf{y}(S'); z)] \leq \epsilon.$$

*2. We say $A$ is $\epsilon$-argument-stable if for any training datasets $S, S' \in \mathcal{Z}^n$ we have*

$$\|A_\mathbf{x}(S) - A_\mathbf{x}(S')\| + \|A_\mathbf{y}(S) - A_\mathbf{y}(S')\| \leq \epsilon.$$

From Definition 3, one can see that the uniform stability measures the sensitivity of the function values, while the argument stability measures the sensitivity of the arguments.

We finally introduce two standard assumptions in minimax problems. Assumption 1 implies $f$ is Lipschitz continuous w.r.t. both $\mathbf{x}$ and $\mathbf{y}$, while Assumption 2 implies $f$ is smooth w.r.t. $(\mathbf{x}, \mathbf{y})$.

**Assumption 1** (Lipschitz continuity). *Let $L > 0$. Assume that for any $\mathbf{x} \in \mathcal{X}$, $\mathbf{y} \in \mathcal{Y}$ and $z \in \mathcal{Z}$, $f(\mathbf{x}, \mathbf{y}; z)$ satisfies*

$$\|\nabla_\mathbf{x} f(\mathbf{x}, \mathbf{y}; z)\| \leq L \quad and \quad \|\nabla_\mathbf{y} f(\mathbf{x}, \mathbf{y}; z)\| \leq L.$$

**Assumption 2** (Smoothness). *Let $\beta > 0$. Assume that for any $\mathbf{x}_1, \mathbf{x}_2 \in \mathcal{X}$, $\mathbf{y}_1, \mathbf{y}_2 \in \mathcal{Y}$ and $z \in \mathcal{Z}$, $f(\mathbf{x}, \mathbf{y}; z)$ satisfies*

$$\left\| \begin{pmatrix} \nabla_\mathbf{x} f(\mathbf{x}_1, \mathbf{y}_1; z) - \nabla_\mathbf{x} f(\mathbf{x}_2, \mathbf{y}_2; z) \\ \nabla_\mathbf{y} f(\mathbf{x}_1, \mathbf{y}_1; z) - \nabla_\mathbf{y} f(\mathbf{x}_2, \mathbf{y}_2; z) \end{pmatrix} \right\| \leq \beta \left\| \begin{pmatrix} \mathbf{x}_1 - \mathbf{x}_2 \\ \mathbf{y}_1 - \mathbf{y}_2 \end{pmatrix} \right\|.$$

Under Assumption 1, the argument stability implies the uniform stability. Therefore, the argument stability is the main stability measure that we will focus on.

## 4 MAIN RESULTS

In this section, we provide sharper high probability bounds for the generalization measures of Definition 1, shown as follows.

**Theorem 1.** *Let $A$ be a learning algorithm and $\epsilon > 0$. Suppose $|f(\mathbf{x}, \mathbf{y}; z)| \leq M$ for some $M > 0$ and $\mathbf{x} \in \mathcal{X}, \mathbf{y} \in \mathcal{Y}, z \in \mathcal{Z}$. Fixed any $\eta > 0$. There exists an absolute positive constant $C$.*

*(a.) If the algorithm $A$ is $\epsilon$-uniformly stable, then for any $\delta > 0$, with probability at least $1 - \delta$,*

$$F(A_{\mathbf{x}}(S), A_{\mathbf{y}}(S)) \leq (1 + \eta)F_S(A_{\mathbf{x}}(S), A_{\mathbf{y}}(S)) + C\frac{1 + \eta}{\eta}\Big(\frac{M}{n}\log(1/\delta) + \epsilon \log_2 n \log\frac{1}{\delta}\Big).$$

*(b.) Assume that for all $\mathbf{x}$, the function $\mathbf{y} \mapsto F(\mathbf{x}, \mathbf{y})$ is $\mu$-strongly-concave. If the algorithm $A$ is $\epsilon$-argument stable and Assumptions 1 and 2 hold, then for any $\delta > 0$, with probability at least $1 - \delta$,*

$$R(A_{\mathbf{x}}(S)) \leq (1 + \eta)R_S(A_{\mathbf{x}}(S)) + C\frac{1 + \eta}{\eta}\Big(\frac{M}{n}\log\frac{1}{\delta} + \Big(\frac{\beta}{\mu} + 1\Big)L\epsilon\log_2 n \log\frac{1}{\delta}\Big).$$

*(c.) Assume that for all $\mathbf{x}$ and $\mathbf{y}$, the function $F(\mathbf{x}, \mathbf{y})$ is $\mu$-SC-SC. If the algorithm $A$ is $\epsilon$-argument stable and Assumptions 1 and 2 hold, then for any $\delta > 0$, with probability at least $1 - \delta$,*

$$\triangle^s (A_{\mathbf{x}}(S), A_{\mathbf{y}}(S)) \leq \triangle_S^s(A_{\mathbf{x}}(S), A_{\mathbf{y}}(S)) + \eta\mathbb{E}_S \triangle_S^s (A_{\mathbf{x}}(S), A_{\mathbf{y}}(S))$$
$$+ C(1 + \eta)\Big(\frac{L^2(1 + \eta)}{n\mu\eta} + \frac{M}{n} + \Big(1 + \frac{\beta}{\mu}\Big)\epsilon L \log_2 n\Big)\log\Big(\frac{1}{\delta}\Big).$$

*(d.) Assume that for all $\mathbf{x}$ and $\mathbf{y}$, the function $F(\mathbf{x}, \mathbf{y})$ is $\mu$-SC-SC. If the algorithm $A$ is $\epsilon$-argument stable and Assumptions 1 and 2 hold, then for any $\delta > 0$, with probability at least $1 - \delta$,*

$$\triangle^s (A_{\mathbf{x}}(S), A_{\mathbf{y}}(S)) - \triangle_S^s(A_{\mathbf{x}}(S), A_{\mathbf{y}}(S)) \leq \eta\mathbb{E}_S \triangle_S^s (A_{\mathbf{x}}(S), A_{\mathbf{y}}(S))$$
$$+ C(1 + \eta)\Big(\frac{L^2(1 + \eta)}{n\mu\eta} + \frac{M}{n} + \Big(1 + \frac{\beta}{\mu}\Big)\epsilon L \log_2 n\Big)\log\Big(\frac{1}{\delta}\Big).$$

*(e.) Assume that for all $\mathbf{x}$, the function $\mathbf{y} \mapsto F(\mathbf{x}, \mathbf{y})$ is $\mu$-strongly-concave. If the algorithm $A$ is $\epsilon$-argument stable and Assumptions 1 and 2 hold, then for any $\delta > 0$, with probability at least $1 - \delta$,*

$$R(A_{\mathbf{x}}(S)) \leq (1 + \eta)\inf_{\mathbf{x} \in \mathcal{X}} R(\mathbf{x})$$
$$+ C\frac{2 + \eta}{\eta}\Big(\frac{M}{n}\log\frac{1}{\delta} + \Big(\frac{\beta}{\mu} + 1\Big)L\epsilon\log_2 n \log\frac{1}{\delta} + \triangle_S^s(A_{\mathbf{x}}(S), A_{\mathbf{y}}(S))\Big).$$

According to Definition 1 and Jensen's inequality, we know that $\triangle^w(\mathbf{x}, \mathbf{y}) \leq \mathbb{E}[\triangle^s(\mathbf{x}, \mathbf{y})]$ and $\triangle_S^w(\mathbf{x}, \mathbf{y}) \leq \mathbb{E}[\triangle_S^s(\mathbf{x}, \mathbf{y})]$. By this connection, we have $\triangle^w(\mathbf{x}, \mathbf{y}) - \triangle_S^w(\mathbf{x}, \mathbf{y}) \leq \mathbb{E}[\triangle^s(\mathbf{x}, \mathbf{y})] + |\mathbb{E}[\triangle_S^s(\mathbf{x}, \mathbf{y})]|$. We therefore obtain the following results for $\triangle^w(\mathbf{x}, \mathbf{y})$ and $\triangle^w(\mathbf{x}, \mathbf{y}) - \triangle_S^w(\mathbf{x}, \mathbf{y})$.

**Corollary 1.** *Suppose the same conditions as Theorem 1 hold.*

*(f.) If the assumptions of Part (c) in Theorem 1 hold, then with probability at least $1 - \delta$,*

$$\triangle^w (A_{\mathbf{x}}(S), A_{\mathbf{y}}(S)) \leq (1 + \eta)\mathbb{E} \triangle_S^s (A_{\mathbf{x}}(S), A_{\mathbf{y}}(S))$$
$$+ C(1 + \eta)\Big(\frac{L^2(1 + \eta)}{n\mu\eta} + \frac{M}{n} + \Big(1 + \frac{\beta}{\mu}\Big)\epsilon L \log_2 n\Big)\log\Big(\frac{1}{\delta}\Big).$$

*(g.) If the assumptions of Part (d) in Theorem 1 hold, then with probability at least $1 - \delta$,*

$$\triangle^w (A_{\mathbf{x}}(S), A_{\mathbf{y}}(S)) - \triangle_S^w(A_{\mathbf{x}}(S), A_{\mathbf{y}}(S)) \leq |\mathbb{E} \triangle_S^s (A_{\mathbf{x}}(S), A_{\mathbf{y}}(S))|$$
$$+ (1 + \eta)\mathbb{E} \triangle_S^s (A_{\mathbf{x}}(S), A_{\mathbf{y}}(S)) + C(1 + \eta)\Big(\frac{L^2(1 + \eta)}{n\mu\eta} + \frac{M}{n} + \Big(1 + \frac{\beta}{\mu}\Big)\epsilon L \log_2 n\Big)\log\Big(\frac{1}{\delta}\Big).$$

**Remark 2.** In Theorem 1, we have established a quantitative connection between the generalization measures and the stability measures. The complete proof of Theorem 1 is provided in Appendix A.

Part (a) provides the relationship between the uniform stability and the plain generalization error of $(A_{\mathbf{x}}(S), A_{\mathbf{y}}(S))$. If the uniform stability of algorithm $A$ is of fast order $\mathcal{O}(1/n)$, then $F(A_{\mathbf{x}}(S), A_{\mathbf{y}}(S))$ is bounded by $(1 + \eta)\mathcal{O}\big(F_S(A_{\mathbf{x}}(S), A_{\mathbf{y}}(S)) + \frac{\log n \log(1/\delta)}{n\eta}\big)$. Usually for a well-trained model $(A_{\mathbf{x}}(S), A_{\mathbf{y}}(S))$ over the training set, the empirical risk $F_S(A_{\mathbf{x}}(S), A_{\mathbf{y}}(S))$ is small or even zero (Lever et al., 2013; Yang et al., 2019; Cortes et al., 2021). If the empirical risk is of order $\mathcal{O}(1/n)$, then we can choose a proper constant for $\eta$ and the plain generalization error will be of fast order $\mathcal{O}\big(\frac{\log n \log(1/\delta)}{n}\big)$. It is $\mathcal{O}(1/n)$ when we hide the logarithmic term. In the related work, (Lei et al., 2021) also establish the plain generalization error bound under the same assumptions, but their bound is of slow order $\mathcal{O}\big(\epsilon \log n \log(1/\delta) + Mn^{-\frac{1}{2}}\sqrt{\log(1/\delta)}\big)$. Even if they get a sharper bound for stability measure $\epsilon$, the influence of $\mathcal{O}\big(n^{-\frac{1}{2}}\sqrt{\log(1/\delta)}\big)$ can not disappear. By comparison, we have completely removed the $\mathcal{O}(1/\sqrt{n})$ term. Thus, our plain generalization error bound enables the fast $\mathcal{O}(1/n)$ rate when the empirical risk is small.

Part (b) provides the connection between the argument stability and the primal generalization error. Similar to the analysis of Part (a), if both the argument stability of algorithm $A$ and $R_S(A_{\mathbf{x}}(S))$ are of order $\mathcal{O}(1/n)$, then the primal generalization error implies a fast $\mathcal{O}(1/n)$ rate. Considering that we assume the function $f$ is well-behaved, i.e., Lipschitz continuity, smoothness, and the strong-concavity of its population risk $F$, and that $R_S(A_{\mathbf{x}}(S))$ is data-dependent, thus it is reasonable to assume $R_S(A_{\mathbf{x}}(S))$ is small for a well-trained model $A_{\mathbf{x}}(S)$ (Lever et al., 2013; Yang et al., 2019; Cortes et al., 2021). In (Lei et al., 2021), they also establish a bound for primal generalization error under the same assumptions. However, their bound is $\mathcal{O}\big(L\beta\mu^{-1}\epsilon logn \log(1/\delta) + Mn^{-\frac{1}{2}}\sqrt{\log(1/\delta)}\big)$, limited to the $\mathcal{O}(1/\sqrt{n})$ order. In contrast, we successfully removed the $\mathcal{O}(1/\sqrt{n})$ term, which makes the fast rate possible. (Farnia & Ozdaglar, 2021) studies the expected primal generalization error, i.e., bounding $\mathbb{E}_{S,A}[R(A_{\mathbf{x}}(S))]$ by $\mathbb{E}_{S,A}[R_S(A_{\mathbf{x}}(S))]$. They establish the connection between the stability measure and the expected error under the same assumptions as Part (b), which is then used to derive generalization bounds for (S)GDA, (S)GDmax, and PPM algorithms. By comparison, our bound is derived in high probability.

Part (c) provides the relationship between the argument stability and the strong PD population risk. If both the argument stability of algorithm $A$ and the strong PD empirical risk are all of the order $\mathcal{O}(1/n)$, the strong PD population risk will be of the fast order $\mathcal{O}(1/n)$. Note that in our proof for the gradient-based optimization algorithms, the strong PD empirical risk mainly has a dependence on the iterative number $T$ (see Lemma 8 of GDA, Lemma 11 of SGDA, etc.). To obtain sharper generalization bounds, we require $T$ to be associated with $n$, such as $T = \mathcal{O}(n^2)$ for GDA, the strong PD empirical risk finally has a dependence on the sample size $n$. To our best knowledge, this is the first high probability strong PD population risk bound. The expected version of this risk is studied for the ESP solution in (Zhang et al., 2021a). However, the discussion there does not establish the connection between stability and generalization. Their analysis is restricted to the specific ESP problem. Under the same assumptions, they provide the upper bound of order $\mathcal{O}(1/n)$. Compared with their result, our result is presented in high probability. Additionally, our strong PD population risk bound is applicable for any stable minimax optimization algorithms.

Part (d) provides the connection between the argument stability and the strong PD generalization error. Similarly, if both the argument stability of algorithm $A$ and the strong PD empirical risk are all of $\mathcal{O}(1/n)$ order, the strong PD generalization error will be of the fast order $\mathcal{O}(1/n)$. Although Part (c) and Part (d) have a similar upper bound, they are different generalization measures (Lei et al., 2021). To our best knowledge, this is also the first high probability strong PD generalization error bound. The expected version of this generalization error is studied in (Lei et al., 2021), that is $\mathbb{E}_{S,A}[\triangle^s(A_{\mathbf{x}}(S), A_{\mathbf{y}}(S)) - \triangle_S^s(A_{\mathbf{x}}(S), A_{\mathbf{y}}(S))]$. Under the same assumptions, their expected strong PD generalization error is bounded by $(1 + \beta/\mu)L\sqrt{2}\epsilon$, which can also be used to obtain $\mathcal{O}(1/n)$ order rate when $\epsilon$ is of order $\mathcal{O}(1/n)$. However, this bound is provided for the expected error, while our bound is high probabilistic and holds uniformly for any dataset.

Part (e) provides the relationship between the argument stability and the excess primal population risk. Similar to the analysis of Part (a) and Part (b), if the argument stability of algorithm $A$, the strong PD empirical risk, and $\inf_{\mathbf{x} \in \mathcal{X}} R(\mathbf{x})$ are all of the order $\mathcal{O}(1/n)$, the excess primal population

risk will also be of the fast order $\mathcal{O}(1/n)$. Meanwhile, in the minimization learning problems, assuming the optimal population risk $F^*$ is small or even zero, i.e., $F^* \leq \mathcal{O}(1/n)$, can be found in (Lei & Ying, 2021a; Zhang et al., 2017; Zhang & Zhou, 2019; Srebro et al., 2010; Lei & Ying, 2020). Note that the optimal population risk $F^* = O(1/n)$ just to show that the improved bound can be got under low noise conditions. $F^*$ should be independent of $n$. Similar to the assumption on $F^*$ and considering that we assume the function $f$ is well-behaved, it will also be reasonable to assume $\inf_{\mathbf{x} \in \mathcal{X}} R(\mathbf{x})$ is small. High probability excess primal population risk bound is also studied for SGDA in (Lei et al., 2021). Their bound, however, is of slow order $\mathcal{O}\big((\beta/\mu)n^{-\frac{1}{2}} \log n \log^2(1/\delta)\big)$ and is restricted to SGDA. By comparison, our result in Part (e) enables $\mathcal{O}(1/n)$ bounds for stable minimax learning algorithms since we have completely removed the $\mathcal{O}(1/\sqrt{n})$ term.

We discuss a noteworthy difference between (Farnia & Ozdaglar, 2021; Lei et al., 2021) and ours. In Part (a), Part (b), and Part (e), we study the upper bounds of $F(A_{\mathbf{x}}(S), A_{\mathbf{y}}(S))$, $R(A_{\mathbf{x}}(S))$ (w.r.t. $R_S(A_{\mathbf{x}}(S))$), and $R(A_{\mathbf{x}}(S))$ (w.r.t. $\inf_{\mathbf{x} \in \mathcal{X}} R(\mathbf{x})$), respectively, while (Lei et al., 2021; Farnia & Ozdaglar, 2021) study the upper bounds of $F(A_{\mathbf{x}}(S), A_{\mathbf{y}}(S)) - F_S(A_{\mathbf{x}}(S), A_{\mathbf{y}}(S))$, $R(A_{\mathbf{x}}(S)) - R_S(A_{\mathbf{x}}(S))$, and $R(A_{\mathbf{x}}(S)) - \inf_{\mathbf{x} \in \mathcal{X}} R(\mathbf{x})$ (or their expected forms). One of our motivations to study such forms is that, in practice, we are often directly interested in the true risk, i.e., how the learned models behave on the testing data, such as $F(A_{\mathbf{x}}(S), A_{\mathbf{y}}(S))$, instead of the error between the true risk and empirical risk. Note that in the above comparison between Theorem 1 and the results in (Lei et al., 2021; Farnia & Ozdaglar, 2021), we all take the right side of the generalization bound inequalities to compare, which is fair since our bounds can be written as $F(A_{\mathbf{x}}(S), A_{\mathbf{y}}(S)) - F_S(A_{\mathbf{x}}(S), A_{\mathbf{y}}(S)) \leq \eta F_S(A_{\mathbf{x}}(S), A_{\mathbf{y}}(S)) + C\frac{1+\eta}{\eta}(\frac{M}{n} \log(1/\delta) + \epsilon \log_2 n \log \frac{1}{\delta})$, etc.

**Remark 3.** From Remark 2, one can see that compared with (Zhang et al., 2021a; Farnia & Ozdaglar, 2021; Lei et al., 2021), we have established sharper high probability generalization bounds. In the applications of Section 5, we will establish $\mathcal{O}(1/n)$ order bounds for two terms in Theorem 1: stability measures and strong PD empirical risk. Hence, the strong PD population risk and the strong PD generalization error will be of the fast order $\mathcal{O}(1/n)$ when applying Theorem 1 to these applications. These bounds are clearly of order $\mathcal{O}(1/n)$ and sharper than the results in (Zhang et al., 2021a; Farnia & Ozdaglar, 2021; Lei et al., 2021). For the plain generalization error, the primal generalization error, and the excess primal population risk, to obtain $\mathcal{O}(1/n)$ order bounds for these applications, we need to assume the extra corresponding terms $F(A_{\mathbf{x}}(S), A_{\mathbf{y}}(S))$, $R(A_{\mathbf{x}}(S))$, and $\inf_{\mathbf{x} \in \mathcal{X}} R(\mathbf{x})$ are of order $\mathcal{O}(1/n)$, respectively. The clear motivation is that in practice, learning algorithms achieve a small or even zero empirical risk, as discussed in Remark 2.

**Remark 4.** This remark discusses $\eta$ in Part (a), Part (b), and Part (e). (1:) When establishing sharper generalization error bound (i.e., $Pf - P_n f$), the existence of $\eta$ is common in the standard statistical learning theory. Specifically, in the uniform localized convergence theory, the generalization error bound in (Bartlett et al., 2005) is of the form $Pf \leq \frac{\eta}{\eta-1} P_n f + \mathcal{O}(\eta r^* + \frac{\eta \log(1/\delta)}{n})$ with $\eta > 1$ (see Theorem 3.3 and Theorem 4.1). In the PAC-Bayesian theory, the generalization bounds in (Catoni, 2007) (see Theorem 1.2.6), (Lever et al., 2013) (see Theorem 6), (Yang et al., 2019) (see Proposition 3.1 and Theorem 4.3), etc., also have $\eta$. For instance, the Catoni's bound is of the form $PQ \leq \frac{1}{1-e^{-\eta}}\big(\eta P_n Q + \mathcal{O}(\frac{KL(Q\|Prior)+\log(1/\delta)}{n})\big)$ with $\eta > 0$ (Catoni, 2007). In the algorithmic stability theory, Theorem 1.2 in (Klochkov & Zhivotovskiy, 2021) is of the form $Pf \leq (1+\eta)P_n f + \frac{1+\eta}{\eta} \mathcal{O}((\epsilon \log n + \frac{1}{n}) \log(\frac{1}{\delta}))$ with $\eta > 0$. In the recent Cortes's deviation margin bounds (Cortes et al., 2021), they also imply a multiplier $\eta$. The above bounds can be transformed into the form of empirical risk multiplied by $1 + \eta$, similar to our results. It is discussed in (Lever et al., 2013; Yang et al., 2019; Cortes et al., 2021; Bartlett et al., 2005; Klochkov & Zhivotovskiy, 2021) that this type of generalization error bound can obtain a fast rate when the empirical risk is small. Note that Part (e) also involves generalization error bounds due to the decomposition, see (31). The above generalization error analysis thus holds for Part (e). (2:) Furthermore, in (10), we show that $F(A_{\mathbf{x}}(S), A_{\mathbf{y}}(S)) - F_S(A_{\mathbf{x}}(S), A_{\mathbf{y}}(S)) \leq \mathcal{O}\big((\frac{MF(A_{\mathbf{x}}(S), A_{\mathbf{y}}(S)) \log(1/\delta)}{n})^{\frac{1}{2}} + \epsilon \log(\frac{1}{\delta})\big)$, where $M$ means that $|f(\mathbf{x}, \mathbf{y}; z)| \leq M, \forall \mathbf{x}, \mathbf{y}, z$. Using the elementary inequality $\sqrt{ab} \leq \eta a + \frac{1}{\eta} b$ for any $a, b, \eta > 0$ and by some rearrangements, the form of Part (a) appears. This is the reason why $\eta$ exists. The corresponding bound in (Lei et al., 2021) is $F(A_{\mathbf{x}}(S), A_{\mathbf{y}}(S)) - F_S(A_{\mathbf{x}}(S), A_{\mathbf{y}}(S)) \leq \mathcal{O}\big(\epsilon \log n \log(\frac{1}{\delta}) + M n^{-\frac{1}{2}} \log^{1/2}(\frac{1}{\delta})\big)$. Focusing on the dominated term, it is clear that $F(A_{\mathbf{x}}(S), A_{\mathbf{y}}(S)) \ll M$ since $F(A_{\mathbf{x}}(S), A_{\mathbf{y}}(S))$ is data-dependent, which implies that our plain generalization error bound is sharper. Similar analysis holds for Part (b) and Part (e).

| Reference | Algorithm | Assumption | Generalization Measure | Learning Bound |
|---|---|---|---|---|
| Zhang | ESP | SC-SC, Lip | Weak PD Risk | $\mathcal{O}(1/n)$ |
| | | SC-SC, Lip, S | (E.) Strong PD Risk | $\mathcal{O}(1/n)$ |
| | R-ESP | C-C, Lip | Weak PD Risk | $\mathcal{O}(1/\sqrt{n})$ |
| Farnia | SGDA | SC-SC, Lip, S | (E.) Primal generalization | $\mathcal{O}(1/n)$ |
| | SGDmax | SC-SC, Lip, S | (E.) Primal generalization | $\mathcal{O}(1/n)$ |
| | GDA | SC-SC, Lip, S | (E.) Primal generalization | $\mathcal{O}(1/n)$ |
| | GDmax | SC-SC, Lip, S | (E.) Primal generalization | $\mathcal{O}(1/n)$ |
| | PPM | SC-SC, Lip, S | (E.) Primal generalization | $\mathcal{O}(1/n)$ |
| | PPM | C-C, Lip, S | (E.) Primal generalization | $\mathcal{O}(1/\sqrt{n})$ |
| | SGDA | Lip, S | (E.) Primal generalization | $\mathcal{O}\big(T^{\frac{\beta c}{\beta c+1}}/n\big)$ |
| | SGDmax | NC-SC, Lip, S | (E.) Primal generalization | $\mathcal{O}\big(T^{\frac{(k+1)\beta c}{(k+10L\beta+1)}}/n\big)$ |
| Lei | SGDA | C-C, Lip | Weak PD Risk | $\mathcal{O}(1/\sqrt{n})$ |
| | | C-C, Lip, S | Weak PD Risk | $\mathcal{O}(1/\sqrt{n})$ |
| | | SC-SC, Lip | Weak PD Risk | $\mathcal{O}(\sqrt{\log n}/n)$ |
| | | SC-SC, Lip, S | Weak PD Risk | $\mathcal{O}(\log n/n)$ |
| | | C-SC, Lip, S | (E.) Excess Primal Risk | $\mathcal{O}(1/\sqrt{n})$ |
| | | C-SC, Lip, S | (H.P.) Excess Primal Risk | $\mathcal{O}(\log n/\sqrt{n})$ |
| | | C-C, Lip | (H.P.) Plain Generalization | $\mathcal{O}(\log n/\sqrt{n})$ |
| | | WC-WC, Lip | Weak PD Generalization | $\mathcal{O}\big(T^{\frac{2c\mu}{2c\mu+3}}/n^{\frac{2c\mu+1}{2c\mu+3}}\big)$ |
| | | V-WC-WC, Lip, S | Weak PD Generalization | $\mathcal{O}\big(1/\sqrt{n}\big)$ |
| | AGDA | NC-SC, PL, Lip, S | (E.) Excess Primal Risk | $\mathcal{O}\big(n^{-\frac{c\beta+1}{2c\beta+1}}\big)$ |
| Ours | ESP | SC-SC, Lip, LN | Plain Generalization | $\mathcal{O}(\log n/n)$ |
| | | SC-SC, Lip, S, LN | Primal Generalization | $\mathcal{O}(\log n/n)$ |
| | | SC-SC, Lip, S | Strong PD Risk | $\mathcal{O}(\log n/n)$ |
| | | SC-SC, Lip, S | Strong PD Generalization | $\mathcal{O}(\log n/n)$ |
| | | SC-SC, Lip, S, LN | Excess Primal Risk | $\mathcal{O}(\log n/n)$ |
| | GDA | SC-SC, Lip, LN | Plain Generalization | $\mathcal{O}((\log n)^{3/2}/n)$ |
| | | SC-SC, Lip, S, LN | Primal Generalization | $\mathcal{O}((\log n)^{3/2}/n)$ |
| | | SC-SC, Lip, S | Strong PD Risk | $\mathcal{O}((\log n)^{3/2}/n)$ |
| | | SC-SC, Lip, S | Strong PD Generalization | $\mathcal{O}((\log n)^{3/2}/n)$ |
| | | SC-SC, Lip, S, LN | Excess Primal Risk | $\mathcal{O}((\log n)^{3/2}/n)$ |
| | SGDA | SC-SC, Lip, LN | Plain Generalization | $\mathcal{O}(\log n/n)$ |
| | | SC-SC, Lip, S, LN | Primal Generalization | $\mathcal{O}(\log n/n)$ |
| | | SC-SC, Lip, S | Strong PD Risk | $\mathcal{O}(\log n/n)$ |
| | | SC-SC, Lip, S | Strong PD Generalization | $\mathcal{O}(\log n/n)$ |
| | | SC-SC, Lip, S, LN | Excess Primal Risk | $\mathcal{O}(\log n/n)$ |
| | PPM | SC-SC, Lip, S, LN | Plain Generalization | $\mathcal{O}(\log n/n)$ |
| | | SC-SC, Lip, S, LN | Primal Generalization | $\mathcal{O}(\log n/n)$ |
| | | SC-SC, Lip, S | Strong PD Risk | $\mathcal{O}(\log n/n)$ |
| | | SC-SC, Lip, S | Strong PD Generalization | $\mathcal{O}(\log n/n)$ |
| | | SC-SC, Lip, S, LN | Excess Primal Risk | $\mathcal{O}(\log n/n)$ |
| | EG | SC-SC, Lip, S, LN | Plain Generalization | $\mathcal{O}(\log n/n)$ |
| | | SC-SC, Lip, S, LN | Primal Generalization | $\mathcal{O}(\log n/n)$ |
| | | SC-SC, Lip, S | Strong PD Risk | $\mathcal{O}(\log n/n)$ |
| | | SC-SC, Lip, S | Strong PD Generalization | $\mathcal{O}(\log n/n)$ |
| | | SC-SC, Lip, S, LN | Excess Primal Risk | $\mathcal{O}(\log n/n)$ |
| | OGDA | SC-SC, Lip, S, LN | Plain Generalization | $\mathcal{O}(\log n/n)$ |
| | | SC-SC, Lip, S, LN | Primal Generalization | $\mathcal{O}(\log n/n)$ |
| | | SC-SC, Lip, S | Strong PD Risk | $\mathcal{O}(\log n/n)$ |
| | | SC-SC, Lip, S | Strong PD Generalization | $\mathcal{O}(\log n/n)$ |
| | | SC-SC, Lip, S, LN | Excess Primal Risk | $\mathcal{O}(\log n/n)$ |

Table 1: Summary of Results. Here, "Zhang" means reference (Zhang et al., 2021a), "Farnia" means reference (Farnia & Ozdaglar, 2021), and "Lei" means reference (Lei et al., 2021). The bounds are established by choosing an optimal iterate number $T$. "LN" means the low noise condition, see Section 5.1. Other auxiliary descriptions of Table 1 are shown in Appendix H.

In summary, the above analyses from two different perspectives support our claim that Part (a), Part (b), and Part (e) provide sharper high probability generalization bounds.

**Remark 5.** Different measures quantify different degrees of the generalization error. Thus, deriving bounds of different generalization measures requires different assumptions (Lei et al., 2021; Zhang et al., 2021a; Farnia & Ozdaglar, 2021). Strong measures require stronger assumptions compared with the weak (Lei et al., 2021). For instance, for the term $\sup_{\mathbf{y} \in \mathcal{Y}} F(A_{\mathbf{x}}(S), \mathbf{y})$ in $\triangle^s(A_{\mathbf{x}}(S), A_{\mathbf{y}}(S))$, one has to consider the fact that for different $A_{\mathbf{x}}(S)$, $\mathbf{y}$ is different, which makes the proof more challenging. While in $\triangle^w(A_{\mathbf{x}}(S), A_{\mathbf{y}}(S))$ and $\triangle^w(A_{\mathbf{x}}(S), A_{\mathbf{y}}(S)) - \triangle^w_S(A_{\mathbf{x}}(S), A_{\mathbf{y}}(S))$, both the supremum over $A_{\mathbf{x}}(S)$ and $A_{\mathbf{y}}(S)$ are outside the expectation operator, thus one does not need to consider the coupling between $A_{\mathbf{x}}(S)$ and $\mathbf{y}$. The upper bounds shown in Corollary 1 directly derived from Theorem 1 are sub-optimal since $\triangle^w(A_{\mathbf{x}}(S), A_{\mathbf{y}}(S))$ and $\triangle^w(A_{\mathbf{x}}(S), A_{\mathbf{y}}(S)) - \triangle^w_S(A_{\mathbf{x}}(S), A_{\mathbf{y}}(S))$ are pretty weak generalization measures (Lei et al., 2021). We list Corollary 1 here to suggest that, when Theorem 1 is established, the fast order $\mathcal{O}(1/n)$ is easy to be achieved for $\triangle^w(A_{\mathbf{x}}(S), A_{\mathbf{y}}(S))$ and $\triangle^w(A_{\mathbf{x}}(S), A_{\mathbf{y}}(S)) - \triangle^w_S(A_{\mathbf{x}}(S), A_{\mathbf{y}}(S))$. On the other hand, the two weak generalization measures in (Zhang et al., 2021a) is studied for the specific ESP solution, while Corollary 1 is applicable for any stable minimax learning algorithms, it thus may be useful in some applications.

## 5 APPLICATIONS

We now apply Theorem 1 to the ESP solution and several gradient-based optimization algorithms: GDA, SGDA, PPM, EG, and OGDA. Considering the length limit, we postpone the introductions and theorems of these applications to the Appendix. Here, we list the generalization bounds of these optimization algorithms in Table 1.

### 5.1 DESCRIPTIONS OF TABLE 1

Table 1 gives almost all existing generalization bounds in minimax learning. In Table 1, "LN" means the low noise conditions, i.e., the corresponding $F(A_{\mathbf{x}}(S), A_{\mathbf{y}}(S))$, $R(A_{\mathbf{x}}(S))$, or $\inf_{\mathbf{x} \in \mathcal{X}} R(\mathbf{x})$ of these applications is of the order $\mathcal{O}(1/n)$. For instance, for the ESP solution $(\hat{\mathbf{x}}^*_S, \hat{\mathbf{y}}^*_S)$, we assume $F(\hat{\mathbf{x}}^*_S, \hat{\mathbf{y}}^*_S)$, $R(\hat{\mathbf{x}}^*_S)$, and $\inf_{\mathbf{x} \in \mathcal{X}} R(\mathbf{x})$ are of the order $\mathcal{O}(1/n)$ for the plain generalization error, the primal generalization error, and the excess primal population risk, respectively. For other learning algorithms, please refer to the Remarks in the Appendix. In Table 1, we compare our results with (Zhang et al., 2021a; Farnia & Ozdaglar, 2021; Lei et al., 2021) in the way described in the last paragraph of Remark 2. (E.) denotes that the bound is derived in expectation, while (H.P.) denotes high probability. Since our results are all established with high probability, we thus omit (H.P.) for brevity. The descriptions of other notations are shown in Appendix H.

In Table 1, (Zhang et al., 2021a) and (Farnia & Ozdaglar, 2021) focus on the expected generalization measures. We improve the learning bounds of the ESP solution in (Zhang et al., 2021a) to high probability guarantees. Compared with (Farnia & Ozdaglar, 2021), we have provided high probability primal generalization error bounds for GDA, SGDA, and PPM. Additionally, we also study other generalization measures. (Lei et al., 2021) focus on SGDA and mainly provide guarantees for weak generalization measures, i.e., weak PD risk and weak PD generalization error. In contrast, we have developed bounds for strong PD risk and strong PD generalization error. Note that the two type of bounds don't require the "LN" condition. Moreover, although (Lei et al., 2021) provides two high probability bounds, however, in slow order. Note that in addition to the classical GDA, SGDA, and PPM, we also provide sharper high probability bounds for EG and OGDA in that their widespread use in training GANs (Mokhtari et al., 2019; Daskalakis et al., 2017; Liang & Stokes, 2019).

## 6 CONCLUSION

In this paper, we provide a systematical analysis of sharper generalization bounds for minimax problems. We first establish sharper high probability bounds for almost all existing generalization measures via algorithmic stability and then apply these bounds to several important applications. We believe that our research can provide in-depth insights into minimax learning problems. For future work, it would be important to relax the assumptions in this paper. Also, it would be interesting to investigate how well other theoretical tools perform on the generalization of minimax problems.

## ACKNOWLEDGMENTS

We appreciate all the anonymous reviewers for their invaluable and constructive comments. This work is supported in part by the National Natural Science Foundation of China (No. 62076234, No.61703396, No. 62106257), Beijing Outstanding Young Scientist Program NO.BJJWZYJH012019100020098, Intelligent Social Governance Platform, Major Innovation & Planning Interdisciplinary Platform for the "Double-First Class" initiative, Renmin University of China, China Unicom Innovation Ecological Cooperation Plan, Public Computing Cloud of Renmin University of China, Beijing Natural Science Foundation (No. 4222029).

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

## A    PROOF OF THEOREM 1

We now provide proofs of Theorem 1. For better readability, we restate Theorem 1 below.

**Theorem 2.** *Let $A$ be a learning algorithm and $\epsilon > 0$. Suppose $|f(\mathbf{x}, \mathbf{y}; z)| \leq M$ for some $M > 0$ and $\mathbf{x} \in \mathcal{X}, \mathbf{y} \in \mathcal{Y}, z \in \mathcal{Z}$. Fixed any $\eta > 0$. There exists an absolute positive constant $C$.*

*(a.) If the algorithm $A$ is $\epsilon$-uniformly stable, then for any $\delta > 0$, with probability at least $1 - \delta$,*

$$F(A_{\mathbf{x}}(S), A_{\mathbf{y}}(S)) \leq (1 + \eta)F_S(A_{\mathbf{x}}(S), A_{\mathbf{y}}(S)) + C\frac{1+\eta}{\eta}\Big(\frac{M}{n}\log(1/\delta) + \epsilon \log_2 n \log\frac{1}{\delta}\Big).$$

*(b.) Assume that for all $\mathbf{x}$, the function $\mathbf{y} \mapsto F(\mathbf{x}, \mathbf{y})$ is $\mu$-strongly-concave. If the algorithm $A$ is $\epsilon$-argument stable and Assumptions 1 and 2 hold, then for any $\delta > 0$, with probability at least $1 - \delta$,*

$$R(A_{\mathbf{x}}(S)) \leq (1 + \eta)R_S(A_{\mathbf{x}}(S)) + C\frac{1+\eta}{\eta}\Big(\frac{M}{n}\log\frac{1}{\delta} + \Big(\frac{\beta}{\mu} + 1\Big)L\epsilon \log_2 n \log\frac{1}{\delta}\Big).$$

*(c.) Assume that for all $\mathbf{x}$ and $\mathbf{y}$, the function $F(\mathbf{x}, \mathbf{y})$ is $\mu$-SC-SC. If the algorithm $A$ is $\epsilon$-argument stable and Assumptions 1 and 2 hold, then for any $\delta > 0$, with probability at least $1 - \delta$,*

$$\triangle^s(A_{\mathbf{x}}(S), A_{\mathbf{y}}(S)) \leq \triangle^s_S(A_{\mathbf{x}}(S), A_{\mathbf{y}}(S)) + \eta \mathbb{E}_S \triangle^s_S(A_{\mathbf{x}}(S), A_{\mathbf{y}}(S))$$
$$+ C(1 + \eta)\Big(\frac{L^2(1+\eta)}{n\mu\eta} + \frac{M}{n} + \Big(1 + \frac{\beta}{\mu}\Big)\epsilon L \log_2 n\Big)\log\Big(\frac{1}{\delta}\Big).$$

*(d.) Assume that for all $\mathbf{x}$ and $\mathbf{y}$, the function $F(\mathbf{x}, \mathbf{y})$ is $\mu$-SC-SC. If the algorithm $A$ is $\epsilon$-argument stable and Assumptions 1 and 2 hold, then for any $\delta > 0$, with probability at least $1 - \delta$,*

$$\triangle^s(A_{\mathbf{x}}(S), A_{\mathbf{y}}(S)) - \triangle^s_S(A_{\mathbf{x}}(S), A_{\mathbf{y}}(S)) \leq \eta \mathbb{E}_S \triangle^s_S(A_{\mathbf{x}}(S), A_{\mathbf{y}}(S))$$
$$+ C(1 + \eta)\Big(\frac{L^2(1+\eta)}{n\mu\eta} + \frac{M}{n} + \Big(1 + \frac{\beta}{\mu}\Big)\epsilon L \log_2 n\Big)\log\Big(\frac{1}{\delta}\Big).$$

*(e.) Assume that for all $\mathbf{x}$, the function $\mathbf{y} \mapsto F(\mathbf{x}, \mathbf{y})$ is $\mu$-strongly-concave. If the algorithm $A$ is $\epsilon$-argument stable and Assumptions 1 and 2 hold, then for any $\delta > 0$, with probability at least $1 - \delta$,*

$$R(A_{\mathbf{x}}(S)) \leq (1 + \eta)\inf_{\mathbf{x} \in \mathcal{X}} R(\mathbf{x})$$
$$+ C\frac{2+\eta}{\eta}\Big(\frac{M}{n}\log\frac{1}{\delta} + \Big(\frac{\beta}{\mu} + 1\Big)L\epsilon \log_2 n \log\frac{1}{\delta} + \triangle^s_S(A_{\mathbf{x}}(S), A_{\mathbf{y}}(S))\Big).$$

**Remark 6.** To prove sharper high probability bounds than (Lei et al., 2021), the concentration inequality for a summation of weakly-dependent random variables proposed in (Bousquet et al., 2020) (Lemma 2 in Appendix A) plays a key role in our analysis. However, the direct use of this inequality will inevitably lead to a slow order bound since it contains a sampling error of slow order $\mathcal{O}(1/\sqrt{n})$. We exploit the proof techniques of the recent breakthrough work (Klochkov & Zhivotovskiy, 2021) to make new constructions of $g_i(S)$ so that the parameter $M$ in Lemma 2 is 0. However, the proof techniques of (Klochkov & Zhivotovskiy, 2021) can not be directly extended to minimax problems. The coupling construction between the minimization variables and the maximization variables makes the proofs of minimax problems more difficult than the minimization problem studied by (Klochkov & Zhivotovskiy, 2021). We must proceed with novel decompositions for the generalization measures. Note that different decompositions are required for different generalization measures. A pretty technical decomposition is exploited in the proof of Part (c).

Moreover, the proof of minimax problems needs refined analyses due to the minimax structure. For instance, in proving the primal generalization error, we need to quantify the fact that for different $A_{\mathbf{x}}(S)$, the optimal $\mathbf{y}$ is different in $R(A_{\mathbf{x}}(S))$. And the analysis of excess primal population risk in Part (e) is different from the excess risk analysis in (Klochkov & Zhivotovskiy, 2021). The reason is that the supremum operator in $\inf_{\mathbf{x} \in \mathcal{X}} R(\mathbf{x})$ makes the Bernstein condition used in (Klochkov & Zhivotovskiy, 2021) not applicable for excess primal population risk. Additionally, (Klochkov & Zhivotovskiy, 2021) only study ERM and GD, while we study more optimization algorithms: SGDA, PPM, EG, and OGDA.

**Remark 7.** According to Definition 1 and Jensen's inequality, we know that $\triangle_S^w(\mathbf{x}, \mathbf{y}) \leq \mathbb{E}[\triangle_S^s(\mathbf{x}, \mathbf{y})]$. Meanwhile, since we will provide the strong PD empirical risk bounds for several important optimization algorithms in Section 5, it thus implies that we also establish bounds with fast rates for $\triangle_S^w(\mathbf{x}, \mathbf{y})$.

To begin the proof of Theorem 1, we first introduce some key lemmas on concentration inequalities. The first lemma translates a moment bound into a high probability bound.

**Lemma 1.** *(Bousquet et al., 2020) Let $Z$ be a random variable with*

$$\|Z\|_p \leq \sqrt{p}a + pb$$

*for some $a, b > 0$ and for any $p \geq 2$. Then for any $\delta \in (0,1)$ we have, with probability at least $1 - \delta$,*

$$|Z| \leq e\Big(a\sqrt{\log\left(\frac{e}{\delta}\right)} + b\log\left(\frac{e}{\delta}\right)\Big),$$

*where $e$ is the base of the natural logarithm.*

The second lemma establishes a concentration inequality for a summation of weakly-dependent random variables.

**Lemma 2.** *(Bousquet et al., 2020) Let $S = \{z_1, ..., z_n\}$ be a set of independent random variables each taking values in $\mathcal{Z}$ and $M > 0$. Denote $[n]$ as the set $\{1, ..., n\}$. Define $S \backslash \{z_i\}$ be set $\{z_1, ..., z_{i-1}, z_{i+1}, ..., z_n\}$. Let $g_1, ..., g_n$ be some functions $g_i : \mathcal{Z}^n \mapsto \mathbb{R}$ such that the following inequalities hold for any $i \in [n]$,*

- $\big|\mathbb{E}_{S \backslash \{z_i\}}[g_i(S)]\big| \leq M$ *almost surely (a.s.),*

- $\mathbb{E}_{z_i}[g_i(S)] = 0$ *a.s.,*

- *for any $j \in [n]$ with $j \neq i$, and $z_j'' \in \mathcal{Z}$*

$$\big|g_i(S) - g_i(z_1, ..., z_{j-1}, z_j'', z_{j+1}, ..., z_n)\big| \leq \beta.$$

*Then, for any $p \geq 2$*

$$\Big\|\sum_{i=1}^n g_i(S)\Big\|_p \leq 12\sqrt{2}pn\beta\lceil\log_2 n\rceil + 4M\sqrt{pn}.$$

The following definition and lemma give the concentration inequality for non-negative weakly self-bounded functions.

**Definition 4.** *(Weakly Self-Bounded Function) Assume that $a, b > 0$. A function $f : \mathcal{Z}^n \mapsto [0, +\infty)$ is said to be $(a, b)$-weakly self-bounded if there exist functions $f_i : \mathcal{Z}^{n-1} \mapsto [0, +\infty)$ that satisfies for all $Z^n \in \mathcal{Z}^n$,*

$$\sum_{i=1}^n (f(Z^n) - f_i(Z^n))^2 \leq af(Z^n) + b.$$

**Lemma 3.** *(Klochkov & Zhivotovskiy, 2021) Suppose that $z_1, ..., z_n$ are independent random variables and the function $f : \mathcal{Z}^n \mapsto [0, +\infty)$ is $(a, b)$-weakly self-bounded and the corresponding function $f_i$ satisfy $f_i(Z^n) \geq f(Z^n)$ for $i = 1, ..., n$ and any $Z^n \in \mathcal{Z}^n$. Then, for any $t > 0$,*

$$Pr(\mathbb{E}f(z_1, ..., z_n) \geq f(z_1, ..., z_n) + t) \leq \exp\Big(-\frac{t^2}{2a\mathbb{E}f(z_1, ..., z_n) + 2b}\Big).$$

The following lemma is the classical Bernstein concentration inequality.

**Lemma 4.** *(Boucheron et al., 2013) Let $z_1, ..., z_n$ be i.i.d. random variables and assume that $\mathbb{E}[z_i] = \mu$. Suppose $|z_i| \leq c$ for any $i$. Then for any $\delta \in (0, 1)$, with probability at least $1 - \delta$,*

$$\left| \frac{1}{n} \sum_{i=1}^{n} z_i - \mu \right| \leq \sqrt{\frac{2\sigma^2 \log(1/\delta)}{n}} + \frac{2c \log(1/\delta)}{3n},$$

*where $\sigma^2$ is the variance of $z_i$.*

### A.1 PROOF OF PART (A)

We first prove the plain generalization error bound.

*Proof.* Let $S = \{z_1, ..., z_n\}$ be a set of independent random variables each taking values in $\mathcal{Z}$ and $S' = \{z'_1, ..., z'_n\}$ be its independent copy. For any $i \in [n]$, define $S^{(i)} = \{z_1, ..., z_{i-1}, z'_i, z_{i+1}, ..., z_n\}$ be a dataset by replacing the $i$-th sample in $S$ with another i.i.d. sample $z'_i$. We first have the following decomposition

$$nF(A_{\mathbf{x}}(S), A_{\mathbf{y}}(S)) - nF_S(A_{\mathbf{x}}(S), A_{\mathbf{y}}(S))$$

$$= \sum_{i=1}^{n} \mathbb{E}_Z[f(A_{\mathbf{x}}(S), A_{\mathbf{y}}(S); Z) - \mathbb{E}_{z'_i}[f(A_{\mathbf{x}}(S^{(i)}), A_{\mathbf{y}}(S^{(i)}); Z)]]$$

$$+ \sum_{i=1}^{n} \mathbb{E}_{z'_i}[\mathbb{E}_Z[f(A_{\mathbf{x}}(S^{(i)}), A_{\mathbf{y}}(S^{(i)}); Z)] - f(A_{\mathbf{x}}(S^{(i)}), A_{\mathbf{y}}(S^{(i)}); z_i)]$$

$$+ \sum_{i=1}^{n} \mathbb{E}_{z'_i}[f(A_{\mathbf{x}}(S^{(i)}), A_{\mathbf{y}}(S^{(i)}); z_i)] - \sum_{i=1}^{n} f(A_{\mathbf{x}}(S), A_{\mathbf{y}}(S); z_i).$$

According to the definition of uniform stability (Part 1 of Definition 3), we have

$$nF(A_{\mathbf{x}}(S), A_{\mathbf{y}}(S)) - nF_S(A_{\mathbf{x}}(S), A_{\mathbf{y}}(S)) \leq 2n\epsilon + \sum_{i=1}^{n} g_i(S),$$

where we have introduced $g_i(S) = \mathbb{E}_{z'_i}[\mathbb{E}_Z[f(A_{\mathbf{x}}(S^{(i)}), A_{\mathbf{y}}(S^{(i)}); Z)] - f(A_{\mathbf{x}}(S^{(i)}), A_{\mathbf{y}}(S^{(i)}); z_i)]$. Thus, by a rearrangement, we have

$$\left| nF(A_{\mathbf{x}}(S), A_{\mathbf{y}}(S)) - nF_S(A_{\mathbf{x}}(S), A_{\mathbf{y}}(S)) - \sum_{i=1}^{n} g_i(S) \right| \leq 2n\epsilon. \tag{3}$$

Then, for any $i = 1, ..., n$, we define $h_i(S) = g_i(S) - \mathbb{E}_{S \setminus \{z_i\}}[g_i(S)]$. It is easy to verify that $\mathbb{E}_{S \setminus \{z_i\}}[h_i(S)] = 0$ and $\mathbb{E}_{z_i}[h_i(S)] = \mathbb{E}_{z_i}[g_i(S)] - \mathbb{E}_{z_i} \mathbb{E}_{S \setminus \{z_i\}}[g_i(S)] = 0 - 0 = 0$. Also, for any $j \in [n]$ with $j \neq i$, and $z''_j \in \mathcal{Z}$, we have the following inequality

$$\left| h_i(S) - h_i(z_1, ..., z_{j-1}, z''_j, z_{j+1}, ..., z_n) \right| \leq \left| g_i(S) - g_i(z_1, ..., z_{j-1}, z''_j, z_{j+1}, ..., z_n) \right|$$
$$+ \left| \mathbb{E}_{S \setminus \{z_i\}}[g_i(S)] - \mathbb{E}_{S \setminus \{z_i\}}[g_i(z_1, ..., z_{j-1}, z''_j, z_{j+1}, ..., z_n)] \right|.$$

For the first term $|g_i(S) - g_i(z_1, ..., z_{j-1}, z''_j, z_{j+1}, ..., z_n)|$, it can be bounded by $2\epsilon$ according to the definition of uniform stability. Similar result holds for the second term $\left| \mathbb{E}_{S \setminus \{z_i\}}[g_i(S)] - \mathbb{E}_{S \setminus \{z_i\}}[g_i(z_1, ..., z_{j-1}, z''_j, z_{j+1}, ..., z_n)] \right|$ according to the definition of uniform stability. By a combination of the above analysis, we get $|h_i(S) - h_i(z_1, ..., z_{j-1}, z''_j, z_{j+1}, ..., z_n)| \leq 4\epsilon$.

We thus have verified that the three conditions in Lemma 2 are satisfied for $h_i(S)$. There will hold the following result for any $p \geq 2$

$$\left\| \sum_{i=1}^{n} h_i(S) \right\|_p \leq 48\sqrt{2}\epsilon pn \lceil \log_2 n \rceil. \tag{4}$$

Furthermore, we can derive that

$$nF(A_{\mathbf{x}}(S), A_{\mathbf{y}}(S)) - nF_S(A_{\mathbf{x}}(S), A_{\mathbf{y}}(S)) - \sum_{i=1}^{n} g_i(S) + \sum_{i=1}^{n} h_i(S)$$

$$=nF(A_{\mathbf{x}}(S), A_{\mathbf{y}}(S)) - nF_S(A_{\mathbf{x}}(S), A_{\mathbf{y}}(S)) - \sum_{i=1}^{n} \mathbb{E}_{S \setminus \{z_i\}}[g_i(S)]$$

$$=nF(A_{\mathbf{x}}(S), A_{\mathbf{y}}(S)) - nF_S(A_{\mathbf{x}}(S), A_{\mathbf{y}}(S)) - n\mathbb{E}_{S'}F(A_{\mathbf{x}}(S'), A_{\mathbf{y}}(S'))$$
$$+ n\mathbb{E}_{S'}F_S(A_{\mathbf{x}}(S'), A_{\mathbf{y}}(S'))$$

Due to the i.i.d. property between $S$ and $S'$, we know that $\mathbb{E}_{S'}F(A_{\mathbf{x}}(S'), A_{\mathbf{y}}(S')) = \mathbb{E}_S F(A_{\mathbf{x}}(S), A_{\mathbf{y}}(S))$.

Thus, combined (3) with (4), we have

$$\|nF(A_{\mathbf{x}}(S), A_{\mathbf{y}}(S)) - nF_S(A_{\mathbf{x}}(S), A_{\mathbf{y}}(S)) - n\mathbb{E}_S F(A_{\mathbf{x}}(S), A_{\mathbf{y}}(S)) + n\mathbb{E}_{S'} F_S(A_{\mathbf{x}}(S'), A_{\mathbf{y}}(S'))\|_p$$

$$\leq \left\| nF(A_{\mathbf{x}}(S), A_{\mathbf{y}}(S)) - nF_S(A_{\mathbf{x}}(S), A_{\mathbf{y}}(S)) - \sum_{i=1}^{n} g_i(S) \right\|_p$$

$$+ \left\| \sum_{i=1}^{n} g_i(S) - n\mathbb{E}_S F(A_{\mathbf{x}}(S), A_{\mathbf{y}}(S)) + n\mathbb{E}_{S'} F_S(A_{\mathbf{x}}(S'), A_{\mathbf{y}}(S')) \right\|_p$$

$$= \left\| nF(A_{\mathbf{x}}(S), A_{\mathbf{y}}(S)) - nF_S(A_{\mathbf{x}}(S), A_{\mathbf{y}}(S)) - \sum_{i=1}^{n} g_i(S) \right\|_p + \left\| \sum_{i=1}^{n} h_i(S) \right\|_p$$

$$\leq 2n\epsilon + 48\sqrt{2}\epsilon pn \lceil \log_2 n \rceil$$
$$\leq 50\sqrt{2}\epsilon pn \lceil \log_2 n \rceil.$$

According to Lemma 1, for any $\delta \in (0, 1)$, with probability at least $1 - \delta/3$, we have

$$F(A_{\mathbf{x}}(S), A_{\mathbf{y}}(S)) - F_S(A_{\mathbf{x}}(S), A_{\mathbf{y}}(S))$$
$$\leq |\mathbb{E}_{S'} F_S(A_{\mathbf{x}}(S'), A_{\mathbf{y}}(S')) - \mathbb{E}_S F(A_{\mathbf{x}}(S), A_{\mathbf{y}}(S))| + 50\sqrt{2}e\epsilon \lceil \log_2 n \rceil \log(3e/\delta). \qquad (5)$$

We now begin to bound the term $\mathbb{E}_{S'} F_S(A_{\mathbf{x}}(S'), A_{\mathbf{y}}(S')) - \mathbb{E}_S F(A_{\mathbf{x}}(S), A_{\mathbf{y}}(S))$. There holds that $\mathbb{E}_S \mathbb{E}_{S'} F_S(A_{\mathbf{x}}(S'), A_{\mathbf{y}}(S')) = \mathbb{E}_S F(A_{\mathbf{x}}(S), A_{\mathbf{y}}(S))$. We first consider the variance of $\mathbb{E}_{S'} f(A_{\mathbf{x}}(S'), A_{\mathbf{y}}(S'); z_i)$. By the Jensen's inequality, we have

$$\mathbb{E}_{z_i}[(\mathbb{E}_{S'} f(A_{\mathbf{x}}(S'), A_{\mathbf{y}}(S'); z_i))^2] \leq \mathbb{E}_{z_i} \mathbb{E}_{S'}[(f(A_{\mathbf{x}}(S'), A_{\mathbf{y}}(S'); z_i))^2]$$
$$= \mathbb{E}_Z \mathbb{E}_{S'}[(f(A_{\mathbf{x}}(S'), A_{\mathbf{y}}(S'); Z))^2]$$
$$= \mathbb{E}_Z \mathbb{E}_S[(f(A_{\mathbf{x}}(S), A_{\mathbf{y}}(S); Z))^2].$$

Then, by the Bernstein inequality in Lemma 4, we obtain the following inequality with probability at least $1 - \delta/3$,

$$|\mathbb{E}_{S'} F_S(A_{\mathbf{x}}(S'), A_{\mathbf{y}}(S')) - \mathbb{E}_S F(A_{\mathbf{x}}(S), A_{\mathbf{y}}(S))| \qquad (6)$$
$$\leq \sqrt{\frac{2\mathbb{E}_Z \mathbb{E}_S[(f(A_{\mathbf{x}}(S), A_{\mathbf{y}}(S); Z))^2] \log(3/\delta)}{n}} + \frac{2M \log(3/\delta)}{3n}.$$

Combined (5) with (6), we finally obtain that with probability at least $1 - 2\delta/3$,

$$F(A_{\mathbf{x}}(S), A_{\mathbf{y}}(S)) - F_S(A_{\mathbf{x}}(S), A_{\mathbf{y}}(S))$$
$$\leq \sqrt{\frac{2\mathbb{E}_Z \mathbb{E}_S[(f(A_{\mathbf{x}}(S), A_{\mathbf{y}}(S); Z))^2] \log(3/\delta)}{n}} + \frac{2M \log(3/\delta)}{3n} + 50\sqrt{2}e\epsilon \lceil \log_2 n \rceil \log(3e/\delta).$$
$$(7)$$

In the following, we define $q = q(z_1, ..., z_n) = \mathbb{E}_Z[(f(A_{\mathbf{x}}(S), A_{\mathbf{y}}(S); Z))^2]$ and $q_i = q_i(z_1, ..., z_n) = \sup_{z_i \in \mathcal{Z}} q(z_1, ..., z_n)$. So there holds $q_i \geq q$ for any $i = 1, .., n$ and any

$\{z_1, ..., z_n\} \in \mathcal{Z}^n$. Also, there holds that

$$
\sum_{i=1}^{n}(q - q_i)^2
$$

$$
= \sum_{i=1}^{n}\left(\mathbb{E}_Z[(f(A_{\mathbf{x}}(S), A_{\mathbf{y}}(S); Z))^2] - \sup_{z_i \in \mathcal{Z}}\mathbb{E}_Z[(f(A_{\mathbf{x}}(S), A_{\mathbf{y}}(S); Z))^2]\right)^2
$$

$$
\leq \sum_{i=1}^{n}\epsilon^2\left(\mathbb{E}_Z\left[f(A_{\mathbf{x}}(S), A_{\mathbf{y}}(S); Z) + \sup_{z_i \in \mathcal{Z}} f(A_{\mathbf{x}}(S), A_{\mathbf{y}}(S); Z)\right]\right)^2
$$

$$
\leq n\epsilon^2\left(2\mathbb{E}_Z[(f(A_{\mathbf{x}}(S), A_{\mathbf{y}}(S); Z))] + \epsilon\right)^2
$$

$$
\leq 8n\epsilon^2 q + 2n\epsilon^4, \tag{8}
$$

where the first inequality follows from the Jensen's inequality and the definition of uniform stability, and where the second inequality also follows from the definition of uniform stability.

From (8), we know that $q$ is $(8n\epsilon^2, 2n\epsilon^4)$ weakly self-bounded. Thus, by Lemma 3, we obtain that with probability at least $1 - \delta/3$,

$$
\mathbb{E}_S\mathbb{E}_Z[(f(A_{\mathbf{x}}(S), A_{\mathbf{y}}(S); Z))^2] - \mathbb{E}_Z[(f(A_{\mathbf{x}}(S), A_{\mathbf{y}}(S); Z))^2]
$$

$$
\leq \sqrt{(16n\epsilon^2\mathbb{E}_S\mathbb{E}_Z[(f(A_{\mathbf{x}}(S), A_{\mathbf{y}}(S); Z))^2] + 4n\epsilon^4)\log(3/\delta)}
$$

$$
= \sqrt{\left(\mathbb{E}_S\mathbb{E}_Z[(f(A_{\mathbf{x}}(S), A_{\mathbf{y}}(S); Z))^2] + \frac{1}{4}\epsilon^2\right)16n\epsilon^2\log(3/\delta)}
$$

$$
\leq \frac{1}{2}\left(\mathbb{E}_S\mathbb{E}_Z[(f(A_{\mathbf{x}}(S), A_{\mathbf{y}}(S); Z))^2] + \frac{1}{4}\epsilon^2\right) + 8n\epsilon^2\log(3/\delta),
$$

where the last inequality follows from that $\sqrt{ab} \leq \frac{a+b}{2}$ for all $a, b > 0$.

Since $\mathbb{E}_Z[(f(A_{\mathbf{x}}(S), A_{\mathbf{y}}(S); Z))^2] \leq MF(A_{\mathbf{x}}(S), A_{\mathbf{y}}(S))$, we have

$$
\mathbb{E}_S\mathbb{E}_Z[(f(A_{\mathbf{x}}(S), A_{\mathbf{y}}(S); Z))^2] - 2MF(A_{\mathbf{x}}(S), A_{\mathbf{y}}(S)) \leq \frac{1}{4}\epsilon^2 + 16n\epsilon^2\log(3/\delta). \tag{9}
$$

Substituting (9) into (7), we finally obtain that with probability at least $1 - \delta$,

$$
F(A_{\mathbf{x}}(S), A_{\mathbf{y}}(S)) - F_S(A_{\mathbf{x}}(S), A_{\mathbf{y}}(S))
$$

$$
\leq \sqrt{\frac{(4MF(A_{\mathbf{x}}(S), A_{\mathbf{y}}(S)) + \frac{1}{2}\epsilon^2 + 32n\epsilon^2\log(3/\delta))\log(3/\delta)}{n}}
$$

$$
+ \frac{2M\log(3/\delta)}{3n} + 50\sqrt{2}e\epsilon\lceil\log_2 n\rceil\log(3e/\delta). \tag{10}
$$

According to inequalities $\sqrt{ab} \leq \eta a + \frac{1}{\eta}b$ and $\sqrt{a+b} \leq \sqrt{a} + \sqrt{b}$ for any $a, b, \eta > 0$, we have the following inequality with probability at least $1 - \delta$

$$
F(A_{\mathbf{x}}(S), A_{\mathbf{y}}(S)) - F_S(A_{\mathbf{x}}(S), A_{\mathbf{y}}(S))
$$

$$
\leq \sqrt{\frac{(\frac{1}{2}\epsilon^2 + 32n\epsilon^2\log(3/\delta))\log(3/\delta)}{n}} + \frac{\eta}{1+\eta}F(A_{\mathbf{x}}(S), A_{\mathbf{y}}(S)) + \frac{1+\eta}{\eta}\frac{4M\log(3/\delta)}{n}
$$

$$
+ \frac{2M\log(3/\delta)}{3n} + 50\sqrt{2}e\epsilon\lceil\log_2 n\rceil\log(3e/\delta),
$$

which implies that

$$
F(A_{\mathbf{x}}(S), A_{\mathbf{y}}(S)) - (1+\eta)F_S(A_{\mathbf{x}}(S), A_{\mathbf{y}}(S))
$$

$$
\leq (1+\eta)\left(\sqrt{\frac{(\frac{1}{2}\epsilon^2 + 32n\epsilon^2\log(3/\delta))\log(3/\delta)}{n}} + \frac{1+\eta}{\eta}\frac{4M\log(3/\delta)}{n}\right.
$$

$$
\left. + \frac{2M\log(3/\delta)}{3n} + 50\sqrt{2}e\epsilon\lceil\log_2 n\rceil\log(3e/\delta)\right)
$$

$$
\leq C\frac{1+\eta}{\eta}\left(\frac{M}{n}\log(1/\delta) + \epsilon\log_2 n\log(1/\delta)\right),
$$

where $C$ is an absolute constant. The proof is complete. $\qquad\square$

## A.2 PROOF OF PART (B)

We then prove the primal generalization error bound. Before presenting the proof, we first introduce a lemma that quantifies the sensitivity of the optimal $\mathbf{y}$ and $\mathbf{x}$ w.r.t the perturbation of $\mathbf{x}$ and $\mathbf{y}$ respectively.

**Lemma 5.** *(Zhang et al., 2021a) Let $f : \mathcal{X} \times \mathcal{Y} \mapsto \mathbb{R}$. Assume that $f$ is $\mu$-strongly-convex-strongly-concave. Suppose that for any $\mathbf{x}, \mathbf{x}' \in \mathcal{X}$ and $\mathbf{y}, \mathbf{y}' \in \mathcal{Y}$ we have*

$$\|\nabla_{\mathbf{y}} f(\mathbf{x}, \mathbf{y}) - \nabla_{\mathbf{y}} f(\mathbf{x}', \mathbf{y})\| \le \beta \|\mathbf{x} - \mathbf{x}'\| \quad and \quad \|\nabla_{\mathbf{x}} f(\mathbf{x}, \mathbf{y}) - \nabla_{\mathbf{x}} f(\mathbf{x}, \mathbf{y}')\| \le \beta \|\mathbf{y} - \mathbf{y}'\|.$$

*Define $\mathbf{x}^*(\mathbf{y}) = \arg\min_{\mathbf{x} \in \mathcal{X}} h(\mathbf{x}, \mathbf{y})$ and $\mathbf{y}^*(\mathbf{x}) = \arg\max_{\mathbf{y} \in \mathcal{Y}} h(\mathbf{x}, \mathbf{y})$ for any $\mathbf{y}$ and $\mathbf{x}$ respectively. Then, for any $\mathbf{x}, \mathbf{x}' \in \mathcal{X}$ and $\mathbf{y}, \mathbf{y}' \in \mathcal{Y}$ there holds that*

$$\|\mathbf{y}^*(\mathbf{x}) - \mathbf{y}^*(\mathbf{x}')\| \le \frac{\beta}{\mu} \|\mathbf{x} - \mathbf{x}'\| \quad and \quad \|\mathbf{x}^*(\mathbf{y}) - \mathbf{x}^*(\mathbf{y}')\| \le \frac{\beta}{\mu} \|\mathbf{y} - \mathbf{y}'\|.$$

The proof of Part (b) shares similar proof techniques with Part (a), but requires a novel decomposition and several important changes. For instance, Lemma 5 should be needed to quantify the fact that for different $A_{\mathbf{x}}(S)$, the optimal $\mathbf{y}$ is different in $R(A_{\mathbf{x}}(S))$.

*Proof.* Let $S = \{z_1, ..., z_n\}$ be a set of independent random variables each taking values in $\mathcal{Z}$ and $S' = \{z'_1, ..., z'_n\}$ be its independent copy. For any $i \in [n]$, define $S^{(i)} = \{z_1, ..., z_{i-1}, z'_i, z_{i+1}, ..., z_n\}$ be a dataset by replacing the $i$-th sample in $S$ with another i.i.d. sample $z'_i$. Denote $\mathbf{y}^*_S = \arg\max_{\mathbf{y} \in \mathcal{Y}} F(A_{\mathbf{x}}(S), \mathbf{y})$ and $\hat{\mathbf{y}}^*_S = \arg\max_{\mathbf{y} \in \mathcal{Y}} F_S(A_{\mathbf{x}}(S), \mathbf{y})$. We have the following decomposition

$$
\begin{aligned}
&nR(A_{\mathbf{x}}(S)) - nR_S(A_{\mathbf{x}}(S)) \\
=&nF(A_{\mathbf{x}}(S), \mathbf{y}^*_S) - nF_S(A_{\mathbf{x}}(S), \hat{\mathbf{y}}^*_S) \\
=&\sum_{i=1}^{n} \mathbb{E}_Z[f(A_{\mathbf{x}}(S), \mathbf{y}^*_S; Z) - \mathbb{E}_{z'_i}[f(A_{\mathbf{x}}(S^{(i)}), \mathbf{y}^*_{S^{(i)}}; Z)]] \\
&+ \sum_{i=1}^{n} \mathbb{E}_{z'_i}[\mathbb{E}_Z[f(A_{\mathbf{x}}(S^{(i)}), \mathbf{y}^*_{S^{(i)}}; Z)] - f(A_{\mathbf{x}}(S^{(i)}), \mathbf{y}^*_{S^{(i)}}; z_i)] \\
&+ \sum_{i=1}^{n} \mathbb{E}_{z'_i}[f(A_{\mathbf{x}}(S^{(i)}), \mathbf{y}^*_{S^{(i)}}; z_i)] - \sum_{i=1}^{n} f(A_{\mathbf{x}}(S), \hat{\mathbf{y}}^*_S; z_i). \quad (11)
\end{aligned}
$$

Firstly, we have

$$
\begin{aligned}
&f(A_{\mathbf{x}}(S), \mathbf{y}^*_S; Z) - f(A_{\mathbf{x}}(S^{(i)}), \mathbf{y}^*_{S^{(i)}}; Z) \\
=&f(A_{\mathbf{x}}(S), \mathbf{y}^*_S; Z) - f(A_{\mathbf{x}}(S), \mathbf{y}^*_{S^{(i)}}; Z) + f(A_{\mathbf{x}}(S), \mathbf{y}^*_{S^{(i)}}; Z) - f(A_{\mathbf{x}}(S^{(i)}), \mathbf{y}^*_{S^{(i)}}; Z) \\
\le& L\|\mathbf{y}^*_S - \mathbf{y}^*_{S^{(i)}}\| + L\|A_{\mathbf{x}}(S) - A_{\mathbf{x}}(S^{(i)})\| \\
\le& \left(1 + \frac{\beta}{\mu}\right) L\|A_{\mathbf{x}}(S) - A_{\mathbf{x}}(S^{(i)})\| \\
\le& \left(1 + \frac{\beta}{\mu}\right) L\epsilon, \quad (12)
\end{aligned}
$$

where the second inequality follows from Lemma 5 with the fact that $F$ is smooth and $\mu$-SC-SC.

Secondly, we have

$$\sum_{i=1}^{n} \mathbb{E}_{z_i'}[f(A_{\mathbf{x}}(S^{(i)}), \mathbf{y}_{S^{(i)}}^*; z_i)]$$

$$= \sum_{i=1}^{n} \mathbb{E}_{z_i'}[f(A_{\mathbf{x}}(S^{(i)}), \mathbf{y}_{S^{(i)}}^*; z_i) - f(A_{\mathbf{x}}(S), \mathbf{y}_S^*; z_i) + f(A_{\mathbf{x}}(S), \mathbf{y}_S^*; z_i)]$$

$$\leq \sum_{i=1}^{n} \mathbb{E}_{z_i'}\Big(1 + \frac{\beta}{\mu}\Big) L \|A_{\mathbf{x}}(S) - A_{\mathbf{x}}(S^{(i)})\| + \sum_{i=1}^{n} f(A_{\mathbf{x}}(S), \mathbf{y}_S^*; z_i)$$

$$\leq \Big(1 + \frac{\beta}{\mu}\Big) Ln\epsilon + \sum_{i=1}^{n} f(A_{\mathbf{x}}(S), \mathbf{y}_S^*; z_i),$$

where the first and the last inequalities follow from (12). Substituting the above two results into (11), we obtain that

$$nF(A_{\mathbf{x}}(S), \mathbf{y}_S^*) - nF_S(A_{\mathbf{x}}(S), \hat{\mathbf{y}}_S^*)$$

$$\leq 2\Big(1 + \frac{\beta}{\mu}\Big) Ln\epsilon + \sum_{i=1}^{n} f(A_{\mathbf{x}}(S), \mathbf{y}_S^*; z_i) + \sum_{i=1}^{n} g_i(S) - \sum_{i=1}^{n} f(A_{\mathbf{x}}(S), \hat{\mathbf{y}}_S^*; z_i)$$

$$\leq 2\Big(1 + \frac{\beta}{\mu}\Big) Ln\epsilon + \sum_{i=1}^{n} g_i(S),$$

where the last inequality follows from the facts that $\sum_{i=1}^{n} f(A_{\mathbf{x}}(S), \mathbf{y}_S^*; z_i) - \sum_{i=1}^{n} f(A_{\mathbf{x}}(S), \hat{\mathbf{y}}_S^*; z_i) \leq 0$ and that we have introduced $g_i(S) = \mathbb{E}_{z_i'}[\mathbb{E}_Z[f(A_{\mathbf{x}}(S^{(i)}), \mathbf{y}_{S^{(i)}}^*; Z)] - f(A_{\mathbf{x}}(S^{(i)}), \mathbf{y}_{S^{(i)}}^*; z_i)]$.

Now we get

$$nF(A_{\mathbf{x}}(S), \mathbf{y}_S^*) - nF_S(A_{\mathbf{x}}(S), \hat{\mathbf{y}}_S^*) - \sum_{i=1}^{n} g_i(S) \leq 2\Big(1 + \frac{\beta}{\mu}\Big) Ln\epsilon. \tag{13}$$

For any $i = 1, ..., n$, we define $h_i(S) = g_i(S) - \mathbb{E}_{S \setminus \{z_i\}}[g_i(S)]$.

We also get that $\mathbb{E}_{S \setminus \{z_i\}}[h_i(S)] = 0$ and $\mathbb{E}_{z_i}[h_i(S)] = \mathbb{E}_{z_i}[g_i(S)] - \mathbb{E}_{z_i}\mathbb{E}_{S \setminus \{z_i\}}[g_i(S)] = 0 - 0 = 0$. Moreover, for any $j \in [n]$ with $j \neq i$, and $z_j'' \in \mathcal{Z}$, we have the following inequality

$$|h_i(S) - h_i(z_1, ..., z_{j-1}, z_j'', z_{j+1}, ..., z_n)| \leq |g_i(S) - g_i(z_1, ..., z_{j-1}, z_j'', z_{j+1}, ..., z_n)|$$
$$+ |\mathbb{E}_{S \setminus \{z_i\}}[g_i(S)] - \mathbb{E}_{S \setminus \{z_i\}}[g_i(z_1, ..., z_{j-1}, z_j'', z_{j+1}, ..., z_n)]|.$$

Denote $S_j^{(i)}$ as the set collected by replacing the $j$-th element of $S^{(i)}$ with $z_j''$. For the first term $|g_i(S) - g_i(z_1, ..., z_{j-1}, z_j'', z_{j+1}, ..., z_n)|$, we have

$$\Big|g_i(S) - g_i(z_1, ..., z_{j-1}, z_j'', z_{j+1}, ..., z_n)\Big|$$

$$= \Big|\mathbb{E}_{z_i'}\Big[\mathbb{E}_Z[f(A_{\mathbf{x}}(S^{(i)}), \mathbf{y}_{S^{(i)}}^*; Z)] - f(A_{\mathbf{x}}(S^{(i)}), \mathbf{y}_{S^{(i)}}^*; z_i)\Big]$$

$$- \mathbb{E}_{z_i'}\Big[\mathbb{E}_Z[f(A_{\mathbf{x}}(S_j^{(i)}), \mathbf{y}_{S_j^{(i)}}^*; Z)] - f(A_{\mathbf{x}}(S_j^{(i)}), \mathbf{y}_{S_j^{(i)}}^*; z_i)\Big]\Big|$$

$$\leq \Big|\mathbb{E}_{z_i'}\Big[\mathbb{E}_Z[f(A_{\mathbf{x}}(S^{(i)}), \mathbf{y}_{S^{(i)}}^*; Z)] - f(A_{\mathbf{x}}(S_j^{(i)}), \mathbf{y}_{S_j^{(i)}}^*; Z)]\Big]\Big|$$

$$+ \Big|\mathbb{E}_{z_i'}\Big[f(A_{\mathbf{x}}(S^{(i)}), \mathbf{y}_{S^{(i)}}^*; z_i) - f(A_{\mathbf{x}}(S_j^{(i)}), \mathbf{y}_{S_j^{(i)}}^*; z_i)\Big]\Big|. \tag{14}$$

Furthermore, for any $z$, we have the following result which can help to bound the above inequality.

$$
\left| f(A_{\mathbf{x}}(S^{(i)}), \mathbf{y}^*_{S^{(i)}}; z) - f(A_{\mathbf{x}}(S_j^{(i)}), \mathbf{y}^*_{S_j^{(i)}}; z) \right|
$$

$$
\leq \left| f(A_{\mathbf{x}}(S^{(i)}), \mathbf{y}^*_{S^{(i)}}; z) - f(A_{\mathbf{x}}(S^{(i)}), \mathbf{y}^*_{S_j^{(i)}}; z) \right| + \left| f(A_{\mathbf{x}}(S^{(i)}), \mathbf{y}^*_{S_j^{(i)}}; z) - f(A_{\mathbf{x}}(S_j^{(i)}), \mathbf{y}^*_{S_j^{(i)}}; z) \right|
$$

$$
\leq L \left\| \mathbf{y}^*_{S^{(i)}} - \mathbf{y}^*_{S_j^{(i)}} \right\| + L \left\| A_{\mathbf{x}}(S^{(i)}) - A_{\mathbf{x}}(S_j^{(i)}) \right\|
$$

$$
\leq \left( \frac{\beta}{\mu} + 1 \right) L \left\| A_{\mathbf{x}}(S^{(i)}) - A_{\mathbf{x}}(S_j^{(i)}) \right\|
$$

$$
\leq \left( \frac{\beta}{\mu} + 1 \right) L \epsilon, \tag{15}
$$

where the third inequality follows from Lemma 5. Thus, we can bound the first term by $2 \left( \frac{\beta}{\mu} + 1 \right) L \epsilon$. By a similar analysis, the second term $|\mathbb{E}_{S \setminus \{z_i\}}[g_i(S)] - \mathbb{E}_{S \setminus \{z_i\}}[g_i(z_1, ..., z_{j-1}, z_j'', z_{j+1}, ..., z_n)]|$ can also be bounded by $2 \left( \frac{\beta}{\mu} + 1 \right) L \epsilon$.

We now have verified that the three conditions in Lemma 2 are satisfied for $h_i(S)$. We obtain that for any $p \geq 2$, there holds

$$
\left\| \sum_{i=1}^{n} h_i(S) \right\|_p \leq 48\sqrt{2} p n \left( \frac{\beta}{\mu} + 1 \right) L \epsilon \lceil \log_2 n \rceil. \tag{16}
$$

Thus, combined (13) with (16), we derive that

$$
\left\| n F(A_{\mathbf{x}}(S), \mathbf{y}^*_S) - n F_S(A_{\mathbf{x}}(S), \hat{\mathbf{y}}^*_S) - n \mathbb{E}_{S'} F(A_{\mathbf{x}}(S'), \mathbf{y}^*_{S'}) + n \mathbb{E}_{S'} F_S(A_{\mathbf{x}}(S'), \mathbf{y}^*_{S'}) \right\|_p
$$

$$
= \left\| n F(A_{\mathbf{x}}(S), \mathbf{y}^*_S) - \sum_{i=1}^{n} g_i(S) + \sum_{i=1}^{n} g_i(S) - \sum_{i=1}^{n} \mathbb{E}_{S \setminus \{z_i\}}[g_i(S)] \right\|_p
$$

$$
\leq \left\| n F(A_{\mathbf{x}}(S), \mathbf{y}^*_S) - n F_S(A_{\mathbf{x}}(S), \hat{\mathbf{y}}^*_S) - \sum_{i=1}^{n} g_i(S) \right\|_p + \left\| \sum_{i=1}^{n} h_i(S) \right\|_p
$$

$$
\leq 2 \left( 1 + \frac{\beta}{\mu} \right) L n \epsilon + 48\sqrt{2} p n \left( \frac{\beta}{\mu} + 1 \right) L \epsilon \lceil \log_2 n \rceil
$$

$$
\leq 50\sqrt{2} \left( 1 + \frac{\beta}{\mu} \right) \epsilon L p n \lceil \log_2 n \rceil. \tag{17}
$$

Then, according to Lemma 1, for any $\delta \in (0, 1)$, with probability at least $1 - \delta/3$, we have

$$
F(A_{\mathbf{x}}(S), \mathbf{y}^*_S) - F_S(A_{\mathbf{x}}(S), \hat{\mathbf{y}}^*_S)
$$

$$
\leq |\mathbb{E}_{S'} F(A_{\mathbf{x}}(S'), \mathbf{y}^*_{S'}) - \mathbb{E}_{S'} F_S(A_{\mathbf{x}}(S'), \mathbf{y}^*_{S'})| + 50\sqrt{2} \epsilon e L \left( 1 + \frac{\beta}{\mu} \right) \lceil \log_2 n \rceil \log(3e/\delta). \tag{18}
$$

With a similar analysis to the proof of Part (a), we now begin to bound the variance of $\mathbb{E}_{S'} f(A_{\mathbf{x}}(S'), \mathbf{y}^*_{S'}; z_i)$.

$$
\mathbb{E}_{z_i} [(\mathbb{E}_{S'} f(A_{\mathbf{x}}(S'), \mathbf{y}^*_{S'}; z_i))^2] \leq \mathbb{E}_{z_i} \mathbb{E}_{S'} [(f(A_{\mathbf{x}}(S'), \mathbf{y}^*_{S'}; z_i))^2]
$$

$$
= \mathbb{E}_Z \mathbb{E}_{S'} [(f(A_{\mathbf{x}}(S'), \mathbf{y}^*_{S'}; Z))^2]
$$

$$
= \mathbb{E}_Z \mathbb{E}_S [(f(A_{\mathbf{x}}(S), \mathbf{y}^*_S; Z))^2].
$$

There holds that $\mathbb{E}_S \mathbb{E}_{S'} F_S(A_{\mathbf{x}}(S'), \mathbf{y}^*_{S'}) = \mathbb{E}_{S'} F(A_{\mathbf{x}}(S'), \mathbf{y}^*_{S'})$. Then, by the Bernstein inequality in Lemma 4, we obtain that with probability at least $1 - \delta/3$,

$$
|\mathbb{E}_{S'} F(A_{\mathbf{x}}(S'), \mathbf{y}^*_{S'}) - \mathbb{E}_{S'} F_S(A_{\mathbf{x}}(S'), \mathbf{y}^*_{S'})|
$$

$$
\leq \sqrt{\frac{2 \mathbb{E}_Z \mathbb{E}_S [(f(A_{\mathbf{x}}(S), \mathbf{y}^*_S; Z))^2] \log(3/\delta)}{n}} + \frac{2M \log(3/\delta)}{3n}. \tag{19}
$$

Combined (18) with (19), we finally obtain that with probability at least $1 - 2\delta/3$,

$$F(A_{\mathbf{x}}(S), \mathbf{y}_S^*) - F_S(A_{\mathbf{x}}(S), \hat{\mathbf{y}}_S^*)$$

$$\leq \sqrt{\frac{2\mathbb{E}_Z\mathbb{E}_S[(f(A_{\mathbf{x}}(S), \mathbf{y}_S^*; Z))^2]\log\left(\frac{3}{\delta}\right)}{n}} + \frac{2M\log\left(\frac{3}{\delta}\right)}{3n} + 50\sqrt{2}\epsilon eL\frac{\beta + \mu}{\mu}\lceil \log_2 n \rceil \log\left(\frac{3e}{\delta}\right). \tag{20}$$

In the following, we define $q = q(z_1, ..., z_n) = \mathbb{E}_Z[(f(A_{\mathbf{x}}(S), \mathbf{y}_S^*; Z))^2]$ and $q_i = q_i(z_1, ..., z_n) = \sup_{z_i \in \mathcal{Z}} q(z_1, ..., z_n)$. So there holds $q_i \geq q$ for any $i = 1, .., n$ and any $\{z_1, ..., z_n\} \in \mathcal{Z}^n$. Also, there holds that

$$\sum_{i=1}^n (q - q_i)^2$$

$$= \sum_{i=1}^n \left( \mathbb{E}_Z[(f(A_{\mathbf{x}}(S), \mathbf{y}_S^*; Z))^2] - \sup_{z_i \in \mathcal{Z}} \mathbb{E}_Z[(f(A_{\mathbf{x}}(S), \mathbf{y}_S^*; Z))^2] \right)^2$$

$$\leq \sum_{i=1}^n \left( \mathbb{E}_Z\left[ \sup_{z_i \in \mathcal{Z}} (f(A_{\mathbf{x}}(S), \mathbf{y}_S^*; Z))^2 - (f(A_{\mathbf{x}}(S), \mathbf{y}_S^*; Z))^2 \right] \right)^2$$

$$\leq n\left(\frac{\beta}{\mu} + 1\right)^2 L^2\epsilon^2 \left( 2\mathbb{E}_Z[f(A_{\mathbf{x}}(S), \mathbf{y}_S^*; Z)] + \left(\frac{\beta}{\mu} + 1\right)L\epsilon \right)^2$$

$$\leq 8n\left(\frac{\beta}{\mu} + 1\right)^2 L^2\epsilon^2 q + 2n\left(\frac{\beta}{\mu} + 1\right)^4 L^4\epsilon^4,$$

where the first inequality follows from the Jensen's inequality and the second inequality follows from a similar analysis to (12) or (15). Now, we know that $q$ is $\left(8n\left(\frac{\beta}{\mu} + 1\right)^2 L^2\epsilon^2, 2n\left(\frac{\beta}{\mu} + 1\right)^4 L^4\epsilon^4\right)$-weakly self-bounded. Thus, by Lemma 3, we obtain the following inequality with probability at least $1 - \delta/3$,

$$\mathbb{E}_S\mathbb{E}_Z[(f(A_{\mathbf{x}}(S), \mathbf{y}_S^*; Z))^2] - \mathbb{E}_Z[(f(A_{\mathbf{x}}(S), \mathbf{y}_S^*; Z))^2]$$

$$\leq \sqrt{\left( 16n\left(\frac{\beta}{\mu} + 1\right)^2 L^2\epsilon^2\mathbb{E}_S\mathbb{E}_Z[(f(A_{\mathbf{x}}(S), \mathbf{y}_S^*; Z))^2] + 4n\left(\frac{\beta}{\mu} + 1\right)^4 L^4\epsilon^4 \right)\log(3/\delta)}$$

$$= \sqrt{\left( \mathbb{E}_S\mathbb{E}_Z[(f(A_{\mathbf{x}}(S), \mathbf{y}_S^*; Z))^2] + \frac{1}{4}\left(\frac{\beta}{\mu} + 1\right)^2 L^2\epsilon^2 \right)16n\left(\frac{\beta}{\mu} + 1\right)^2 L^2\epsilon^2\log(3/\delta)}$$

$$\leq \frac{1}{2}\left( \mathbb{E}_S\mathbb{E}_Z[(f(A_{\mathbf{x}}(S), \mathbf{y}_S^*; Z))^2] + \frac{1}{4}\left(\frac{\beta}{\mu} + 1\right)^2 L^2\epsilon^2 \right) + 8n\left(\frac{\beta}{\mu} + 1\right)^2 L^2\epsilon^2\log(3/\delta),$$

where the last inequality follows from $\sqrt{ab} \leq \frac{a+b}{2}$ for all $a, b > 0$.

Since $\mathbb{E}_Z[(f(A_{\mathbf{x}}(S), \mathbf{y}_S^*; Z))^2] \leq MF(A_{\mathbf{x}}(S), \mathbf{y}_S^*)$, we have

$$\mathbb{E}_S\mathbb{E}_Z[(f(A_{\mathbf{x}}(S), \mathbf{y}_S^*; Z))^2] - 2MF(A_{\mathbf{x}}(S), \mathbf{y}_S^*)$$

$$\leq \frac{1}{4}\left(\frac{\beta}{\mu} + 1\right)^2 L^2\epsilon^2 + 16n\left(\frac{\beta}{\mu} + 1\right)^2 L^2\epsilon^2\log(3/\delta). \tag{21}$$

Plugging (21) into (20), we finally obtain that with probability at least $1 - \delta$,

$$F(A_{\mathbf{x}}(S), \mathbf{y}_S^*) - F_S(A_{\mathbf{x}}(S), \hat{\mathbf{y}}_S^*)$$

$$\leq \sqrt{\frac{\left( 4MF(A_{\mathbf{x}}(S), \mathbf{y}_S^*) + \frac{1}{2}\left(\frac{\beta}{\mu} + 1\right)^2 L^2\epsilon^2 + 32n\left(\frac{\beta}{\mu} + 1\right)^2 L^2\epsilon^2\log(3/\delta) \right)\log(3/\delta)}{n}}$$

$$+ \frac{2M\log(3/\delta)}{3n} + 50\sqrt{2}\epsilon eL\frac{\beta + \mu}{\mu}\lceil \log_2 n \rceil \log(3e/\delta). \tag{22}$$

By the elementary inequalities $\sqrt{ab} \leq \eta a + \frac{1}{\eta} b$ and $\sqrt{a+b} \leq \sqrt{a} + \sqrt{b}$ for any $a, b, \eta > 0$, we have the following inequality with probability at least $1 - \delta$

$$F(A_{\mathbf{x}}(S), \mathbf{y}_S^*) - (1+\eta)F_S(A_{\mathbf{x}}(S), \hat{\mathbf{y}}_S^*) \leq C\frac{1+\eta}{\eta}\left(\frac{M}{n}\log\frac{1}{\delta} + \left(\frac{\beta}{\mu}+1\right)L\epsilon\log_2 n \log\frac{1}{\delta}\right),$$

where $C$ is an absolute constant. The proof is complete. $\square$

### A.3 PROOF OF PART (C)

We then prove the strong PD population risk bound. To begin, we introduce the following concentration inequality, which is a moment version of the Bernstein inequality.

**Lemma 6.** *(Boucheron et al., 2013) If $z_1, ..., z_n$ are i.i.d., zero mean and $|z_i| \leq M$ almost surely. Then, for any $p \geq 2$,*

$$\left\|\sum_{i=1}^n z_i\right\|_p \leq 6\sqrt{\left(\sum_{i=1}^n \mathbb{E}z_i^2\right)p} + 4pM.$$

*Proof.* Let $S = \{z_1, ..., z_n\}$ be a set of independent random variables each taking values in $\mathcal{Z}$ and $S' = \{z_1', ..., z_n'\}$ be its independent copy. For any $i \in [n]$, define $S^{(i)} = \{z_1, ..., z_{i-1}, z_i', z_{i+1}, ..., z_n\}$ be a dataset by replacing the $i$-th sample in $S$ with another i.i.d. sample $z_i'$. Denote $\mathbf{y}_S^* = \arg\max_{\mathbf{y}\in\mathcal{Y}} F(A_{\mathbf{x}}(S), \mathbf{y})$, $\hat{\mathbf{y}}_S^* = \arg\max_{\mathbf{y}\in\mathcal{Y}} F_S(A_{\mathbf{x}}(S), \mathbf{y})$, $\mathbf{x}_S^* = \arg\min_{\mathbf{x}\in\mathcal{X}} F(\mathbf{x}, A_{\mathbf{y}}(S))$ and $\hat{\mathbf{x}}_S^* = \arg\min_{\mathbf{x}\in\mathcal{X}} F_S(\mathbf{x}, A_{\mathbf{y}}(S))$.

The proof of Part (c) requires a pretty technical error decomposition, i.e.,

$$\triangle^s (A_{\mathbf{x}}(S), A_{\mathbf{y}}(S))$$
$$= F(A_{\mathbf{x}}(S), \mathbf{y}_S^*) - \inf_{\mathbf{x}'\in\mathcal{X}} F(\mathbf{x}', A_{\mathbf{y}}(S))$$
$$= F(A_{\mathbf{x}}(S), \mathbf{y}_S^*) - F_S(A_{\mathbf{x}}(S), \hat{\mathbf{y}}_S^*) + \mathbb{E}_{S'}F_S(A_{\mathbf{x}}(S'), \mathbf{y}_{S'}^*) - \mathbb{E}_S F(A_{\mathbf{x}}(S), \mathbf{y}_S^*)$$
$$+ \mathbb{E}_S F(\mathbf{x}_S^*, A_{\mathbf{y}}(S)) - \mathbb{E}_{S'}F_S(\mathbf{x}_{S'}^*, A_{\mathbf{y}}(S')) + F_S(\hat{\mathbf{x}}_S^*, A_{\mathbf{y}}(S)) - \inf_{\mathbf{x}'\in\mathcal{X}} F(\mathbf{x}', A_{\mathbf{y}}(S))$$
$$- \mathbb{E}_{S'}F_S(A_{\mathbf{x}}(S'), \mathbf{y}_{S'}^*) + \mathbb{E}_S F(A_{\mathbf{x}}(S), \mathbf{y}_S^*) + \mathbb{E}_{S'}F_S(\mathbf{x}_{S'}^*, A_{\mathbf{y}}(S')) - \mathbb{E}_S F(\mathbf{x}_S^*, A_{\mathbf{y}}(S))$$
$$+ F_S(A_{\mathbf{x}}(S), \hat{\mathbf{y}}_S^*) - F_S(\hat{\mathbf{x}}_S^*, A_{\mathbf{y}}(S)).$$

Let's first consider the term $F(A_{\mathbf{x}}(S), \mathbf{y}_S^*) - F_S(A_{\mathbf{x}}(S), \hat{\mathbf{y}}_S^*) + \mathbb{E}_{S'}F_S(A_{\mathbf{x}}(S'), \mathbf{y}_{S'}^*) - \mathbb{E}_S F(A_{\mathbf{x}}(S), \mathbf{y}_S^*)$. It can be then decomposed into

$$F(A_{\mathbf{x}}(S), \mathbf{y}_S^*) - F_S(A_{\mathbf{x}}(S), \hat{\mathbf{y}}_S^*) + \frac{1}{n}\sum_{i=1}^n \mathbb{E}_{z_i'}\left[\mathbb{E}_Z[f(A_{\mathbf{x}}(S^{(i)}), \mathbf{y}_{S^{(i)}}^*; Z)] - f(A_{\mathbf{x}}(S^{(i)}), \mathbf{y}_{S^{(i)}}^*; z_i)\right]$$

$$- \frac{1}{n}\sum_{i=1}^n \mathbb{E}_{z_i'}\left[\mathbb{E}_Z[f(A_{\mathbf{x}}(S^{(i)}), \mathbf{y}_{S^{(i)}}^*; Z)] - f(A_{\mathbf{x}}(S^{(i)}), \mathbf{y}_{S^{(i)}}^*; z_i)\right] + \mathbb{E}_{S'}F_S(A_{\mathbf{x}}(S'), \mathbf{y}_{S'}^*)$$

$$- \mathbb{E}_S F(A_{\mathbf{x}}(S), \mathbf{y}_S^*),$$

which can be bounded by the proof techniques in the proof of Part (b). According to (17), we know that

$$\|F(A_{\mathbf{x}}(S), \mathbf{y}_S^*) - F_S(A_{\mathbf{x}}(S), \hat{\mathbf{y}}_S^*) + \mathbb{E}_{S'}F_S(A_{\mathbf{x}}(S'), \mathbf{y}_{S'}^*) - \mathbb{E}_S F(A_{\mathbf{x}}(S), \mathbf{y}_S^*)\|_p$$
$$\leq 50\sqrt{2}\left(1+\frac{\beta}{\mu}\right)\epsilon Lp\lceil\log_2 n\rceil. \tag{23}$$

For the second term $\mathbb{E}_S F(\mathbf{x}_S^*, A_{\mathbf{y}}(S)) - \mathbb{E}_{S'}F_S(\mathbf{x}_{S'}^*, A_{\mathbf{y}}(S')) + F_S(\hat{\mathbf{x}}_S^*, A_{\mathbf{y}}(S)) - \inf_{\mathbf{x}'\in\mathcal{X}} F(\mathbf{x}', A_{\mathbf{y}}(S))$, we have the following decomposition

$$nF_S(\hat{\mathbf{x}}_S^*, A_{\mathbf{y}}(S)) - n\inf_{\mathbf{x}'\in\mathcal{X}} F(\mathbf{x}', A_{\mathbf{y}}(S))$$

$$= nF_S(\hat{\mathbf{x}}_S^*, A_{\mathbf{y}}(S)) - \sum_{i=1}^n \mathbb{E}_Z\left[f(\mathbf{x}_S^*, A_{\mathbf{y}}(S), Z) - \mathbb{E}_{z_i'}[f(\mathbf{x}_{S^{(i)}}^*, A_{\mathbf{y}}(S^{(i)}); Z)]\right]$$

$$+ \sum_{i=1}^n \mathbb{E}_{z_i'}\left[f(\mathbf{x}_{S^{(i)}}^*, A_{\mathbf{y}}(S^{(i)}); z_i) - \mathbb{E}_Z[f(\mathbf{x}_{S^{(i)}}^*, A_{\mathbf{y}}(S^{(i)}); Z)]\right] - \sum_{i=1}^n \mathbb{E}_{z_i'}\left[f(\mathbf{x}_{S^{(i)}}^*, A_{\mathbf{y}}(S^{(i)}); z_i)\right].$$

It is clear that

$$\sum_{i=1}^{n} \mathbb{E}_{z_i'} \left[ f(\mathbf{x}_{S^{(i)}}^*, A_{\mathbf{y}}(S^{(i)}); z_i) \right]$$

$$= \sum_{i=1}^{n} \mathbb{E}_{z_i'} \left[ f(\mathbf{x}_{S^{(i)}}^*, A_{\mathbf{y}}(S^{(i)}); z_i) - f(\mathbf{x}_S^*, A_{\mathbf{y}}(S); z_i) + f(\mathbf{x}_S^*, A_{\mathbf{y}}(S); z_i) \right]$$

$$\geq nF_S(\hat{\mathbf{x}}_S^*, A_{\mathbf{y}}(S)) + \sum_{i=1}^{n} \mathbb{E}_{z_i'} \left[ f(\mathbf{x}_{S^{(i)}}^*, A_{\mathbf{y}}(S^{(i)}); z_i) - f(\mathbf{x}_S^*, A_{\mathbf{y}}(S); z_i) \right]. \tag{24}$$

Denote $g_i(S) = \mathbb{E}_{z_i'} \left[ f(\mathbf{x}_{S^{(i)}}^*, A_{\mathbf{y}}(S^{(i)}); z_i) - \mathbb{E}_Z[f(\mathbf{x}_{S^{(i)}}^*, A_{\mathbf{y}}(S^{(i)}); Z)] \right]$. By (24), we now get

$$nF_S(\hat{\mathbf{x}}_S^*, A_{\mathbf{y}}(S)) - n \inf_{\mathbf{x}' \in \mathcal{X}} F(\mathbf{x}', A_{\mathbf{y}}(S)) - \sum_{i=1}^{n} g_i(S)$$

$$\leq -\sum_{i=1}^{n} \mathbb{E}_Z \left[ f(\mathbf{x}_S^*, A_{\mathbf{y}}(S); Z) - \mathbb{E}_{z_i'}[f(\mathbf{x}_{S^{(i)}}^*, A_{\mathbf{y}}(S^{(i)}); Z)] \right]$$

$$- \sum_{i=1}^{n} \mathbb{E}_{z_i'} \left[ f(\mathbf{x}_{S^{(i)}}^*, A_{\mathbf{y}}(S^{(i)}); z_i) - f(\mathbf{x}_S^*, A_{\mathbf{y}}(S); z_i) \right].$$

Furthermore, according to Lemma 5, we have

$$\left| f(\mathbf{x}_S^*, A_{\mathbf{y}}(S); Z) - f(\mathbf{x}_{S^{(i)}}^*, A_{\mathbf{y}}(S^{(i)}); Z) \right| \leq \left(1 + \frac{\beta}{\mu}\right) L \|A_{\mathbf{y}}(S) - A_{\mathbf{y}}(S^{(i)})\|.$$

Similarly,

$$\left| f(\mathbf{x}_{S^{(i)}}^*, A_{\mathbf{y}}(S^{(i)}); z_i) - f(\mathbf{x}_S^*, A_{\mathbf{y}}(S); z_i) \right| \leq \left(1 + \frac{\beta}{\mu}\right) L \|A_{\mathbf{y}}(S) - A_{\mathbf{y}}(S^{(i)})\|.$$

Thus, we obtain

$$\left| nF_S(\hat{\mathbf{x}}_S^*, A_{\mathbf{y}}(S)) - n \inf_{\mathbf{x}' \in \mathcal{X}} F(\mathbf{x}', A_{\mathbf{y}}(S)) - \sum_{i=1}^{n} g_i(S) \right|$$

$$\leq \sum_{i=1}^{n} 2\left(1 + \frac{\beta}{\mu}\right) L \|A_{\mathbf{y}}(S^{(i)}) - A_{\mathbf{y}}(S)\| \leq 2n\left(1 + \frac{\beta}{\mu}\right) L\epsilon, \tag{25}$$

where the last inequality follows from the definition of argument stability (Part 2 of Definition 3).

Furthermore, we define $h_i(S) = g_i(S) - \mathbb{E}_{S \setminus \{z_i\}}[g_i(S)]$. For $h_i(S)$, We have $\mathbb{E}_{S \setminus \{z_i\}}[h_i(S)] = 0$ and $\mathbb{E}_{z_i}[h_i(S)] = \mathbb{E}_{z_i}[g_i(S)] - \mathbb{E}_{z_i}\mathbb{E}_{S \setminus \{z_i\}}[g_i(S)] = 0 - 0 = 0$. Moreover, for any $j \in [n]$ with $j \neq i$, and $z_j'' \in \mathcal{Z}$, we get

$$|h_i(S) - h_i(z_1, ..., z_{j-1}, z_j'', z_{j+1}, ..., z_n)| \leq 2\left(1 + \frac{\beta}{\mu}\right) L\epsilon,$$

where this inequality follows from a similar analysis to (14) and (15) of the proof of Part (b). We thus can obtain that for any $p \geq 2$, there holds

$$\left\| \sum_{i=1}^{n} h_i(S) \right\|_p \leq 48\sqrt{2}pn\left(\frac{\beta}{\mu} + 1\right) L\epsilon \lceil \log_2 n \rceil. \tag{26}$$

Combined (25) with (26), we finally get the bound of the second term:

$$\left\| \mathbb{E}_S F(\mathbf{x}_S^*, A_{\mathbf{y}}(S)) - \mathbb{E}_{S'} F_S(\mathbf{x}_{S'}^*, A_{\mathbf{y}}(S')) + F_S(\hat{\mathbf{x}}_S^*, A_{\mathbf{y}}(S)) - \inf_{\mathbf{x}' \in \mathcal{X}} F(\mathbf{x}', A_{\mathbf{y}}(S)) \right\|_p$$

$$\leq \left\| F_S(\hat{\mathbf{x}}_S^*, A_{\mathbf{y}}(S)) - \inf_{\mathbf{x}' \in \mathcal{X}} F(\mathbf{x}', A_{\mathbf{y}}(S)) - \frac{1}{n} \sum_{i=1}^{n} g_i(S) \right\|_p + \left\| \frac{1}{n} \sum_{i=1}^{n} h_i(S) \right\|_p$$

$$\leq 50\sqrt{2}\left(1 + \frac{\beta}{\mu}\right) \epsilon Lp \lceil \log_2 n \rceil. \tag{27}$$

We then consider the third term $-\mathbb{E}_{S'}F_S(A_{\mathbf{x}}(S'), \mathbf{y}_{S'}^*) + \mathbb{E}_S F(A_{\mathbf{x}}(S), \mathbf{y}_S^*) + \mathbb{E}_{S'}F_S(\mathbf{x}_{S'}^*, A_{\mathbf{y}}(S')) - \mathbb{E}_S F(\mathbf{x}_S^*, A_{\mathbf{y}}(S))$. It is clear that $\mathbb{E}_S[\mathbb{E}_{S'}F_S(\mathbf{x}_{S'}^*, A_{\mathbf{y}}(S'))] = \mathbb{E}_S F(\mathbf{x}_S^*, A_{\mathbf{y}}(S))$ and $\mathbb{E}_S[\mathbb{E}_{S'}F_S(A_{\mathbf{x}}(S'), \mathbf{y}_{S'}^*)] = \mathbb{E}_S F(A_{\mathbf{x}}(S), \mathbf{y}_S^*)$.

Moreover, we having the following important property due to the strong convexity and strong concavity of $F$,

$$
\mathbb{E}\left[(\mathbb{E}_{S'}f(\mathbf{x}_{S'}^*, A_{\mathbf{y}}(S'); z_i) - \mathbb{E}_{S'}f(A_{\mathbf{x}}(S'), \mathbf{y}_{S'}^*; z_i))^2\right]
$$
$$
\leq \mathbb{E}\mathbb{E}_{S'}\left[L^2(\|\mathbf{x}_{S'}^* - A_{\mathbf{x}}(S')\| + \|A_{\mathbf{y}}(S') - \mathbf{y}_{S'}^*\|)^2\right]
$$
$$
\leq 2L^2 \mathbb{E}\mathbb{E}_{S'}[\|\mathbf{x}_{S'}^* - A_{\mathbf{x}}(S')\|^2 + \|A_{\mathbf{y}}(S') - \mathbf{y}_{S'}^*\|^2]
$$
$$
\leq 4L^2\mu^{-1}\mathbb{E}\mathbb{E}_{S'}[F(A_{\mathbf{x}}(S'), \mathbf{y}_{S'}^*) - F(\mathbf{x}_{S'}^*, A_{\mathbf{x}}(S'))]
$$
$$
= 4L^2\mu^{-1}\mathbb{E}_{S'}[F(A_{\mathbf{x}}(S'), \mathbf{y}_{S'}^*) - F(\mathbf{x}_{S'}^*, A_{\mathbf{x}}(S'))], \tag{28}
$$

where the first inequality follows from Jensen's inequality and the Lipschitz continuity of $f$ (Assumption 1), the second inequality follows from that $(a+b)^2 \leq 2a^2 + 2b^2$ and the third inequality follows from the property of strong convexity and strong concavity of $F$ and the optimality condition, derived as follows,

$$
F(A_{\mathbf{x}}(S'), \mathbf{y}_{S'}^*) - F(\mathbf{x}_{S'}^*, A_{\mathbf{x}}(S'))
$$
$$
= F(A_{\mathbf{x}}(S'), \mathbf{y}_{S'}^*) - F(\mathbf{x}_{S'}^*, \mathbf{y}_{S'}^*) + F(\mathbf{x}_{S'}^*, \mathbf{y}_{S'}^*) - F(\mathbf{x}_{S'}^*, A_{\mathbf{x}}(S'))
$$
$$
\geq \frac{\mu}{2}\left[\|A_{\mathbf{x}}(S') - \mathbf{x}_{S'}^*\|^2 + \|\mathbf{y}_{S'}^* - A_{\mathbf{x}}(S')\|\right].
$$

It is clear that $\mathbb{E}[(Z - \mathbb{E}Z)^2] \leq \mathbb{E}[Z^2]$. Therefore, by the variance bound in (28) and applying the moment Bernstein inequality in Lemma 6 to the sum of independent random variables $-\mathbb{E}_{S'}f(A_{\mathbf{x}}(S'), \mathbf{y}_{S'}^*; z_i) + \mathbb{E}_{S'}f(\mathbf{x}_{S'}^*, A_{\mathbf{y}}(S'); z_i) + \mathbb{E}_S F(A_{\mathbf{x}}(S), \mathbf{y}_S^*) - \mathbb{E}_S F(\mathbf{x}_S^*, A_{\mathbf{y}}(S))$, we have the following inequality for all $p \geq 2$,

$$
\left\|\frac{1}{n}\sum_{i=1}^n -\mathbb{E}_{S'}f(A_{\mathbf{x}}(S'), \mathbf{y}_{S'}^*; z_i) + \mathbb{E}_{S'}f(\mathbf{x}_{S'}^*, A_{\mathbf{y}}(S'); z_i) + \mathbb{E}_S F(A_{\mathbf{x}}(S), \mathbf{y}_S^*) - \mathbb{E}_S F(\mathbf{x}_S^*, A_{\mathbf{y}}(S))\right\|_p
$$
$$
\leq 6\sqrt{\mathbb{E}_{S'}[F(A_{\mathbf{x}}(S'), \mathbf{y}_{S'}^*) - F(\mathbf{x}_{S'}^*, A_{\mathbf{x}}(S'))]\frac{4L^2 p}{n\mu}} + \frac{16pM}{n}.
$$

From Definition 1, we know that the last term $F_S(A_{\mathbf{x}}(S), \hat{\mathbf{y}}_S^*) - F_S(\hat{\mathbf{x}}_S^*, A_{\mathbf{y}}(S))$ is actually the strong PD empirical risk $\triangle_S^s(A_{\mathbf{x}}(S), A_{\mathbf{y}}(S))$.

Based on the above analysis, we have derived that for each $p \geq 2$,

$$
\left\|F(A_{\mathbf{x}}(S), \mathbf{y}_S^*) - \inf_{\mathbf{x}' \in \mathcal{X}} F(\mathbf{x}', A_{\mathbf{y}}(S)) - \triangle_S^s(A_{\mathbf{x}}(S), A_{\mathbf{y}}(S))\right\|_p
$$
$$
\leq 12\sqrt{\mathbb{E}_{S'}[F(A_{\mathbf{x}}(S'), \mathbf{y}_{S'}^*) - F(\mathbf{x}_{S'}^*, A_{\mathbf{x}}(S'))]\frac{pL^2}{n\mu}} + \frac{16pM}{n} + 100\sqrt{2}\left(1 + \frac{\beta}{\mu}\right)\epsilon Lp\lceil\log_2 n\rceil
$$
$$
\leq \frac{\eta}{1+\eta}\mathbb{E}_{S'}[F(A_{\mathbf{x}}(S'), \mathbf{y}_{S'}^*) - F(\mathbf{x}_{S'}^*, A_{\mathbf{x}}(S'))] + \frac{1+\eta}{\eta}\frac{12pL^2}{n\mu} + \frac{16pM}{n}
$$
$$
+ 100\sqrt{2}\left(1 + \frac{\beta}{\mu}\right)\epsilon Lp\lceil\log_2 n\rceil, \tag{29}
$$

where the last inequality holds since for any $a, b, \eta > 0$, $\sqrt{ab} \leq \eta a + \frac{b}{\eta}$.

Taking $p = 2$ and using the Cauchy-Schwarz inequality, we obtain that

$$
\mathbb{E}_S\left[F(A_{\mathbf{x}}(S), \mathbf{y}_S^*) - \inf_{\mathbf{x}' \in \mathcal{X}} F(\mathbf{x}', A_{\mathbf{y}}(S)) - \triangle_S^s(A_{\mathbf{x}}(S), A_{\mathbf{y}}(S))\right]
$$
$$
\leq \|F(A_{\mathbf{x}}(S), \mathbf{y}_S^*) - \inf_{\mathbf{x}' \in \mathcal{X}} F(\mathbf{x}', A_{\mathbf{y}}(S)) - \triangle_S^s(A_{\mathbf{x}}(S), A_{\mathbf{y}}(S))\|_2
$$
$$
\leq \frac{\eta}{1+\eta}\mathbb{E}_{S'}[F(A_{\mathbf{x}}(S'), \mathbf{y}_{S'}^*) - F(\mathbf{x}_{S'}^*, A_{\mathbf{x}}(S'))] + \frac{1+\eta}{\eta}\frac{24L^2}{n\mu} + \frac{32M}{n}
$$
$$
+ 200\sqrt{2}\left(1 + \frac{\beta}{\mu}\right)\epsilon L\lceil\log_2 n\rceil.
$$

Since $\mathbb{E}_{S'}[F(A_{\mathbf{x}}(S'), \mathbf{y}_{S'}^*) - F(\mathbf{x}_{S'}^*, A_{\mathbf{x}}(S'))] = \mathbb{E}_S[F(A_{\mathbf{x}}(S), \mathbf{y}_S^*) - F(\mathbf{x}_S^*, A_{\mathbf{x}}(S))]$, we finally get

$$\mathbb{E}_S[F(A_{\mathbf{x}}(S), \mathbf{y}_S^*) - \inf_{\mathbf{x}' \in \mathcal{X}} F(\mathbf{x}', A_{\mathbf{y}}(S))]$$

$$\leq (1+\eta)\Big(\mathbb{E}_S \triangle_S^s (A_{\mathbf{x}}(S), A_{\mathbf{y}}(S)) + \frac{24L^2(1+\eta)}{n\mu\eta} + \frac{32M}{n} + 200\sqrt{2}\Big(1 + \frac{\beta}{\mu}\Big)\epsilon L\lceil \log_2 n\rceil\Big).$$

Plugging this inequality into (29), we thus have that for each $p \geq 2$,

$$\Big\| F(A_{\mathbf{x}}(S), \mathbf{y}_S^*) - \inf_{\mathbf{x}' \in \mathcal{X}} F(\mathbf{x}', A_{\mathbf{y}}(S)) - \triangle_S^s(A_{\mathbf{x}}(S), A_{\mathbf{y}}(S))\Big\|_p$$

$$\leq \eta\Big(\mathbb{E}_S \triangle_S^s (A_{\mathbf{x}}(S), A_{\mathbf{y}}(S)) + \frac{24L^2(1+\eta)}{n\mu\eta} + \frac{32M}{n} + 200\sqrt{2}\Big(1 + \frac{\beta}{\mu}\Big)\epsilon L\lceil \log_2 n\rceil\Big)$$

$$+ \frac{12pL^2(1+\eta)}{n\mu\eta} + \frac{16pM}{n} + 100\sqrt{2}\Big(1 + \frac{\beta}{\mu}\Big)\epsilon Lp\lceil \log_2 n\rceil.$$

According to Lemma 1, for any $\delta > 0$, with probability at least $1 - \delta$, there holds that

$$F(A_{\mathbf{x}}(S), \mathbf{y}_S^*) - \inf_{\mathbf{x}' \in \mathcal{X}} F(\mathbf{x}', A_{\mathbf{y}}(S)) \leq \triangle_S^s(A_{\mathbf{x}}(S), A_{\mathbf{y}}(S)) + \eta\mathbb{E}_S \triangle_S^s (A_{\mathbf{x}}(S), A_{\mathbf{y}}(S))$$

$$+ C(1+\eta)\Big(\frac{L^2(1+\eta)}{n\mu\eta} + \frac{M}{n} + \Big(1 + \frac{\beta}{\mu}\Big)\epsilon L \log_2 n\Big) \log\Big(\frac{1}{\delta}\Big), \tag{30}$$

where $C > 0$ is an absolute constant. The proof is complete. $\qquad\square$

## A.4 PROOF OF PART (D)

We now prove the strong PD generalization error bound.

*Proof.* From (30) in the proof of Part (c), we know that for any $\delta > 0$, with probability at least $1 - \delta$, there holds that

$$\triangle^s (A_{\mathbf{x}}(S), A_{\mathbf{y}}(S)) - \triangle_S^s(A_{\mathbf{x}}(S), A_{\mathbf{y}}(S))$$
$$= F(A_{\mathbf{x}}(S), \mathbf{y}_S^*) - \inf_{\mathbf{x}' \in \mathcal{X}} F(\mathbf{x}', A_{\mathbf{y}}(S)) - \triangle_S^s(A_{\mathbf{x}}(S), A_{\mathbf{y}}(S))$$
$$\leq \eta\mathbb{E}_S \triangle_S^s (A_{\mathbf{x}}(S), A_{\mathbf{y}}(S)) + C(1+\eta)\Big(\frac{L^2(1+\eta)}{n\mu\eta} + \frac{M}{n} + \Big(1 + \frac{\beta}{\mu}\Big)\epsilon L \log_2 n\Big) \log\Big(\frac{1}{\delta}\Big).$$

Therefore, the proof is complete. $\qquad\square$

## A.5 PROOF OF PART (E)

We finally prove the excess primal population risk bound.

*Proof.* Denote $\mathbf{x}^* = \arg\min_{\mathbf{x} \in \mathcal{X}} R(\mathbf{x})$ and $\mathbf{y}_S^* = \arg\max_{\mathbf{y} \in \mathcal{Y}} F(A_{\mathbf{x}}(S), \mathbf{y})$. Firstly, we have the following decomposition

$$R(A_{\mathbf{x}}(S)) - \inf_{\mathbf{x}' \in \mathcal{X}} R(\mathbf{x}') = R(A_{\mathbf{x}}(S)) - R_S(A_{\mathbf{x}}(S)) + R_S(A_{\mathbf{x}}(S)) - F_S(\mathbf{x}^*, A_{\mathbf{y}}(S))$$
$$+ F_S(\mathbf{x}^*, A_{\mathbf{y}}(S)) - F(\mathbf{x}^*, A_{\mathbf{y}}(S)) + F(\mathbf{x}^*, A_{\mathbf{y}}(S)) - R(\mathbf{x}^*). \tag{31}$$

Consider the first term $R(A_{\mathbf{x}}(S)) - R_S(A_{\mathbf{x}}(S))$. From (22) of the Part (b), we know that with probability at least $1 - \delta$,

$$R(A_{\mathbf{x}}(S)) - R_S(A_{\mathbf{x}}(S)) \leq \frac{2M \log(3/\delta)}{3n} + 50\sqrt{2}\epsilon e L\frac{\beta + \mu}{\mu}\lceil \log_2 n\rceil \log(3e/\delta)$$

$$+ \sqrt{\frac{\Big(4MF(A_{\mathbf{x}}(S), \mathbf{y}_S^*) + \frac{1}{2}\Big(\frac{\beta}{\mu} + 1\Big)^2 L^2\epsilon^2 + 32n\Big(\frac{\beta}{\mu} + 1\Big)^2 L^2\epsilon^2 \log(3/\delta)\Big) \log(3/\delta)}{n}}.$$

For the second term $R_S(A_{\mathbf{x}}(S)) - F_S(\mathbf{x}^*, A_{\mathbf{y}}(S))$, we have $R_S(A_{\mathbf{x}}(S)) - F_S(\mathbf{x}^*, A_{\mathbf{y}}(S)) \leq R_S(A_{\mathbf{x}}(S)) - \inf_{\mathbf{x}' \in \mathcal{Y}} F_S(\mathbf{x}', A_{\mathbf{y}}(S)) = \triangle_S^s(A_{\mathbf{x}}(S), A_{\mathbf{y}}(S))$.

Note that under Assumption 1, the argument stability implies the uniform stability. Therefore, for the third term $F_S(\mathbf{x}^*, A_{\mathbf{y}}(S)) - F(\mathbf{x}^*, A_{\mathbf{y}}(S))$, from (10) of Part (a), we know that with probability at least $1 - \delta$

$$F_S(\mathbf{x}^*, A_{\mathbf{y}}(S)) - F(\mathbf{x}^*, A_{\mathbf{y}}(S)) \leq \frac{2M \log(3/\delta)}{3n} + 50\sqrt{2} e\epsilon \lceil \log_2 n \rceil \log(3e/\delta)$$
$$+ \sqrt{\frac{(4MF(\mathbf{x}^*, A_{\mathbf{y}}(S)) + \frac{1}{2}\epsilon^2 + 32n\epsilon^2 \log(3/\delta)) \log(3/\delta)}{n}}.$$

It is clear that $F(\mathbf{x}^*, A_{\mathbf{y}}(S)) - R(\mathbf{x}^*) \leq 0$.

Since $F(\mathbf{x}^*, A_{\mathbf{y}}(S)) \leq \sup_{\mathbf{y}' \in \mathcal{Y}} F(\mathbf{x}^*, \mathbf{y}') = R(\mathbf{x}^*) = \inf_{\mathbf{x}' \in \mathcal{X}} R(\mathbf{x})$, based on the above results, we have the following inequality with probability at least $1 - 2\delta$

$$R(A_{\mathbf{x}}(S)) - \inf_{\mathbf{x}' \in \mathcal{X}} R(\mathbf{x})$$

$$\leq \sqrt{\frac{\left(4MF(A_{\mathbf{x}}(S), \mathbf{y}_S^*) + \frac{1}{2}\left(\frac{\beta}{\mu}+1\right)^2 L^2\epsilon^2 + 32n\left(\frac{\beta}{\mu}+1\right)^2 L^2\epsilon^2 \log(3/\delta)\right) \log(3/\delta)}{n}}$$

$$+ \frac{4M \log(3/\delta)}{3n} + \sqrt{\frac{(4M \inf_{\mathbf{x}' \in \mathcal{X}} R(\mathbf{x}) + \frac{1}{2}\epsilon^2 + 32n\epsilon^2 \log(3/\delta)) \log(3/\delta)}{n}}$$

$$+ 50\sqrt{2}\epsilon eL\frac{\beta + \mu}{\mu}\lceil \log_2 n \rceil \log(3e/\delta) + \triangle_S^s(A_{\mathbf{x}}(S), A_{\mathbf{y}}(S)) + 50\sqrt{2}e\epsilon\lceil \log_2 n \rceil \log(3e/\delta)$$

$$\leq \sqrt{\frac{\left(\frac{1}{2}\left(\frac{\beta}{\mu}+1\right)^2 L^2\epsilon^2 + 32n\left(\frac{\beta}{\mu}+1\right)^2 L^2\epsilon^2 \log(3/\delta)\right) \log(3/\delta)}{n}} + \frac{\eta}{1+\eta}F(A_{\mathbf{x}}(S), \mathbf{y}_S^*)$$

$$+ \frac{1+\eta}{\eta}\frac{4M \log(3/\delta)}{n} + \frac{4M \log(3/\delta)}{3n} + \sqrt{\frac{(\frac{1}{2}\epsilon^2 + 32n\epsilon^2 \log(3/\delta)) \log(3/\delta)}{n}}$$

$$+ \frac{\eta}{1+\eta}\inf_{\mathbf{x}' \in \mathcal{X}} R(\mathbf{x}) + \frac{1+\eta}{\eta}\frac{4M \log(3/\delta)}{n}$$

$$+ 50\sqrt{2}\epsilon eL\frac{\beta + \mu}{\mu}\lceil \log_2 n \rceil \log(3e/\delta) + \triangle_S^s(A_{\mathbf{x}}(S), A_{\mathbf{y}}(S)) + 50\sqrt{2}e\epsilon\lceil \log_2 n \rceil \log(3e/\delta),$$

where the last inequality follows from the elementary inequalities $\sqrt{ab} \leq \eta a + \frac{1}{\eta}b$ and $\sqrt{a+b} \leq \sqrt{a} + \sqrt{b}$ for any $a, b > 0$. Therefore, by a rearrangement, we have the following inequality with probability at least $1 - \delta$

$$R(A_{\mathbf{x}}(S)) - (1 + 2\eta)\inf_{\mathbf{x}' \in \mathcal{X}} R(\mathbf{x})$$
$$\leq C\frac{1+\eta}{\eta}\left(\frac{M}{n}\log\frac{1}{\delta} + \left(\frac{\beta}{\mu}+1\right)L\epsilon \log_2 n \log\frac{1}{\delta} + \triangle_S^s(A_{\mathbf{x}}(S), A_{\mathbf{y}}(S))\right),$$

that is

$$R(A_{\mathbf{x}}(S)) - (1 + \eta)\inf_{\mathbf{x}' \in \mathcal{X}} R(\mathbf{x})$$
$$\leq C\frac{2+\eta}{\eta}\left(\frac{M}{n}\log\frac{1}{\delta} + \left(\frac{\beta}{\mu}+1\right)L\epsilon \log_2 n \log\frac{1}{\delta} + \triangle_S^s(A_{\mathbf{x}}(S), A_{\mathbf{y}}(S))\right),$$

where $C$ is an absolute constant. The proof is complete. $\qquad \square$

Till here, the proof of Theorem 1 is complete.

## B  EMPIRICAL SADDLE POINT

Empirical saddle point (ESP) problem refers to problem (2), which is also known as sample average approximation (SAA) (Zhang et al., 2021a). We denote $(\hat{\mathbf{x}}_S^*, \hat{\mathbf{y}}_S^*)$ as the ESP solution to (2), which is analogy to the ERM in stochastic optimization (Shalev-Shwartz et al., 2010). We first provide the main theorem of the ESP solution, as shown below.

**Theorem 3.** *Assume for all $z$, the function $(\mathbf{x}, \mathbf{y}) \mapsto f(\mathbf{x}, \mathbf{y}; z)$ is $\mu$-SC-SC. Suppose $|f(\mathbf{x}, \mathbf{y}; z)| \leq M$ for some $M > 0$ and $\mathbf{x} \in \mathcal{X}, \mathbf{y} \in \mathcal{Y}, z \in \mathcal{Z}$. Denote $A_{\mathbf{x}}(S) = \hat{\mathbf{x}}_S^*$ and $A_{\mathbf{y}}(S) = \hat{\mathbf{y}}_S^*$ for $(\hat{\mathbf{x}}_S^*, \hat{\mathbf{y}}_S^*)$. Fixed any $\eta > 0$. There exists an absolute positive constant $C$.*

*(a) If Assumption 1 holds, then for any $\delta > 0$, with probability at least $1 - \delta$, we have*

$$F(\hat{\mathbf{x}}_S^*, \hat{\mathbf{y}}_S^*) \leq (1 + \eta) F_S(\hat{\mathbf{x}}_S^*, \hat{\mathbf{y}}_S^*) + C \frac{1 + \eta}{\eta} \Big( \frac{M}{n} \log(1/\delta) + \frac{4L}{n\mu} \log_2 n \log(1/\delta) \Big).$$

*(b) If Assumptions 1 and 2 hold, then for any $\delta > 0$, with probability at least $1 - \delta$, we have*

$$R(\hat{\mathbf{x}}_S^*) \leq (1 + \eta) R_S(\hat{\mathbf{x}}_S^*) + C \frac{1 + \eta}{\eta} \Big( \frac{M}{n} \log \frac{1}{\delta} + \Big( \frac{\beta}{\mu} + 1 \Big) \frac{4L^2}{n\mu} \log_2 n \log \frac{1}{\delta} \Big).$$

*(c) If Assumptions 1 and 2 hold, then for any $\delta > 0$, with probability at least $1 - \delta$, we have*

$$\triangle^s(\hat{\mathbf{x}}_S^*, \hat{\mathbf{y}}_S^*) \leq C(1 + \eta) \Big( \frac{L^2(1 + \eta)}{n\mu\eta} + \frac{M}{n} + \Big( 1 + \frac{\beta}{\mu} \Big) \frac{4L^2}{n\mu} \log_2 n \Big) \log \Big( \frac{1}{\delta} \Big).$$

*(d) If Assumptions 1 and 2 hold, then for any $\delta > 0$, with probability at least $1 - \delta$, we have*

$$\triangle^s(\hat{\mathbf{x}}_S^*, \hat{\mathbf{y}}_S^*) - \triangle_S^s(\hat{\mathbf{x}}_S^*, \hat{\mathbf{y}}_S^*)$$
$$\leq C(1 + \eta) \Big( \frac{L^2(1 + \eta)}{n\mu\eta} + \frac{M}{n} + \Big( 1 + \frac{\beta}{\mu} \Big) \frac{4L^2}{n\mu} \log_2 n \Big) \log \Big( \frac{1}{\delta} \Big).$$

*(e) If Assumptions 1 and 2 hold, then for any $\delta > 0$, with probability at least $1 - \delta$, we have*

$$R(\hat{\mathbf{x}}_S^*) \leq (1 + \eta) \inf_{\mathbf{x} \in \mathcal{X}} R(\mathbf{x}) + C \frac{2 + \eta}{\eta} \Big( \frac{M}{n} \log \frac{1}{\delta} + \Big( \frac{\beta}{\mu} + 1 \Big) \frac{4L^2}{n\mu} \log_2 n \log \frac{1}{\delta} \Big).$$

*Proof.* To prove Theorem 3, we should derive the strong PD empirical risk bound and the stability bound of $(\hat{\mathbf{x}}_S^*, \hat{\mathbf{y}}_S^*)$. It is easy to verify that $\triangle_S^s(\hat{\mathbf{x}}_S^*, \hat{\mathbf{y}}_S^*) = 0$ (Zhang et al., 2021a). We then investigate the stability bound of $(\hat{\mathbf{x}}_S^*, \hat{\mathbf{y}}_S^*)$.

Let $S = \{z_1, ..., z_n\}$ be a set of independent random variables each taking values in $\mathcal{Z}$. For any $i \in [n]$, define $S^{(i)} = \{z_1, ..., z_{i-1}, z_i', z_{i+1}, ..., z_n\}$ be a dataset by replacing the $i$-th sample in $S$ with another i.i.d. sample $z_i'$. We define $F_{S^{(i)}}$ be the empirical risk on dataset $S^{(i)}$ and define $(\hat{\mathbf{x}}_{S^{(i)}}^*, \hat{\mathbf{y}}_{S^{(i)}}^*)$ be the ESP solution on dataset $S^{(i)}$.

Then we have

$$F_S(\hat{\mathbf{x}}_{S^{(i)}}^*, \hat{\mathbf{y}}_S^*) - F_S(\hat{\mathbf{x}}_S^*, \hat{\mathbf{y}}_{S^{(i)}}^*)$$

$$= \frac{1}{n} \sum_{j=1}^n \left( f(\hat{\mathbf{x}}_{S^{(i)}}^*, \hat{\mathbf{y}}_S^*; z_j) - f(\hat{\mathbf{x}}_S^*, \hat{\mathbf{y}}_{S^{(i)}}^*; z_j) \right)$$

$$= \frac{1}{n} \Big( \sum_{j=1, j\neq i}^n (f(\hat{\mathbf{x}}_{S^{(i)}}^*, \hat{\mathbf{y}}_S^*; z_j) - f(\hat{\mathbf{x}}_S^*, \hat{\mathbf{y}}_{S^{(i)}}^*; z_j)) + f(\hat{\mathbf{x}}_{S^{(i)}}^*, \hat{\mathbf{y}}_S^*; z_i') - f(\hat{\mathbf{x}}_S^*, \hat{\mathbf{y}}_{S^{(i)}}^*; z_i') \Big)$$

$$+ \frac{1}{n} \left( f(\hat{\mathbf{x}}_{S^{(i)}}^*, \hat{\mathbf{y}}_S^*; z_i) - f(\hat{\mathbf{x}}_S^*, \hat{\mathbf{y}}_{S^{(i)}}^*; z_i) \right) - \frac{1}{n} \left( f(\hat{\mathbf{x}}_{S^{(i)}}^*, \hat{\mathbf{y}}_S^*; z_i') - f(\hat{\mathbf{x}}_S^*, \hat{\mathbf{y}}_{S^{(i)}}^*; z_i') \right)$$

$$= F_{S^{(i)}}(\hat{\mathbf{x}}_{S^{(i)}}^*, \hat{\mathbf{y}}_S^*) - F_{S^{(i)}}(\hat{\mathbf{x}}_S^*, \hat{\mathbf{y}}_{S^{(i)}}^*)$$

$$+ \frac{1}{n} \left( f(\hat{\mathbf{x}}_{S^{(i)}}^*, \hat{\mathbf{y}}_S^*; z_i) - f(\hat{\mathbf{x}}_S^*, \hat{\mathbf{y}}_S^*; z_i) + f(\hat{\mathbf{x}}_S^*, \hat{\mathbf{y}}_S^*; z_i) - f(\hat{\mathbf{x}}_S^*, \hat{\mathbf{y}}_{S^{(i)}}^*; z_i) \right)$$

$$- \frac{1}{n} \left( f(\hat{\mathbf{x}}_{S^{(i)}}^*, \hat{\mathbf{y}}_S^*; z_i') - f(\hat{\mathbf{x}}_S^*, \hat{\mathbf{y}}_S^*; z_i') + f(\hat{\mathbf{x}}_S^*, \hat{\mathbf{y}}_S^*; z_i') - f(\hat{\mathbf{x}}_S^*, \hat{\mathbf{y}}_{S^{(i)}}^*; z_i') \right)$$

$$\leq F_{S^{(i)}}(\hat{\mathbf{x}}_{S^{(i)}}^*, \hat{\mathbf{y}}_S^*) - F_{S^{(i)}}(\hat{\mathbf{x}}_S^*, \hat{\mathbf{y}}_{S^{(i)}}^*) + \frac{2L}{n}(\|\hat{\mathbf{x}}_{S^{(i)}}^* - \hat{\mathbf{x}}_S^*\| + \|\hat{\mathbf{y}}_S^* - \hat{\mathbf{y}}_{S^{(i)}}^*\|)$$

$$= F_{S^{(i)}}(\hat{\mathbf{x}}_{S^{(i)}}^*, \hat{\mathbf{y}}_S^*) - F_{S^{(i)}}(\hat{\mathbf{x}}_{S^{(i)}}^*, \hat{\mathbf{y}}_{S^{(i)}}^*) + F_{S^{(i)}}(\hat{\mathbf{x}}_{S^{(i)}}^*, \hat{\mathbf{y}}_{S^{(i)}}^*) - F_{S^{(i)}}(\hat{\mathbf{x}}_S^*, \hat{\mathbf{y}}_{S^{(i)}}^*)$$

$$+ \frac{2L}{n}(\|\hat{\mathbf{x}}_{S^{(i)}}^* - \hat{\mathbf{x}}_S^*\| + \|\hat{\mathbf{y}}_S^* - \hat{\mathbf{y}}_{S^{(i)}}^*\|)$$

$$\leq -\frac{\mu}{2}\|\hat{\mathbf{x}}_{S^{(i)}}^* - \hat{\mathbf{x}}_S^*\|^2 - \frac{\mu}{2}\|\hat{\mathbf{y}}_S^* - \hat{\mathbf{y}}_{S^{(i)}}^*\|^2 + \frac{2L}{n}(\|\hat{\mathbf{x}}_{S^{(i)}}^* - \hat{\mathbf{x}}_S^*\| + \|\hat{\mathbf{y}}_S^* - \hat{\mathbf{y}}_{S^{(i)}}^*\|),$$

where the first inequality follows from the Lipschitz continuous assumption, and where the second inequality follows from the facts that the $\mu$-SC-SC property of $F_{S^{(i)}}$ and $(\hat{\mathbf{x}}_{S^{(i)}}^*, \hat{\mathbf{y}}_{S^{(i)}}^*)$ is the ESP solution of $F_{S^{(i)}}$.

Similarly, according to the $\mu$-SC-SC property of $F_S$, we have

$$F_S(\hat{\mathbf{x}}_{S^{(i)}}^*, \hat{\mathbf{y}}_S^*) - F_S(\hat{\mathbf{x}}_S^*, \hat{\mathbf{y}}_{S^{(i)}}^*)$$
$$= F_S(\hat{\mathbf{x}}_{S^{(i)}}^*, \hat{\mathbf{y}}_S^*) - F_S(\hat{\mathbf{x}}_S^*, \hat{\mathbf{y}}_S^*) + F_S(\hat{\mathbf{x}}_S^*, \hat{\mathbf{y}}_S^*) - F_S(\hat{\mathbf{x}}_S^*, \hat{\mathbf{y}}_{S^{(i)}}^*)$$
$$\geq \frac{\mu}{2}\|\hat{\mathbf{x}}_{S^{(i)}}^* - \hat{\mathbf{x}}_S^*\|^2 + \frac{\mu}{2}\|\hat{\mathbf{y}}_S^* - \hat{\mathbf{y}}_{S^{(i)}}^*\|^2 \tag{32}$$

Based on the above results, we have

$$\mu\|\hat{\mathbf{x}}_{S^{(i)}}^* - \hat{\mathbf{x}}_S^*\|^2 + \mu\|\hat{\mathbf{y}}_S^* - \hat{\mathbf{y}}_{S^{(i)}}^*\|^2$$
$$\leq \frac{2L}{n}(\|\hat{\mathbf{x}}_{S^{(i)}}^* - \hat{\mathbf{x}}_S^*\| + \|\hat{\mathbf{y}}_S^* - \hat{\mathbf{y}}_{S^{(i)}}^*\|)$$
$$\leq \frac{2L}{n}\sqrt{2}\sqrt{\|\hat{\mathbf{x}}_{S^{(i)}}^* - \hat{\mathbf{x}}_S^*\|^2 + \|\hat{\mathbf{y}}_S^* - \hat{\mathbf{y}}_{S^{(i)}}^*\|^2},$$

where the last inequality uses the Caucy-Schwarz inequality. Therefore, we have

$$\|\hat{\mathbf{x}}_{S^{(i)}}^* - \hat{\mathbf{x}}_S^*\| + \|\hat{\mathbf{y}}_S^* - \hat{\mathbf{y}}_{S^{(i)}}^*\|$$
$$\leq \sqrt{2(\|\hat{\mathbf{x}}_{S^{(i)}}^* - \hat{\mathbf{x}}_S^*\|^2 + \|\hat{\mathbf{y}}_S^* - \hat{\mathbf{y}}_{S^{(i)}}^*\|^2)}$$
$$\leq \frac{4L}{n\mu}. \tag{33}$$

Now, plugging this stability bound into Theorem 1, we obtain generalization bounds of the ESP solution. The proof of Theorem 3 is complete. $\square$

**Remark 8.** When conditions in Theorem 3 hold, we obtain that (a) If Assumption 1 holds and $F_S(\hat{\mathbf{x}}_S^*, \hat{\mathbf{y}}_S^*) = \mathcal{O}\left(\frac{1}{n}\right)$, then for any $\delta > 0$, with probability at least $1 - \delta$, the plain generalization error of $(\hat{\mathbf{x}}_S^*, \hat{\mathbf{y}}_S^*)$ is of the order $\mathcal{O}\left(\frac{\log_2 n}{n}\log(1/\delta)\right)$. (b) If Assumptions 1 and 2 hold and $R_S(\hat{\mathbf{x}}_S^*) = \mathcal{O}\left(\frac{1}{n}\right)$, then for any $\delta > 0$, with probability at least $1 - \delta$, the primal generalization error of

$(\hat{\mathbf{x}}_S^*, \hat{\mathbf{y}}_S^*)$ is of the order $\mathcal{O}\left(\frac{\log_2 n}{n} \log(1/\delta)\right)$. (c) If Assumptions 1 and 2 hold, then for any $\delta > 0$, with probability at least $1 - \delta$, the strong PD population risk and the strong PD generalization error of $(\hat{\mathbf{x}}_S^*, \hat{\mathbf{y}}_S^*)$ are all of the order $\mathcal{O}\left(\frac{\log_2 n}{n} \log(1/\delta)\right)$. (d) If Assumptions 1 and 2 hold and $\inf_{\mathbf{x} \in \mathcal{X}} R(\mathbf{x}) = \mathcal{O}\left(\frac{1}{n}\right)$, then for any $\delta > 0$, with probability at least $1 - \delta$, the excess primal population risk of $(\hat{\mathbf{x}}_S^*, \hat{\mathbf{y}}_S^*)$ is of the order $\mathcal{O}\left(\frac{\log_2 n}{n} \log(1/\delta)\right)$.

**Remark 9.** (Zhang et al., 2021a) also studies the generalization bound of the ESP solution. They provide $\mathcal{O}(1/n)$ order bounds for weak PD population risk and expected strong PD population risk. Their proofs also show that the expected strong PD population risk is more difficult to analyze than the former. They have to consider the fact that different $\hat{\mathbf{x}}_S^*$ corresponds to different $\mathbf{y}$, as discussed in Remark 5. Moreover, the expectation operator in expected strong PD population risk also relaxes the difficulty of proof. Specifically, define $S^{(i)}$ be a dataset by replacing the $i$-th sample in $S$ with another i.i.d. sample $z_i'$ and $\mathbf{y}^*(\mathbf{x}) = \arg\max_{\mathbf{y} \in \mathcal{Y}} F(\mathbf{x}, \mathbf{y})$, there holds the following important property $\mathbb{E}\left[\sup_{\mathbf{y} \in \mathcal{Y}} F(\hat{\mathbf{x}}_S^*, \mathbf{y})\right] = \frac{1}{n}\sum_{i=1}^n \mathbb{E}\left[F(\hat{\mathbf{x}}_{S^{(i)}}^*, \mathbf{y}^*(\hat{\mathbf{x}}_{S^{(i)}}^*))\right] = \frac{1}{n}\sum_{i=1}^n \mathbb{E}\left[f(\hat{\mathbf{x}}_{S^{(i)}}^*, \mathbf{y}^*(\hat{\mathbf{x}}_{S^{(i)}}^*); z_i)\right]$ because $(\hat{\mathbf{x}}_S^*, \mathbf{y}^*(\hat{\mathbf{x}}_S^*))$ and $(\hat{\mathbf{x}}_{S^{(i)}}^*, \mathbf{y}^*(\hat{\mathbf{x}}_{S^{(i)}}^*))$ are identically distributed and the independence between $z_i$ and $S^{(i)}$. On the contrary, when there is no expectation operator, we do not have this property and the proof is much more challenging.

## C  GRADIENT DESCENT ASCENT

We need some notations to state results on GDA. Specifically, assume the initial point satisfies $\mathbf{x}_1 = 0$ and $\mathbf{y}_1 = 0$. Let $\{\eta_t\}$ be a sequence of positive step sizes. At the $t$-th iteration, GDA updates

$$\begin{aligned} \mathbf{x}_{t+1} &= \mathbf{x}_t - \eta_t \nabla_{\mathbf{x}} F_S(\mathbf{x}_t, \mathbf{y}_t), \\ \mathbf{y}_{t+1} &= \mathbf{y}_t + \eta_t \nabla_{\mathbf{y}} F_S(\mathbf{x}_t, \mathbf{y}_t). \end{aligned} \tag{34}$$

We denote the average of iterates by

$$\bar{\mathbf{x}}_T = \frac{\sum_{t=1}^T \mathbf{x}_t}{T} \quad \text{and} \quad \bar{\mathbf{y}}_T = \frac{\sum_{t=1}^T \mathbf{y}_t}{T}. \tag{35}$$

Here, we first provide an important lemma to connect the argument stability with the strong PD empirical risk, which will also be used in the remaining applications.

**Lemma 7.** *For any $i \in [n]$, define $S^{(i)} = \{z_1, ..., z_{i-1}, z_i', z_{i+1}, ..., z_n\}$. Let $(\mathbf{x}_t, \mathbf{y}_t)$ be the output produced by $F_S$ on dataset $S$ in running a minimax learning algorithm. Let $(\mathbf{x}_t^i, \mathbf{y}_t^i)$ be the corresponding output produced by $F_{S^{(i)}}$ on dataset $S^{(i)}$, where $F_{S^{(i)}}$ is empirical risk on dataset $S^{(i)}$. Suppose Assumption 1 holds. Assume for all $z$, the function $(\mathbf{x}, \mathbf{y}) \mapsto f(\mathbf{x}, \mathbf{y}; z)$ is $\mu$-SC-SC. For any $S^{(i)}$ and $S$, we have*

$$\|\mathbf{x}_t^i - \mathbf{x}_t\| + \|\mathbf{y}_t^i - \mathbf{y}_t\| \leq \frac{4L}{n\mu} + 4\sqrt{\frac{1}{\mu}}\sqrt{\triangle_S^s(\mathbf{x}_t, \mathbf{y}_t)}.$$

*Proof.* Define $(\hat{\mathbf{x}}_{S^{(i)}}^*, \hat{\mathbf{y}}_{S^{(i)}}^*)$ be the ESP solution on dataset $S^{(i)}$ and $(\hat{\mathbf{x}}_S^*, \hat{\mathbf{y}}_S^*)$ be the ESP solution on dataset $S$. To prove the stability bound, we consider

$$\begin{aligned} &\|\mathbf{x}_t^i - \mathbf{x}_t\| + \|\mathbf{y}_t^i - \mathbf{y}_t\| \\ =&\|\mathbf{x}_t^i - \hat{\mathbf{x}}_{S^{(i)}}^* + \hat{\mathbf{x}}_{S^{(i)}}^* - \hat{\mathbf{x}}_S^* + \hat{\mathbf{x}}_S^* - \mathbf{x}_t\| + \|\mathbf{y}_t^i - \hat{\mathbf{y}}_{S^{(i)}}^* + \hat{\mathbf{y}}_{S^{(i)}}^* - \hat{\mathbf{y}}_S^* + \hat{\mathbf{y}}_S^* - \mathbf{y}_t\| \\ \leq&\|\mathbf{x}_t^i - \hat{\mathbf{x}}_{S^{(i)}}^*\| + \|\hat{\mathbf{x}}_{S^{(i)}}^* - \hat{\mathbf{x}}_S^*\| + \|\hat{\mathbf{x}}_S^* - \mathbf{x}_t\| + \|\mathbf{y}_t^i - \hat{\mathbf{y}}_{S^{(i)}}^*\| + \|\hat{\mathbf{y}}_{S^{(i)}}^* - \hat{\mathbf{y}}_S^*\| + \|\hat{\mathbf{y}}_S^* - \mathbf{y}_t\| \\ \leq&\frac{4L}{n\mu} + \|\mathbf{x}_t^i - \hat{\mathbf{x}}_{S^{(i)}}^*\| + \|\hat{\mathbf{x}}_S^* - \mathbf{x}_t\| + \|\mathbf{y}_t^i - \hat{\mathbf{y}}_{S^{(i)}}^*\| + \|\hat{\mathbf{y}}_S^* - \mathbf{y}_t\| \\ \leq&\frac{4L}{n\mu} + \sqrt{2}\sqrt{\|\mathbf{x}_t^i - \hat{\mathbf{x}}_{S^{(i)}}^*\|^2 + \|\mathbf{y}_t^i - \hat{\mathbf{y}}_{S^{(i)}}^*\|^2} + \sqrt{2}\sqrt{\|\hat{\mathbf{y}}_S^* - \mathbf{y}_t\|^2 + \|\hat{\mathbf{x}}_S^* - \mathbf{x}_t\|^2} \\ \leq&\frac{4L}{n\mu} + \sqrt{\frac{4}{\mu}}\sqrt{F_{S^{(i)}}(\mathbf{x}_t^i, \hat{\mathbf{y}}_{S^{(i)}}^*) - F_{S^{(i)}}(\hat{\mathbf{x}}_{S^{(i)}}^*, \mathbf{y}_t^i)} + \sqrt{\frac{4}{\mu}}\sqrt{F_S(\mathbf{x}_t, \hat{\mathbf{y}}_S^*) - F_S(\hat{\mathbf{x}}_S^*, \mathbf{y}_t)}, \end{aligned}$$

where the second inequality uses the result in (33), the third inequality uses the Caucy-Schwarz inequality, and the last inequality uses the strong convexity and strong concavity of $F_{S^{(i)}}$ and $F_S$ and the optimality condition (please refer to (32)). As will see in the rest paper, we bound $F_{S^{(i)}}(\mathbf{x}_t^i, \hat{\mathbf{y}}_{S^{(i)}}^*) - F_{S^{(i)}}(\hat{\mathbf{x}}_{S^{(i)}}^*, \hat{\mathbf{y}}_{S^{(i)}}^*)$ and $F_S(\mathbf{x}_t, \hat{\mathbf{y}}_S^*) - F_S(\hat{\mathbf{x}}_S^*, \mathbf{y}_t)$ with the same upper bound since they are all strong PD empirical risk. Thus, for brevity, we derive the following inequality

$$\|\mathbf{x}_t^i - \mathbf{x}_t\| + \|\mathbf{y}_t^i - \mathbf{y}_t\|$$
$$\leq \frac{4L}{n\mu} + 4\sqrt{\frac{1}{\mu}}\sqrt{F_S(\mathbf{x}_t, \hat{\mathbf{y}}_S^*) - F_S(\hat{\mathbf{x}}_S^*, \mathbf{y}_t)}$$
$$\leq \frac{4L}{n\mu} + 4\sqrt{\frac{1}{\mu}}\sqrt{\triangle_S^s(\mathbf{x}_t, \mathbf{y}_t)}.$$

The proof is complete. □

**Remark 10.** Lemma 7 provides the connection between the stability bound and the strong PD empirical risk. The subscript $t$ here represents not only the output of an iterative optimization algorithm, but any output of the empirical risk of any minimax learning algorithm.

**Remark 11.** In studying the stability bound of gradient-based optimization algorithms, a popular approach is to use the property of smoothness to establish the nonexpansiveness of gradient mapping, proposed in the seminal work (Hardt et al., 2016). (Farnia & Ozdaglar, 2021; Lei et al., 2021) extend this approach to the minimax problems and use it to analyze the stability bound of SGDA, GDA, PPM, etc. However, their stability bounds are often derived in expectation. In (Lei et al., 2021), the authors also use the Chernoff bounds of Bernoulli variables to establish high probability stability bounds when they are to derive high probability generalization bounds. Unfortunately, these stability bounds are often of slow order $\mathcal{O}(1/\sqrt{n})$. To derive sharper stability bounds, we established Lemma 7.

The following lemma shows the strong PD empirical risk of GDA.

**Lemma 8.** *Suppose Assumption 1 holds and $F_S(\cdot, \cdot)$ be $\mu$-SC-SC with $\mu > 0$. Let $\{\mathbf{x}_t, \mathbf{y}_t\}$ be the sequence produced by (34) with $\eta_t = \frac{1}{\mu(t+t_0)}$. Assume $t_0 \geq 0$. Suppose $\sup_{\mathbf{x} \in \mathcal{X}} \|\mathbf{x}\| \leq R_X$ and $\sup_{\mathbf{y} \in \mathcal{Y}} \|\mathbf{y}\| \leq R_Y$. Then for $(\bar{\mathbf{x}}_T, \bar{\mathbf{y}}_T)$ in (35) we have*

$$\sup_{\mathbf{y} \in \mathcal{Y}} F_S(\bar{\mathbf{x}}_T, \mathbf{y}) - \inf_{\mathbf{x} \in \mathcal{X}} F_S(\mathbf{x}, \bar{\mathbf{y}}_T) \leq \frac{\mu t_0 (R_X^2 + R_Y^2)}{T} + \frac{L^2 \log(eT)}{\mu T}.$$

*If $t_0 = 0$, then*

$$\sup_{\mathbf{y} \in \mathcal{Y}} F_S(\bar{\mathbf{x}}_T, \mathbf{y}) - \inf_{\mathbf{x} \in \mathcal{X}} F_S(\mathbf{x}, \bar{\mathbf{y}}_T) \leq \frac{L^2 \log(eT)}{\mu T}.$$

*Proof.* Firstly, we have

$$\|\mathbf{x}_{t+1} - \mathbf{x}\|^2 = \|\mathbf{x}_t - \eta_t \nabla_\mathbf{x} F_S(\mathbf{x}_t, \mathbf{y}_t) - \mathbf{x}\|^2$$
$$= \|\mathbf{x}_t - \mathbf{x}\|^2 + \eta_t^2 \|\nabla_\mathbf{x} F_S(\mathbf{x}_t, \mathbf{y}_t)\|^2 + 2\eta_t \langle \mathbf{x} - \mathbf{x}_t, \nabla_\mathbf{x} F_S(\mathbf{x}_t, \mathbf{y}_t) \rangle$$
$$\leq \|\mathbf{x}_t - \mathbf{x}\|^2 + \eta_t^2 L^2 + 2\eta_t \langle \mathbf{x} - \mathbf{x}_t, \nabla_\mathbf{x} F_S(\mathbf{x}_t, \mathbf{y}_t) \rangle,$$

where the first inequality holds because of Assumption 1. By the strong convexity of $F_S(\cdot, \mathbf{y}_t)$, we have

$$2\eta_t (F_S(\mathbf{x}_t, \mathbf{y}_t) - F_S(\mathbf{x}, \mathbf{y}_t)) \leq (1 - \eta_t \mu)\|\mathbf{x}_t - \mathbf{x}\|^2 - \|\mathbf{x}_{t+1} - \mathbf{x}\|^2 + \eta_t^2 L^2.$$

Since $\eta_t = \frac{1}{\mu(t+t_0)}$, we further get

$$\frac{2}{\mu(t+t_0)}(F_S(\mathbf{x}_t, \mathbf{y}_t) - F_S(\mathbf{x}, \mathbf{y}_t)) \leq \left(1 - \frac{1}{(t+t_0)}\right)\|\mathbf{x}_t - \mathbf{x}\|^2 - \|\mathbf{x}_{t+1} - \mathbf{x}\|^2 + \left(\frac{1}{\mu(t+t_0)}\right)^2 L^2.$$

Multiplying both sides by $t + t_0$, we have

$$\frac{2}{\mu}(F_S(\mathbf{x}_t, \mathbf{y}_t) - F_S(\mathbf{x}, \mathbf{y}_t)) \leq (t + t_0 - 1)\|\mathbf{x}_t - \mathbf{x}\|^2 - (t + t_0)\|\mathbf{x}_{t+1} - \mathbf{x}\|^2 + \frac{L^2}{\mu^2(t + t_0)}.$$

Since $\mathbf{x}_1 = 0$ and $\sum_{t=1}^{T} t^{-1} \leq \log(eT)$, by taking a summation of the above inequality from $t = 1$ to $T$, we obtain

$$\sum_{t=1}^{T}(F_S(\mathbf{x}_t, \mathbf{y}_t) - F_S(\mathbf{x}, \mathbf{y}_t)) \leq \frac{\mu}{2} t_0 R_X^2 + \frac{L^2 \log(eT)}{2\mu}.$$

From the concavity of $F_S(\mathbf{x}, \cdot)$ we get

$$\sum_{t=1}^{T}(F_S(\mathbf{x}_t, \mathbf{y}_t) - F_S(\mathbf{x}, \bar{\mathbf{y}}_T)) \leq \frac{\mu}{2} t_0 R_X^2 + \frac{L^2 \log(eT)}{2\mu}.$$

Since this inequality holds for any $\mathbf{x}$, we get

$$\sum_{t=1}^{T}(F_S(\mathbf{x}_t, \mathbf{y}_t) - \inf_{\mathbf{x} \in \mathcal{X}} F_S(\mathbf{x}, \bar{\mathbf{y}}_T)) \leq \frac{\mu}{2} t_0 R_X^2 + \frac{L^2 \log(eT)}{2\mu}.$$

This implies that

$$\frac{1}{T}\sum_{t=1}^{T}(F_S(\mathbf{x}_t, \mathbf{y}_t) - \inf_{\mathbf{x} \in \mathcal{X}} F_S(\mathbf{x}, \bar{\mathbf{y}}_T)) \leq \frac{\mu t_0 R_X^2}{2T} + \frac{L^2 \log(eT)}{2\mu T}.$$

In a similar way, we have the following inequality

$$\sup_{\mathbf{y} \in \mathcal{Y}} F_S(\bar{\mathbf{x}}_T, \mathbf{y}) - \frac{1}{T}\sum_{t=1}^{T}(F_S(\mathbf{x}_t, \mathbf{y}_t)) \leq \frac{\mu t_0 R_Y^2}{2T} + \frac{L^2 \log(eT)}{2\mu T}.$$

Combined the above two inequalities together we get

$$\sup_{\mathbf{y} \in \mathcal{Y}} F_S(\bar{\mathbf{x}}_T, \mathbf{y}) - \inf_{\mathbf{x} \in \mathcal{X}} F_S(\mathbf{x}, \bar{\mathbf{y}}_T) \leq \frac{\mu t_0 (R_X^2 + R_Y^2)}{T} + \frac{L^2 \log(eT)}{\mu T}.$$

$\square$

For optimization algorithm GDA, substituting the strong PD empirical risk bound of $(\bar{\mathbf{x}}_T, \bar{\mathbf{y}}_T)$ into Lemma 7, we get the following stability bound,

$$\|\bar{\mathbf{x}}_T^i - \bar{\mathbf{x}}_T\| + \|\bar{\mathbf{y}}_T^i - \bar{\mathbf{y}}_T\| \leq \frac{4L}{n\mu} + 4\sqrt{\frac{1}{\mu}}\sqrt{\triangle_S^s(\bar{\mathbf{x}}_T, \bar{\mathbf{y}}_T)}$$

$$\leq \frac{4L}{n\mu} + 4\sqrt{\frac{1}{\mu}}\sqrt{\frac{\mu t_0(R_X^2 + R_Y^2)}{T} + \frac{L^2 \log(eT)}{\mu T}}. \tag{36}$$

Furthermore, for any $\mathbf{x} \in \mathcal{X}$, $\mathbf{y} \in \mathcal{Y}$ and $z \in \mathcal{Z}$,

$$f(\mathbf{x}, \mathbf{y}; z) - f(0, 0; z) \leq L\|\mathbf{x} - 0\| + L\|\mathbf{y} - 0\| \leq L(R_Y + R_Y),$$

which implies that

$$f(\mathbf{x}, \mathbf{y}; z) \leq \sup_{z \in \mathcal{Z}} f(0, 0; z) + L(R_Y + R_Y). \tag{37}$$

Till here, plugging (37), the stability bound in (36) and the strong PD empirical risk bound in Lemma 8 into Theorem 1, we obtain generalization bounds of GDA.

We now write the main theorem of GDA.

**Theorem 4.** *Assume for all $z$, the function $(\mathbf{x}, \mathbf{y}) \mapsto f(\mathbf{x}, \mathbf{y}; z)$ is $\mu$-SC-SC. Suppose $\sup_{\mathbf{x} \in \mathcal{X}} \|\mathbf{x}\| \leq R_X$ and $\sup_{\mathbf{y} \in \mathcal{Y}} \|\mathbf{y}\| \leq R_Y$. Let $\{\mathbf{x}_t, \mathbf{y}_t\}$ be produced by (34) with $\eta_t = \frac{1}{\mu(t+t_0)}$. Assume $t_0 \geq 0$. Denote $A_{\mathbf{x}}(S) = \bar{\mathbf{x}}_T$ and $A_{\mathbf{y}}(S) = \bar{\mathbf{y}}_T$ for $(\bar{\mathbf{x}}_T, \bar{\mathbf{y}}_T)$ in (35). Let $M = \sup_{z \in \mathcal{Z}} f(0, 0; z) + L(R_X + R_Y)$. Fixed any $\eta > 0$. There exists an absolute positive constant $C$.*

*(a) If Assumption 1 holds, then for any $\delta > 0$, with probability at least $1 - \delta$, we have*

$$F(\bar{\mathbf{x}}_T, \bar{\mathbf{y}}_T) \leq (1 + \eta) F_S(\bar{\mathbf{x}}_T, \bar{\mathbf{y}}_T)$$

$$+ C \frac{1 + \eta}{\eta} \Big( \frac{M}{n} \log(1/\delta) + \Big( \frac{4L}{n\mu} + 4\sqrt{\frac{1}{\mu}} \sqrt{\frac{\mu t_0 (R_X^2 + R_Y^2)}{T} + \frac{L^2 \log(eT)}{\mu T}} \Big) \log_2 n \log(1/\delta) \Big).$$

*(b) If Assumptions 1 and 2 hold, then for any $\delta > 0$, with probability at least $1 - \delta$, we have*

$$R(\bar{\mathbf{x}}_T) \leq (1 + \eta) R_S(\bar{\mathbf{x}}_T) + C \frac{1 + \eta}{\eta}$$

$$\times \Big( \frac{M \log \frac{1}{\delta}}{n} + \frac{\beta + \mu}{\mu} L \Big( \frac{4L}{n\mu} + 4\sqrt{\frac{1}{\mu}} \sqrt{\frac{\mu t_0 (R_X^2 + R_Y^2)}{T} + \frac{L^2 \log(eT)}{\mu T}} \Big) \log_2 n \log \frac{1}{\delta} \Big).$$

*(c) If Assumptions 1 and 2 hold, then for any $\delta > 0$, with probability at least $1 - \delta$, we have*

$$\triangle^s (\bar{\mathbf{x}}_T, \bar{\mathbf{y}}_T) \leq (1 + \eta) \Big( \frac{\mu t_0 (R_X^2 + R_Y^2)}{T} + \frac{L^2 \log(eT)}{\mu T} \Big) + C(1 + \eta)$$

$$\times \Big( \frac{L^2 (1 + \eta)}{n\mu\eta} + \frac{M}{n} + \frac{\beta + \mu}{\mu} \Big( \frac{4L}{n\mu} + 4\sqrt{\frac{1}{\mu}} \sqrt{\frac{\mu t_0 (R_X^2 + R_Y^2)}{T} + \frac{L^2 \log(eT)}{\mu T}} \Big) L \log_2 n \Big) \log \Big( \frac{1}{\delta} \Big).$$

*(d) If Assumptions 1 and 2 hold, then for any $\delta > 0$, with probability at least $1 - \delta$, we have*

$$\triangle^s (\bar{\mathbf{x}}_T, \bar{\mathbf{y}}_T) - \triangle_S^s (\bar{\mathbf{x}}_T, \bar{\mathbf{y}}_T) \leq \eta \Big( \frac{\mu t_0 (R_X^2 + R_Y^2)}{T} + \frac{L^2 \log(eT)}{\mu T} \Big) + C(1 + \eta)$$

$$\times \Big( \frac{L^2 (1 + \eta)}{n\mu\eta} + \frac{M}{n} + \frac{\beta + \mu}{\mu} \Big( \frac{4L}{n\mu} + 4\sqrt{\frac{1}{\mu}} \sqrt{\frac{\mu t_0 (R_X^2 + R_Y^2)}{T} + \frac{L^2 \log(eT)}{\mu T}} \Big) L \log_2 n \Big) \log \Big( \frac{1}{\delta} \Big).$$

*(e) If Assumptions 1 and 2 hold, then for any $\delta > 0$, with probability at least $1 - \delta$, we have*

$$R(\bar{\mathbf{x}}_T) \leq (1 + \eta) \inf_{\mathbf{x} \in \mathcal{X}} R(\mathbf{x}) + C \frac{2 + \eta}{\eta} \Big( \frac{\mu t_0 (R_X^2 + R_Y^2)}{T} + \frac{L^2 \log(eT)}{\mu T} \Big)$$

$$+ C \frac{2 + \eta}{\eta} \Big( \frac{M}{n} \log \frac{1}{\delta} + \Big( \frac{\beta}{\mu} + 1 \Big) L \Big( \frac{4L}{n\mu} + 4\sqrt{\frac{1}{\mu}} \sqrt{\frac{\mu t_0 (R_X^2 + R_Y^2)}{T} + \frac{L^2 \log(eT)}{\mu T}} \Big) \log_2 n \log \frac{1}{\delta} \Big).$$

**Remark 12.** When conditions in Theorem 4 hold, we obtain that (a) If Assumption 1 holds and $F_S(\bar{\mathbf{x}}_T, \bar{\mathbf{y}}_T) = \mathcal{O}\big(\frac{1}{n}\big)$, then for any $\delta > 0$, with probability at least $1 - \delta$, the plain generalization error of $(\bar{\mathbf{x}}_T, \bar{\mathbf{y}}_T)$ of GDA is of the order $\mathcal{O}\Big( \big( \frac{1}{n} + \sqrt{\frac{\log T}{T}} \big) \log_2 n \log(1/\delta) \Big)$. (b) If Assumptions 1 and 2 hold and $R_S(\bar{\mathbf{x}}_T) = \mathcal{O}\big(\frac{1}{n}\big)$, then for any $\delta > 0$, with probability at least $1 - \delta$, the primal generalization error of $(\bar{\mathbf{x}}_T, \bar{\mathbf{y}}_T)$ of GDA is of the order $\mathcal{O}\Big( \big( \frac{1}{n} + \sqrt{\frac{\log T}{T}} \big) \log_2 n \log(1/\delta) \Big)$. (c) If Assumptions 1 and 2 hold, then for any $\delta > 0$, with probability at least $1 - \delta$, the strong PD population risk and the strong PD generalization error of $(\bar{\mathbf{x}}_T, \bar{\mathbf{y}}_T)$ of GDA are all of the order $\mathcal{O}\Big( \big( \frac{1}{n} + \sqrt{\frac{\log T}{T}} \big) \log_2 n \log(1/\delta) \Big)$. (d) If Assumptions 1 and 2 hold and $\inf_{\mathbf{x} \in \mathcal{X}} R(\mathbf{x}) = \mathcal{O}\big(\frac{1}{n}\big)$, then for any $\delta > 0$, with probability at least $1 - \delta$, the excess primal population risk of $(\bar{\mathbf{x}}_T, \bar{\mathbf{y}}_T)$ of GDA is of the order $\mathcal{O}\Big( \big( \frac{1}{n} + \sqrt{\frac{\log T}{T}} \big) \log_2 n \log(1/\delta) \Big)$. For the above bounds, we can take $T = \mathcal{O}(n^2)$ gradient evaluations to get bound of the order $\mathcal{O}\Big( \frac{\log^{3/2} n}{n} \log(1/\delta) \Big)$.

## D  STOCHASTIC GRADIENT DESCENT ASCENT

We need some notations to state results on SGDA. Specifically, assume the initial point satisfies $\mathbf{x}_1 = 0$ and $\mathbf{y}_1 = 0$. Let $\{\eta_t\}$ be a sequence of positive step sizes. At the $t$-th iteration, SGDA

first randomly select an index $i_t$ form the uniform distribution over $[n] := \{1, ..., n\}$ and then do the update

$$
\begin{aligned}
\mathbf{x}_{t+1} &= \mathbf{x}_t - \eta_t \nabla_{\mathbf{x}} f(\mathbf{x}_t, \mathbf{y}_t, z_{i_t}), \\
\mathbf{y}_{t+1} &= \mathbf{y}_t + \eta_t \nabla_{\mathbf{y}} f(\mathbf{x}_t, \mathbf{y}_t, z_{i_t}).
\end{aligned}
\tag{38}
$$

We denote the average of iterates by

$$
\bar{\mathbf{x}}_T = \frac{\sum_{t=1}^T \mathbf{x}_t}{T} \quad \text{and} \quad \bar{\mathbf{y}}_T = \frac{\sum_{t=1}^T \mathbf{y}_t}{T}.
\tag{39}
$$

Let's first introduce two concentration inequalities for martingales, which are required in deriving the strong PD empirical risk bound of SGDA.

**Lemma 9.** *(Boucheron et al., 2013) Let $z_1, ..., z_n$ be a sequence of random variables such that $z_k$ may depend the previous variables $z_1, ..., z_{k-1}$ for all $k = 1, ..., n$. Consider a sequence of functionals $\xi_k(z_1, ..., z_k)$, $k = 1, ..., n$. Assume $|\xi_k - \mathbb{E}_{z_k}[\xi_k]| \le b_k$ for each $k$. Let $\delta \in (0, 1)$. With probability at least $1 - \delta$*

$$
\sum_{k=1}^n \xi_k - \sum_{k=1}^n \mathbb{E}_{z_k}[\xi_k] \le \left( 2 \sum_{k=1}^n b_k^2 \log \frac{1}{\delta} \right)^{\frac{1}{2}}.
$$

**Lemma 10.** *(Tarres & Yao, 2014) Let $\{\xi_k\}_{k \in \mathbb{N}}$ be a martingale difference sequence in $\mathbb{R}^d$. Suppose that almost surely $\|\xi_k\| \le D$ and $\sum_{k=1}^t \mathbb{E}[\|\xi_k\|^2 | \xi_1, ..., \xi_{k-1}] \le \sigma_t^2$. Then, for any $0 < \delta < 1$, the following inequality holds with probability at least $1 - \delta$*

$$
\max_{1 \le j \le t} \left\| \sum_{k=1}^j \xi_k \right\| \le 2 \left( \frac{D}{3} + \sigma_t \right) \log \frac{2}{\delta}.
$$

The following lemma shows the strong PD empirical risk bound of SGDA.

**Lemma 11.** *Suppose Assumption 1 holds and $F_S(\cdot, \cdot)$ be $\mu$-SC-SC with $\mu > 0$. Let $\{\mathbf{x}_t, \mathbf{y}_t\}$ be the sequence produced by (38) with $\eta_t = \frac{1}{\mu(t+t_0)}$. Assume $t_0 \ge 0$. Suppose $\sup_{\mathbf{x} \in \mathcal{X}} \|\mathbf{x}\| \le R_X$ and $\sup_{\mathbf{y} \in \mathcal{Y}} \|\mathbf{y}\| \le R_Y$. Let $\delta > 0$. Then for $(\bar{\mathbf{x}}_T, \bar{\mathbf{y}}_T)$ in (39), with probability at least $1 - \delta$ we have*

$$
\sup_{\mathbf{y} \in \mathcal{Y}} F_S(\bar{\mathbf{x}}_T, \mathbf{y}) - \inf_{\mathbf{x} \in \mathcal{X}} F_S(\mathbf{x}, \bar{\mathbf{y}}_T) \le \frac{2\mu t_0 (R_X^2 + R_Y^2)}{T} + \frac{L^2 \log(eT)}{\mu T}
$$
$$
+ \frac{2(R_X + R_Y)}{T} \left( \frac{2L}{3} + 2L\sqrt{T} \right) \log \frac{6}{\delta} + \frac{2L(R_X + R_Y)(2T \log(6/\delta))^{\frac{1}{2}}}{T}.
$$

*If $t_0 = 0$, then with probability at least $1 - \delta$ we have*

$$
\sup_{\mathbf{y} \in \mathcal{Y}} F_S(\bar{\mathbf{x}}_T, \mathbf{y}) - \inf_{\mathbf{x} \in \mathcal{X}} F_S(\mathbf{x}, \bar{\mathbf{y}}_T) \le \frac{L^2 \log(eT)}{\mu T} + \frac{2(R_X + R_Y)}{T} \left( \frac{2L}{3} + 2L\sqrt{T} \right) \log \frac{6}{\delta}
$$
$$
+ \frac{2L(R_X + R_Y)(2T \log(6/\delta))^{\frac{1}{2}}}{T}.
$$

*Proof.* This proof follows from (Lei et al., 2021). Firstly, we have

$$
\|\mathbf{x}_{t+1} - \mathbf{x}\|^2 = \|\mathbf{x}_t - \eta_t \nabla_{\mathbf{x}} f(\mathbf{x}_t, \mathbf{y}_t; z_{i_t}) - \mathbf{x}\|^2
$$
$$
= \|\mathbf{x}_t - \mathbf{x}\|^2 + \eta_t^2 \|\nabla_{\mathbf{x}} f(\mathbf{x}_t, \mathbf{y}_t; z_{i_t})\|^2 + 2\eta_t \langle \mathbf{x} - \mathbf{x}_t, \nabla_{\mathbf{x}} f(\mathbf{x}_t, \mathbf{y}_t; z_{i_t}) \rangle
$$
$$
\le \|\mathbf{x}_t - \mathbf{x}\|^2 + \eta_t^2 L^2 + 2\eta_t \langle \mathbf{x} - \mathbf{x}_t, \nabla_{\mathbf{x}} f(\mathbf{x}_t, \mathbf{y}_t; z_{i_t}) - \nabla_{\mathbf{x}} F_S(\mathbf{x}_t, \mathbf{y}_t) \rangle + 2\eta_t \langle \mathbf{x} - \mathbf{x}_t, \nabla_{\mathbf{x}} F_S(\mathbf{x}_t, \mathbf{y}_t) \rangle,
$$

where the first inequality holds because of Assumption 1. By the strong convexity of $F_S(\cdot, \mathbf{y}_t)$, we have

$$
2\eta_t (F_S(\mathbf{x}_t, \mathbf{y}_t) - F_S(\mathbf{x}, \mathbf{y}_t)) \le (1 - \eta_t \mu) \|\mathbf{x}_t - \mathbf{x}\|^2 - \|\mathbf{x}_{t+1} - \mathbf{x}\|^2 + \eta_t^2 L^2
$$
$$
+ 2\eta_t \langle \mathbf{x} - \mathbf{x}_t, \nabla_{\mathbf{x}} f(\mathbf{x}_t, \mathbf{y}_t; z_{i_t}) - \nabla_{\mathbf{x}} F_S(\mathbf{x}_t, \mathbf{y}_t) \rangle.
$$

Since $\eta_t = \frac{1}{\mu(t+t_0)}$, we further get

$$\frac{2}{\mu(t+t_0)}(F_S(\mathbf{x}_t, \mathbf{y}_t) - F_S(\mathbf{x}, \mathbf{y}_t)) \leq \left(1 - \frac{1}{(t+t_0)}\right)\|\mathbf{x}_t - \mathbf{x}\|^2 - \|\mathbf{x}_{t+1} - \mathbf{x}\|^2$$
$$+ \left(\frac{1}{\mu(t+t_0)}\right)^2 L^2 + \frac{2}{\mu(t+t_0)}\langle \mathbf{x} - \mathbf{x}_t, \nabla_{\mathbf{x}} f(\mathbf{x}_t, \mathbf{y}_t; z_{i_t}) - \nabla_{\mathbf{x}} F_S(\mathbf{x}_t, \mathbf{y}_t)\rangle.$$

Multiplying both sides by $t + t_0$, we have

$$\frac{2}{\mu}(F_S(\mathbf{x}_t, \mathbf{y}_t) - F_S(\mathbf{x}, \mathbf{y}_t)) \leq (t + t_0 - 1)\|\mathbf{x}_t - \mathbf{x}\|^2 - (t + t_0)\|\mathbf{x}_{t+1} - \mathbf{x}\|^2 + \frac{L^2}{\mu^2(t+t_0)}$$
$$+ \frac{2}{\mu}\langle \mathbf{x} - \mathbf{x}_t, \nabla_{\mathbf{x}} f(\mathbf{x}_t, \mathbf{y}_t; z_{i_t}) - \nabla_{\mathbf{x}} F_S(\mathbf{x}_t, \mathbf{y}_t)\rangle.$$

Since $\mathbf{x}_1 = 0$ and $\sum_{t=1}^{T} t^{-1} \leq \log(eT)$, by taking a summation of the above inequality from $t = 1$ to $T$, we obtain

$$\sum_{t=1}^{T}(F_S(\mathbf{x}_t, \mathbf{y}_t) - F_S(\mathbf{x}, \mathbf{y}_t)) \leq \frac{\mu}{2}t_0 R_X^2 + \frac{L^2 \log(eT)}{2\mu}$$
$$+ \sum_{t=1}^{T}\langle \mathbf{x}, \nabla_{\mathbf{x}} f(\mathbf{x}_t, \mathbf{y}_t; z_{i_t}) - \nabla_{\mathbf{x}} F_S(\mathbf{x}_t, \mathbf{y}_t)\rangle + \sum_{t=1}^{T}\langle \mathbf{x}_t, \nabla_{\mathbf{x}} F_S(\mathbf{x}_t, \mathbf{y}_t) - \nabla_{\mathbf{x}} f(\mathbf{x}_t, \mathbf{y}_t; z_{i_t})\rangle.$$

From the concavity of $F_S(\mathbf{x}, \cdot)$ we get

$$\sum_{t=1}^{T}(F_S(\mathbf{x}_t, \mathbf{y}_t) - F_S(\mathbf{x}, \bar{\mathbf{y}}_T)) \leq \frac{\mu}{2}t_0 R_X^2 + \frac{L^2 \log(eT)}{2\mu}$$
$$+ \sum_{t=1}^{T}\langle \mathbf{x}, \nabla_{\mathbf{x}} f(\mathbf{x}_t, \mathbf{y}_t; z_{i_t}) - \nabla_{\mathbf{x}} F_S(\mathbf{x}_t, \mathbf{y}_t)\rangle + \sum_{t=1}^{T}\langle \mathbf{x}_t, \nabla_{\mathbf{x}} F_S(\mathbf{x}_t, \mathbf{y}_t) - \nabla_{\mathbf{x}} f(\mathbf{x}_t, \mathbf{y}_t; z_{i_t})\rangle.$$

Since this inequality holds for any $\mathbf{x}$, we get

$$\sum_{t=1}^{T}(F_S(\mathbf{x}_t, \mathbf{y}_t) - \inf_{\mathbf{x} \in \mathcal{X}} F_S(\mathbf{x}, \bar{\mathbf{y}}_T)) \leq \frac{\mu}{2}t_0 R_X^2 + \frac{L^2 \log(eT)}{2\mu}$$
$$+ \sum_{t=1}^{T}\sup_{\mathbf{x} \in \mathcal{X}}\langle \mathbf{x}, \nabla_{\mathbf{x}} f(\mathbf{x}_t, \mathbf{y}_t; z_{i_t}) - \nabla_{\mathbf{x}} F_S(\mathbf{x}_t, \mathbf{y}_t)\rangle + \sum_{t=1}^{T}\langle \mathbf{x}_t, \nabla_{\mathbf{x}} F_S(\mathbf{x}_t, \mathbf{y}_t) - \nabla_{\mathbf{x}} f(\mathbf{x}_t, \mathbf{y}_t; z_{i_t})\rangle.$$

By Schwarz's inequality, we have

$$\sum_{t=1}^{T}(F_S(\mathbf{x}_t, \mathbf{y}_t) - \inf_{\mathbf{x} \in \mathcal{X}} F_S(\mathbf{x}, \bar{\mathbf{y}}_T)) \leq \frac{\mu}{2}t_0 R_X^2 + \frac{L^2 \log(eT)}{2\mu}$$
$$+ R_X \left\|\sum_{t=1}^{T} \nabla_{\mathbf{x}} f(\mathbf{x}_t, \mathbf{y}_t; z_{i_t}) - \nabla_{\mathbf{x}} F_S(\mathbf{x}_t, \mathbf{y}_t)\right\| + \sum_{t=1}^{T}\langle \mathbf{x}_t, \nabla_{\mathbf{x}} F_S(\mathbf{x}_t, \mathbf{y}_t) - \nabla_{\mathbf{x}} f(\mathbf{x}_t, \mathbf{y}_t; z_{i_t})\rangle.$$

Denote $\xi_t = \langle \mathbf{x}_t, \nabla_{\mathbf{x}} F_S(\mathbf{x}_t, \mathbf{y}_t) - \nabla_{\mathbf{x}} f(\mathbf{x}_t, \mathbf{y}_t; z_{i_t})\rangle$. Since $\mathbb{E}_{i_t}[\langle \mathbf{x}_t, \nabla_{\mathbf{x}} F_S(\mathbf{x}_t, \mathbf{y}_t) - \nabla_{\mathbf{x}} f(\mathbf{x}_t, \mathbf{y}_t; z_{i_t})\rangle] = 0$, so $\{\xi_t | t = 1, ..., T\}$ is a martingale difference sequence. By Schwarz's inequality and Assumption 1, we know that $|\langle \mathbf{x}_t, \nabla_{\mathbf{x}} F_S(\mathbf{x}_t, \mathbf{y}_t) - \nabla_{\mathbf{x}} f(\mathbf{x}_t, \mathbf{y}_t; z_{i_t})\rangle| \leq 2LR_X$. Then, according to Lemma 9, we have the following inequality with probability at least $1 - \delta/6$

$$\sum_{t=1}^{T}\langle \mathbf{x}_t, \nabla_{\mathbf{x}} F_S(\mathbf{x}_t, \mathbf{y}_t) - \nabla_{\mathbf{x}} f(\mathbf{x}_t, \mathbf{y}_t; z_{i_t})\rangle \leq 2LR_X(2T\log(6/\delta))^{\frac{1}{2}}.$$

Define $\xi_t' = \nabla_{\mathbf{x}} f(\mathbf{x}_t, \mathbf{y}_t; z_{i_t}) - \nabla_{\mathbf{x}} F_S(\mathbf{x}_t, \mathbf{y}_t)$. Then we get $\|\xi_t'\| \leq 2L$ and

$$\sum_{t=1}^{T} \mathbb{E}[\|\xi_t'\|^2 | \xi_1', ..., \xi_{t-1}'] \leq 4TL^2.$$

Applying Lemma 10 to the martingale difference sequence $\{\xi'_t\}$, we have the following inequality with probability at least $1 - \delta/3$

$$\Big\|\sum_{t=1}^{T}\xi'_t\Big\| \leq 2\Big(\frac{2L}{3} + 2L\sqrt{T}\Big)\log\frac{6}{\delta}.$$

That is, with probability at least $1 - \delta/3$

$$\Big\|\sum_{t=1}^{T}\nabla_{\mathbf{x}}f(\mathbf{x}_t, \mathbf{y}_t; z_{i_t}) - \nabla_{\mathbf{x}}F_S(\mathbf{x}_t, \mathbf{y}_t)\Big\| \leq 2\Big(\frac{2L}{3} + 2L\sqrt{T}\Big)\log\frac{6}{\delta}.$$

Combined with the above results, we finally have the following inequality with probability at least $1 - \delta/2$

$$\frac{1}{T}\sum_{t=1}^{T}(F_S(\mathbf{x}_t, \mathbf{y}_t) - \inf_{\mathbf{x}\in\mathcal{X}}F_S(\mathbf{x}, \bar{\mathbf{y}}_T)) \leq \frac{\mu t_0 R_X^2}{2T} + \frac{L^2\log(eT)}{2\mu T}$$

$$+\frac{2R_X}{T}\Big(\frac{2L}{3} + 2L\sqrt{T}\Big)\log\frac{6}{\delta} + \frac{2LR_X(2T\log(6/\delta))^{\frac{1}{2}}}{T}.$$

In a similar way, we have the following inequality with probability at least $1 - \delta/2$

$$\sup_{\mathbf{y}\in\mathcal{Y}}F_S(\bar{\mathbf{x}}_T, \mathbf{y}) - \frac{1}{T}\sum_{t=1}^{T}(F_S(\mathbf{x}_t, \mathbf{y}_t)) \leq \frac{\mu t_0 R_Y^2}{2T} + \frac{L^2\log(eT)}{2\mu T}$$

$$+\frac{2R_Y}{T}\Big(\frac{2L}{3} + 2L\sqrt{T}\Big)\log\frac{6}{\delta} + \frac{2LR_Y(2T\log(6/\delta))^{\frac{1}{2}}}{T}.$$

Combined the above two inequalities together we get the result with probability at least $1 - \delta$

$$\sup_{\mathbf{y}\in\mathcal{Y}}F_S(\bar{\mathbf{x}}_T, \mathbf{y}) - \inf_{\mathbf{x}\in\mathcal{X}}F_S(\mathbf{x}, \bar{\mathbf{y}}_T) \leq \frac{\mu t_0(R_X^2 + R_Y^2)}{T} + \frac{L^2\log(eT)}{\mu T}$$

$$+\frac{2(R_X + R_Y)}{T}\Big(\frac{2L}{3} + 2L\sqrt{T}\Big)\log\frac{6}{\delta} + \frac{2L(R_X + R_Y)(2T\log(6/\delta))^{\frac{1}{2}}}{T}.$$

$\square$

Denote $E = \frac{\mu t_0(R_X^2 + R_Y^2)}{T} + \frac{L^2\log(eT)}{\mu T} + \frac{2(R_X + R_Y)(\frac{2L}{3} + 2L\sqrt{T})\log\frac{6}{\delta}}{T} + \frac{2L(R_X + R_Y)(2T\log(6/\delta))^{\frac{1}{2}}}{T}$.
Now, plugging Lemma 11 to Lemma 7, we know that the argument stability bound of SGDA is

$$\|\bar{\mathbf{x}}_T^i - \bar{\mathbf{x}}_T\| + \|\bar{\mathbf{y}}_T^i - \bar{\mathbf{y}}_T\| \leq \frac{4L}{n\mu} + 4\sqrt{\frac{1}{\mu}}\sqrt{\triangle_S^s(\bar{\mathbf{x}}_T, \bar{\mathbf{y}}_T)} \leq 4\sqrt{\frac{1}{\mu}}\sqrt{E} + \frac{4L}{n\mu}. \tag{40}$$

Furthermore, for any $\mathbf{x} \in \mathcal{X}, \mathbf{y} \in \mathcal{Y}$ and $z \in \mathcal{Z}$,

$$f(\mathbf{x}, \mathbf{y}; z) \leq f(0, 0; z) + L\|\mathbf{x} - 0\| + L\|\mathbf{y} - 0\| \leq \sup_{z\in\mathcal{Z}}f(0, 0; z) + L(R_X + R_Y), \tag{41}$$

Note that since SGDA is a randomized algorithm, thus we need the following variant of Theorem 1.

**Theorem 5.** *Let $A$ be a randomized learning algorithm and $\epsilon > 0$. Suppose $|f(\mathbf{x}, \mathbf{y}; z)| \leq M$ for some $M > 0$ and $\mathbf{x} \in \mathcal{X}, \mathbf{y} \in \mathcal{Y}, z \in \mathcal{Z}$. Fixed any $\eta > 0$. There exists an absolute positive constant $C$.*

*(a.) If $A$ has $\epsilon$-uniform stability with probability at least $1 - \delta'$ for some $\delta' \in (0, 1)$ over the randomness of $A$, i.e.,*

$$Pr_A\Big(\sup_z[f(A_{\mathbf{x}}(S), A_{\mathbf{y}}(S); z) - f(A_{\mathbf{x}}(S'), A_{\mathbf{y}}(S'); z)]\Big) \leq \epsilon,$$

*And if the randomness of $A$ is independent of the training set $S$. Then for any $\delta > 0$, with probability at least $1 - \delta' - \delta$,*

$$F(A_{\mathbf{x}}(S), A_{\mathbf{y}}(S)) \leq (1 + \eta)F_S(A_{\mathbf{x}}(S), A_{\mathbf{y}}(S)) + C\frac{1+\eta}{\eta}\Big(\frac{M}{n}\log(1/\delta) + \epsilon\log_2 n\log\frac{1}{\delta}\Big).$$

*(b.) Assume that for all $\mathbf{x}$, the function $\mathbf{y} \mapsto F(\mathbf{x}, \mathbf{y})$ is $\mu$-strongly-concave. Suppose Assumptions 1 and 2 hold. If the algorithm $A$ is $\epsilon$-argument stable with probability at least $1 - \delta'$ for some $\delta' \in (0, 1)$ over the randomness of $A$, i.e.,*

$$Pr_A \Big( \|A_{\mathbf{x}}(S) - A_{\mathbf{x}}(S')\| + \|A_{\mathbf{y}}(S) - A_{\mathbf{y}}(S')\| \Big) \leq \epsilon.$$

*And if the randomness of $A$ is independent of the training set $S$. Then for any $\delta > 0$, with probability at least $1 - \delta' - \delta$,*

$$R(A_{\mathbf{x}}(S)) \leq (1 + \eta) R_S(A_{\mathbf{x}}(S)) + C \frac{1 + \eta}{\eta} \Big( \frac{M}{n} \log \frac{1}{\delta} + \Big( \frac{\beta}{\mu} + 1 \Big) L\epsilon \log_2 n \log \frac{1}{\delta} \Big).$$

*(c.) Assume that for all $\mathbf{x}$ and $\mathbf{y}$, the function $F(\mathbf{x}, \mathbf{y})$ is $\mu$-SC-SC. Suppose Assumptions 1 and 2 hold. If the algorithm $A$ is $\epsilon$-argument stable with probability at least $1 - \delta'$ for some $\delta' \in (0, 1)$ over the randomness of $A$, i.e.,*

$$Pr_A \Big( \|A_{\mathbf{x}}(S) - A_{\mathbf{x}}(S')\| + \|A_{\mathbf{y}}(S) - A_{\mathbf{y}}(S')\| \Big) \leq \epsilon.$$

*And if the randomness of $A$ is independent of the training set $S$. Then for any $\delta > 0$, with probability at least $1 - \delta' - \delta$,*

$$\triangle^s (A_{\mathbf{x}}(S), A_{\mathbf{y}}(S)) \leq \triangle_S^s (A_{\mathbf{x}}(S), A_{\mathbf{y}}(S)) + \eta \mathbb{E}_S \triangle_S^s (A_{\mathbf{x}}(S), A_{\mathbf{y}}(S))$$
$$+ C(1 + \eta) \Big( \frac{L^2 (1 + \eta)}{n \mu \eta} + \frac{M}{n} + \Big( 1 + \frac{\beta}{\mu} \Big) \epsilon L \log_2 n \Big) \log \Big( \frac{1}{\delta} \Big).$$

*(d.) Assume that for all $\mathbf{x}$ and $\mathbf{y}$, the function $F(\mathbf{x}, \mathbf{y})$ is $\mu$-SC-SC. Suppose Assumptions 1 and 2 hold. If the algorithm $A$ is $\epsilon$-argument stable with probability at least $1 - \delta'$ for some $\delta' \in (0, 1)$ over the randomness of $A$, i.e.,*

$$Pr_A \Big( \|A_{\mathbf{x}}(S) - A_{\mathbf{x}}(S')\| + \|A_{\mathbf{y}}(S) - A_{\mathbf{y}}(S')\| \Big) \leq \epsilon.$$

*And if the randomness of $A$ is independent of the training set $S$. Then for any $\delta > 0$, with probability at least $1 - \delta' - \delta$,*

$$\triangle^s (A_{\mathbf{x}}(S), A_{\mathbf{y}}(S)) - \triangle_S^s (A_{\mathbf{x}}(S), A_{\mathbf{y}}(S)) \leq \eta \mathbb{E}_S \triangle_S^s (A_{\mathbf{x}}(S), A_{\mathbf{y}}(S))$$
$$+ C(1 + \eta) \Big( \frac{L^2 (1 + \eta)}{n \mu \eta} + \frac{M}{n} + \Big( 1 + \frac{\beta}{\mu} \Big) \epsilon L \log_2 n \Big) \log \Big( \frac{1}{\delta} \Big).$$

*(e.) Assume that for all $\mathbf{x}$, the function $\mathbf{y} \mapsto F(\mathbf{x}, \mathbf{y})$ is $\mu$-strongly-concave. Suppose Assumptions 1 and 2 hold. If the algorithm $A$ is $\epsilon$-argument stable with probability at least $1 - \delta'$ for some $\delta' \in (0, 1)$ over the randomness of $A$, i.e.,*

$$Pr_A \Big( \|A_{\mathbf{x}}(S) - A_{\mathbf{x}}(S')\| + \|A_{\mathbf{y}}(S) - A_{\mathbf{y}}(S')\| \Big) \leq \epsilon.$$

*And if the randomness of $A$ is independent of the training set $S$. Then for any $\delta > 0$, with probability at least $1 - \delta' - \delta$,*

$$R(A_{\mathbf{x}}(S)) \leq (1 + \eta) \inf_{\mathbf{x} \in \mathcal{X}} R(\mathbf{x})$$
$$+ C \frac{2 + \eta}{\eta} \Big( \frac{M}{n} \log \frac{1}{\delta} + \Big( \frac{\beta}{\mu} + 1 \Big) L\epsilon \log_2 n \log \frac{1}{\delta} + \triangle_S^s (A_{\mathbf{x}}(S), A_{\mathbf{y}}(S)) \Big).$$

Therefore, plugging (41), the stability bound in (40) and the strong PD empirical risk bound in Lemma 11 into Theorem 5, we obtain generalization bounds of SGDA. Now, we write the main theorem of SGDA as follows.

**Theorem 6.** *Assume for all $z$, the function $(\mathbf{x}, \mathbf{y}) \mapsto f(\mathbf{x}, \mathbf{y}; z)$ is $\mu$-SC-SC. Suppose $\sup_{\mathbf{x} \in \mathcal{X}} \|\mathbf{x}\| \leq R_X$ and $\sup_{\mathbf{y} \in \mathcal{Y}} \|\mathbf{y}\| \leq R_Y$. Let $\{\mathbf{x}_t, \mathbf{y}_t\}$ be produced by (38) with $\eta_t = 1/(\mu(t + t_0))$. Assume $t_0 \geq 0$. Denote $A_{\mathbf{x}}(S) = \bar{\mathbf{x}}_T$ and $A_{\mathbf{y}}(S) = \bar{\mathbf{y}}_T$ for $(\bar{\mathbf{x}}_T, \bar{\mathbf{y}}_T)$ in (39). Fixed any $\eta > 0$. Let*

$M = \sup_{z \in \mathcal{Z}} f(0, 0; z) + L(R_X + R_Y)$. Let $E = \frac{\mu t_0 (R_X^2 + R_Y^2)}{T} + \frac{L^2 \log(eT)}{\mu T} + \frac{2(R_X + R_Y)}{T} \left( \frac{2L}{3} + 2L\sqrt{T} \right) \log \frac{6}{\delta} + \frac{2L(R_X + R_Y)(2T \log(6/\delta))^{\frac{1}{2}}}{T}$ and $B = \frac{4L}{n\mu} + 4\sqrt{\frac{1}{\mu}}\sqrt{E}$. *There exists an absolute positive constant $C$.*

*(a) If Assumption 1 holds, then for any $\delta > 0$, with probability at least $1 - 2\delta$, we have*

$$F(\bar{\mathbf{x}}_T, \bar{\mathbf{y}}_T) \leq (1 + \eta) F_S(\bar{\mathbf{x}}_T, \bar{\mathbf{y}}_T) + C \frac{1 + \eta}{\eta} \left( \frac{M}{n} \log(1/\delta) + B \log_2 n \log(1/\delta) \right).$$

*(b) If Assumptions 1 and 2 hold, then for any $\delta > 0$, with probability at least $1 - 2\delta$, we have*

$$R(\bar{\mathbf{x}}_T) \leq (1 + \eta) R_S(\bar{\mathbf{x}}_T) + C \frac{1 + \eta}{\eta} \left( \frac{M}{n} \log \frac{1}{\delta} + \left( \frac{\beta}{\mu} + 1 \right) LB \log_2 n \log \frac{1}{\delta} \right).$$

*(c) If Assumptions 1 and 2 hold, then for any $\delta > 0$, with probability at least $1 - 2\delta$, we have*

$$\triangle^s(\bar{\mathbf{x}}_T, \bar{\mathbf{y}}_T) \leq (1 + \eta) E + C(1 + \eta) \left( \frac{L^2 (1 + \eta)}{n\mu\eta} + \frac{M}{n} + \left( 1 + \frac{\beta}{\mu} \right) BL \log_2 n \right) \log \left( \frac{1}{\delta} \right).$$

*(d) If Assumptions 1 and 2 hold, then for any $\delta > 0$, with probability at least $1 - 2\delta$, we have*

$$\triangle^s(\bar{\mathbf{x}}_T, \bar{\mathbf{y}}_T) - \triangle^s_S(\bar{\mathbf{x}}_T, \bar{\mathbf{y}}_T)$$
$$\leq \eta E + C(1 + \eta) \left( \frac{L^2 (1 + \eta)}{n\mu\eta} + \frac{M}{n} + \left( 1 + \frac{\beta}{\mu} \right) BL \log_2 n \right) \log \left( \frac{1}{\delta} \right).$$

*(e) If Assumptions 1 and 2 hold, then for any $\delta > 0$, with probability at least $1 - 2\delta$, we have*

$$R(\bar{\mathbf{x}}_T) \leq (1 + \eta) \inf_{\mathbf{x} \in \mathcal{X}} R(\mathbf{x}) + C \frac{2 + \eta}{\eta} \left( \frac{M}{n} \log \frac{1}{\delta} + \left( \frac{\beta}{\mu} + 1 \right) LB \log_2 n \log \frac{1}{\delta} + E \right).$$

**Remark 13.** When conditions in Theorem 6 hold, we obtain that (a) If Assumption 1 holds and $F_S(\bar{\mathbf{x}}_T, \bar{\mathbf{y}}_T) = \mathcal{O}\left( \frac{1}{n} \right)$, then for any $\delta > 0$, with probability at least $1 - \delta$, the plain generalization error of $(\bar{\mathbf{x}}_T, \bar{\mathbf{y}}_T)$ of SGDA is of the order $\mathcal{O}\left( \left( \frac{1}{n} + \sqrt{\frac{\log(1/\delta)}{T^{\frac{1}{2}}}} \right) \log_2 n \log(1/\delta) \right)$. (b) If Assumptions 1 and 2 hold and $R_S(\bar{\mathbf{x}}_T) = \mathcal{O}\left( \frac{1}{n} \right)$, then for any $\delta > 0$, with probability at least $1 - \delta$, the primal generalization error of $(\bar{\mathbf{x}}_T, \bar{\mathbf{y}}_T)$ of SGDA is of the order $\mathcal{O}\left( \left( \frac{1}{n} + \sqrt{\frac{\log(1/\delta)}{T^{\frac{1}{2}}}} \right) \log_2 n \log(1/\delta) \right)$. (c) If Assumptions 1 and 2 hold, then for any $\delta > 0$, with probability at least $1 - \delta$, the strong PD population risk and strong PD generalization error of $(\bar{\mathbf{x}}_T, \bar{\mathbf{y}}_T)$ of SGDA are all of the order $\mathcal{O}\left( \left( \frac{1}{n} + \sqrt{\frac{\log(1/\delta)}{T^{\frac{1}{2}}}} \right) \log_2 n \log(1/\delta) \right)$. (d) If Assumptions 1 and 2 hold and $\inf_{\mathbf{x} \in \mathcal{X}} R(\mathbf{x}) = \mathcal{O}\left( \frac{1}{n} \right)$, then for any $\delta > 0$, with probability at least $1 - \delta$, the excess primal population risk of $(\bar{\mathbf{x}}_T, \bar{\mathbf{y}}_T)$ of SGDA is of the order $\mathcal{O}\left( \left( \frac{1}{n} + \sqrt{\frac{\log(1/\delta)}{T^{\frac{1}{2}}}} \right) \log_2 n \log(1/\delta) \right)$. For the above bounds, we can take $T = \mathcal{O}(n^4)$ stochastic gradient evaluations to get bound of the order $\mathcal{O}\left( \frac{\log n}{n} \log^{\frac{3}{2}}(1/\delta) \right)$.

# E PROXIMAL POINT METHOD

One of the classical algorithms studied for solving the minimax problem is the Proximal Point method (Rockafellar, 1976). We denote the $t$-th iterate of PPM as $(\mathbf{x}_t, \mathbf{y}_t)$. The averaged iterate is defined as

$$\bar{\mathbf{x}}_T = \frac{1}{T} \sum_{t=1}^{T} \mathbf{x}_t \quad \text{and} \quad \bar{\mathbf{y}}_T = \frac{1}{T} \sum_{t=1}^{T} \mathbf{y}_t. \tag{42}$$

Given stepsize parameter $\nu$, the PPM generates the iterate $\{\mathbf{x}_{t+1}, \mathbf{y}_{t+1}\}$ by

$$\arg\min_{\mathbf{x}\in\mathcal{X}} \arg\max_{\mathbf{y}\in\mathcal{Y}} \left\{ F_S(\mathbf{x}, \mathbf{y}) + \frac{1}{2\nu}\|\mathbf{x} - \mathbf{x}_t\| - \frac{1}{2\nu}\|\mathbf{y} - \mathbf{y}_t\| \right\}. \tag{43}$$

$\{\mathbf{x}_{t+1}, \mathbf{y}_{t+1}\}$ is the unique solution since the objective function of problem (43) is strongly convex in $\mathbf{x}$ and strongly concave in $\mathbf{y}$. From the discussion in (Mokhtari et al., 2019), the update of PPM can be written as

$$\begin{aligned}
\mathbf{x}_{t+1} &= \mathbf{x}_t - \nu\nabla_{\mathbf{x}}F_S(\mathbf{x}_{t+1}, \mathbf{y}_{t+1}), \\
\mathbf{y}_{t+1} &= \mathbf{y}_t + \nu\nabla_{\mathbf{y}}F_S(\mathbf{x}_{t+1}, \mathbf{y}_{t+1}),
\end{aligned} \tag{44}$$

Assume that the initial point satisfies $\mathbf{x}_0 = 0$ and $\mathbf{y}_0 = 0$.

We now begin to prove the strong PD empirical risk. Firstly, two lemmas are introduced.

**Lemma 12.** *(Nemirovski, 2005) Define vector $\mathbf{v} = [\mathbf{x}, \mathbf{y}] \in \mathbb{R}^{2d}$ and the operator $P : \mathbb{R}^{2d} \mapsto \mathbb{R}^{2d}$ as*

$$P(\mathbf{v}) = [\nabla_{\mathbf{x}}F_S(\mathbf{x}, \mathbf{y}); -\nabla_{\mathbf{y}}F_S(\mathbf{x}, \mathbf{y})]. \tag{45}$$

*Consider $(\bar{\mathbf{x}}_T, \bar{\mathbf{y}}_T)$ in (42). Suppose the ESP solution exists. Assume the function $F_S(\mathbf{x}, \mathbf{y})$ is continuously differentiable in $\mathbf{x}$ and $\mathbf{y}$. Assume that $F_S(\mathbf{x}, \mathbf{y})$ is a convex function of $\mathbf{x}$ for any $\mathbf{y}$ and is a concave function of $\mathbf{y}$ for any $\mathbf{x}$. Then for any $\mathbf{v} = [\mathbf{x}, \mathbf{y}] \in \mathbb{R}^{2d}$, we have*

$$F_S(\bar{\mathbf{x}}_T, \mathbf{y}) - F_S(\mathbf{x}, \bar{\mathbf{y}}_T) \le \frac{1}{T}\sum_{t=1}^{T} P(\mathbf{v}_t)^T(\mathbf{v}_t - \mathbf{v}).$$

**Lemma 13.** *(Mokhtari et al., 2019) Consider the sequence of iterates $\{\mathbf{v}_t\} \in \mathbb{R}^{2d}$ generated by the following update*

$$\mathbf{v}_{t+1} = \mathbf{v}_t - \nu P(\mathbf{v}_{t+1}),$$

*where $P$ is a monotone and Lipschitz continuous operator, and $\nu$ is a positive constant. Then for any $\mathbf{v} \in \mathbb{R}^{2d}$ and for each $t \ge 1$ we have*

$$P(\mathbf{v}_{t+1})^T(\mathbf{v}_{t+1} - \mathbf{v}) = \frac{1}{2\nu}\|\mathbf{v}_t - \mathbf{v}\|^2 - \frac{1}{2\nu}\|\mathbf{v}_{t+1} - \mathbf{v}\|^2 - \frac{1}{2\nu}\|\mathbf{v}_{t+1} - \mathbf{v}_t\|^2.$$

The following lemma is the strong PD empirical risk bound of PPM.

**Lemma 14.** *Let $\{\mathbf{x}_t, \mathbf{y}_t\}$ be the iterates generated by PPM in (44). Assume $\nu$ is a positive constant. Suppose the ESP solution exists. Assume that $F_S(\mathbf{x}, \mathbf{y})$ is a convex function of $\mathbf{x}$ for any $\mathbf{y}$ and is a concave function of $\mathbf{y}$ for any $\mathbf{x}$. Suppose $\sup_{\mathbf{x}\in\mathcal{X}}\|\mathbf{x}\| \le R_X$ and $\sup_{\mathbf{y}\in\mathcal{Y}}\|\mathbf{y}\| \le R_Y$. If Assumption 2 holds, then for all $T \ge 1$, we have*

$$\sup_{\mathbf{y}\in\mathcal{Y}} F_S(\bar{\mathbf{x}}_T, \mathbf{y}) - \inf_{\mathbf{x}\in\mathcal{X}} F_S(\mathbf{x}, \bar{\mathbf{y}}_T) \le \frac{R_X^2 + R_Y^2}{2\nu T}.$$

*Proof.* The update of the PPM in (44) can be written as

$$\mathbf{v}_{t+1} = \mathbf{v}_t - \nu P(\mathbf{v}_{t+1}).$$

According to Lemma 1 in (Mokhtari et al., 2019), if $F_S(\mathbf{x}, \mathbf{y})$ is convex-concave and Assumption 2 holds, then $P(\mathbf{v})$ defined in (45) is monotone and Lipschitz continuous. According to Lemma 13, we have

$$P(\mathbf{v}_{t+1})^T(\mathbf{v}_{t+1} - \mathbf{v}) = \frac{1}{2\nu}\|\mathbf{v}_t - \mathbf{v}\|^2 - \frac{1}{2\nu}\|\mathbf{v}_{t+1} - \mathbf{v}\|^2 - \frac{1}{2\nu}\|\mathbf{v}_{t+1} - \mathbf{v}_t\|^2.$$

Taking a summation of the above inequality from $t = 0$ to $T - 1$, we obtain

$$\sum_{t=0}^{T-1} P(\mathbf{v}_{t+1})^T(\mathbf{v}_{t+1} - \mathbf{v}) \le \frac{1}{2\nu}\|\mathbf{v}_0 - \mathbf{v}\|^2 - \frac{1}{2\nu}\|\mathbf{v}_T - \mathbf{v}\|^2. \tag{46}$$

According to (46), we know that

$$\sum_{t=0}^{T-1} P(\mathbf{v}_{t+1})^T (\mathbf{v}_{t+1} - \mathbf{v}) \leq \frac{1}{2\nu} \|\mathbf{v}_0 - \mathbf{v}\|^2$$

$$= \frac{\|\mathbf{x}_0 - \mathbf{x}\|^2 + \|\mathbf{y}_0 - \mathbf{y}\|^2}{2\nu}.$$

Combined this result with Lemma 12, we can write

$$F_S(\bar{\mathbf{x}}_T, \mathbf{y}) - F_S(\mathbf{x}, \bar{\mathbf{y}}_T) \leq \frac{\|\mathbf{x}_0 - \mathbf{x}\|^2 + \|\mathbf{y}_0 - \mathbf{y}\|^2}{2\nu T},$$

which implies that

$$\sup_{\mathbf{y} \in \mathcal{Y}} F_S(\bar{\mathbf{x}}_T, \mathbf{y}) - \inf_{\mathbf{x} \in \mathcal{X}} F_S(\mathbf{x}, \bar{\mathbf{y}}_T) \leq \frac{R_X^2 + R_Y^2}{2\nu T}.$$

The proof is complete. □

Combined Lemma 7 and Lemma 14, we know that the argument stability bound of PPM is

$$\|\bar{\mathbf{x}}_T^i - \bar{\mathbf{x}}_T\| + \|\bar{\mathbf{y}}_T^i - \bar{\mathbf{y}}_T\| \leq \frac{4L}{n\mu} + 4\sqrt{\frac{1}{\mu}} \sqrt{\triangle_S^s(\bar{\mathbf{x}}_T, \bar{\mathbf{y}}_T)}$$

$$\leq \frac{4L}{n\mu} + 4\sqrt{\frac{1}{\mu}} \sqrt{\frac{R_X^2 + R_Y^2}{2\nu T}}. \tag{47}$$

Furthermore, for any $\mathbf{x} \in \mathcal{X}$, $\mathbf{y} \in \mathcal{Y}$ and $z \in \mathcal{Z}$,

$$f(\mathbf{x}, \mathbf{y}; z) \leq f(0, 0; z) + L\|\mathbf{x} - 0\| + L\|\mathbf{y} - 0\| \leq \sup_{z \in \mathcal{Z}} f(0, 0; z) + L(R_X + R_Y), \tag{48}$$

Therefore, plugging (48), the stability bound in (47) and the strong PD empirical risk bound in Lemma 14 into Theorem 1, we obtain generalization bounds of PPM, shown as below.

**Theorem 7.** *Assume for all $z$, the function $(\mathbf{x}, \mathbf{y}) \mapsto f(\mathbf{x}, \mathbf{y}; z)$ is $\mu$-SC-SC. Suppose $\sup_{\mathbf{x} \in \mathcal{X}} \|\mathbf{x}\| \leq R_X$ and $\sup_{\mathbf{y} \in \mathcal{Y}} \|\mathbf{y}\| \leq R_Y$. Let $\{\mathbf{x}_t, \mathbf{y}_t\}$ be produced by (44). Assume $\nu$ is a positive constant. Denote $A_{\mathbf{x}}(S) = \bar{\mathbf{x}}_T$ and $A_{\mathbf{y}}(S) = \bar{\mathbf{y}}_T$ for $(\bar{\mathbf{x}}_T, \bar{\mathbf{y}}_T)$ in (42). Fixed any $\eta > 0$. Let $M = \sup_{z \in \mathcal{Z}} f(0, 0; z) + L(R_X + R_Y)$. Let $E = \frac{R_X^2 + R_Y^2}{2\nu T}$ and $B = \frac{4L}{n\mu} + 4\sqrt{\frac{1}{\mu}}\sqrt{E}$. There exists an absolute positive constant $C$.*

*(a) If Assumptions 1 and 2 hold, then for any $\delta > 0$, with probability at least $1 - \delta$, we have*

$$F(\bar{\mathbf{x}}_T, \bar{\mathbf{y}}_T) \leq (1 + \eta) F_S(\bar{\mathbf{x}}_T, \bar{\mathbf{y}}_T) + C \frac{1 + \eta}{\eta} \left( \frac{M}{n} \log(1/\delta) + B \log_2 n \log(1/\delta) \right).$$

*(b) If Assumptions 1 and 2 hold, then for any $\delta > 0$, with probability at least $1 - \delta$, we have*

$$R(\bar{\mathbf{x}}_T) \leq (1 + \eta) R_S(\bar{\mathbf{x}}_T) + C \frac{1 + \eta}{\eta} \left( \frac{M}{n} \log \frac{1}{\delta} + \left( \frac{\beta}{\mu} + 1 \right) LB \log_2 n \log \frac{1}{\delta} \right).$$

*(c) If Assumptions 1 and 2 hold, then for any $\delta > 0$, with probability at least $1 - \delta$, we have*

$$\triangle^s(\bar{\mathbf{x}}_T, \bar{\mathbf{y}}_T) \leq (1 + \eta) E + C(1 + \eta) \left( \frac{L^2(1 + \eta)}{n\mu\eta} + \frac{M}{n} + \left( 1 + \frac{\beta}{\mu} \right) BL \log_2 n \right) \log \left( \frac{1}{\delta} \right).$$

*(d) If Assumptions 1 and 2 hold, then for any $\delta > 0$, with probability at least $1 - \delta$, we have*

$$\triangle^s(\bar{\mathbf{x}}_T, \bar{\mathbf{y}}_T) - \triangle_S^s(\bar{\mathbf{x}}_T, \bar{\mathbf{y}}_T)$$

$$\leq \eta E + C(1 + \eta) \left( \frac{L^2(1 + \eta)}{n\mu\eta} + \frac{M}{n} + \left( 1 + \frac{\beta}{\mu} \right) BL \log_2 n \right) \log \left( \frac{1}{\delta} \right).$$

*(e) If Assumptions 1 and 2 hold, then for any $\delta > 0$, with probability at least $1 - \delta$, we have*

$$R(\bar{\mathbf{x}}_T) - (1 + \eta) \inf_{\mathbf{x} \in \mathcal{X}} R(\mathbf{x}) \leq C \frac{2 + \eta}{\eta} \left( \frac{M}{n} \log \frac{1}{\delta} + \left( \frac{\beta}{\mu} + 1 \right) LB \log_2 n \log \frac{1}{\delta} + E \right).$$

**Remark 14.** When conditions in Theorem 7 hold, we obtain that (a) If Assumption 1 and 2 hold and $F_S(\bar{\mathbf{x}}_T, \bar{\mathbf{y}}_T) = \mathcal{O}\left(\frac{1}{n}\right)$, then for any $\delta > 0$, with probability at least $1 - \delta$, the plain generalization error of $(\bar{\mathbf{x}}_T, \bar{\mathbf{y}}_T)$ of PPM is of the order $\mathcal{O}\left(\left(\frac{1}{n} + \sqrt{\frac{1}{T}}\right)\log_2 n \log(1/\delta)\right)$. (b) If Assumptions 1 and 2 hold and $R_S(\bar{\mathbf{x}}_T) = \mathcal{O}\left(\frac{1}{n}\right)$, then for any $\delta > 0$, with probability at least $1 - \delta$, the primal generalization error of $(\bar{\mathbf{x}}_T, \bar{\mathbf{y}}_T)$ of PPM is of the order $\mathcal{O}\left(\left(\frac{1}{n} + \sqrt{\frac{1}{T}}\right)\log_2 n \log(1/\delta)\right)$. (c) If Assumptions 1 and 2 hold, then for any $\delta > 0$, with probability at least $1 - \delta$, the strong PD population risk and strong PD generalization error of $(\bar{\mathbf{x}}_T, \bar{\mathbf{y}}_T)$ of PPM are all of the order $\mathcal{O}\left(\left(\frac{1}{n} + \sqrt{\frac{1}{T}}\right)\log_2 n \log(1/\delta)\right)$. (d) If Assumptions 1 and 2 hold and $\inf_{\mathbf{x} \in \mathcal{X}} R(\mathbf{x}) = \mathcal{O}\left(\frac{1}{n}\right)$, then for any $\delta > 0$, with probability at least $1 - \delta$, the excess primal population risk of $(\bar{\mathbf{x}}_T, \bar{\mathbf{y}}_T)$ of PPM is of the order $\mathcal{O}\left(\left(\frac{1}{n} + \sqrt{\frac{1}{T}}\right)\log_2 n \log(1/\delta)\right)$. For the above bounds, we can take $T = \mathcal{O}(n^2)$ gradient evaluations to get bound of the order $\mathcal{O}\left(\frac{\log n}{n} \log(1/\delta)\right)$.

## F EXTRAGRADIENT METHOD

EG is a classical algorithm for solving minimax problems introduced by (Korpelevich, 1976). We now introduce some notations. Followed (Mokhtari et al., 2019), we consider the following update of EG: given stepsize parameter $\nu$, we first compute a set of mid-point iterates $\{\mathbf{x}_{t+\frac{1}{2}}, \mathbf{y}_{t+\frac{1}{2}}\}$

$$\mathbf{x}_{t+\frac{1}{2}} = \mathbf{x}_t - \nu \nabla_{\mathbf{x}} F_S(\mathbf{x}_t, \mathbf{y}_t),$$
$$\mathbf{y}_{t+\frac{1}{2}} = \mathbf{y}_t + \nu \nabla_{\mathbf{y}} F_S(\mathbf{x}_t, \mathbf{y}_t), \tag{49}$$

we then compute the next iterates $\{\mathbf{x}_{t+1}, \mathbf{y}_{t+1}\}$

$$\mathbf{x}_{t+1} = \mathbf{x}_t - \nu \nabla_{\mathbf{x}} F_S(\mathbf{x}_{t+\frac{1}{2}}, \mathbf{y}_{t+\frac{1}{2}}),$$
$$\mathbf{y}_{t+1} = \mathbf{y}_t + \nu \nabla_{\mathbf{y}} F_S(\mathbf{x}_{t+\frac{1}{2}}, \mathbf{y}_{t+\frac{1}{2}}). \tag{50}$$

Consider the averaged iterate

$$\bar{\mathbf{x}}_T = \frac{1}{T}\sum_{t=1}^{T} \mathbf{x}_t \quad \text{and} \quad \bar{\mathbf{y}}_T = \frac{1}{T}\sum_{t=1}^{T} \mathbf{y}_t. \tag{51}$$

Assume that the initial point satisfies $\mathbf{x}_0 = \mathbf{x}_{-1/2}$ and $\mathbf{y}_0 = \mathbf{y}_{-1/2}$.

We now show the strong PD empirical risk bound for $(\bar{\mathbf{x}}_T, \bar{\mathbf{y}}_T)$ of EG.

**Lemma 15.** *(Mokhtari et al., 2019) Let $\{\mathbf{x}_t, \mathbf{y}_t\}$, $\{\mathbf{x}_{t+1/2}, \mathbf{y}_{t+1/2}\}$ be the iterates generated by the EG updates in (49) and (50). Assume that the initial point satisfies $\mathbf{x}_0 = \mathbf{x}_{-1/2}$ and $\mathbf{y}_0 = \mathbf{y}_{-1/2}$. Suppose the ESP solution $(\hat{\mathbf{x}}^*, \hat{\mathbf{y}}^*)$ exists. Assume that $F_S(\mathbf{x}, \mathbf{y})$ is a convex function of $\mathbf{x}$ for any $\mathbf{y}$ and is a concave function of $\mathbf{y}$ for any $\mathbf{x}$. If Assumption 2 holds and the stepsize $\nu$ satisfies the condition $\nu = \frac{c}{2\beta}$ for any $c \in (0, 1)$, then:*

*(a) the iterates $\{\mathbf{x}_t, \mathbf{y}_t\}$, $\{\mathbf{x}_{t+1/2}, \mathbf{y}_{t+1/2}\}$ stay within the compact convex set*

$$D := \left\{(\mathbf{x}, \mathbf{y}) | \|\mathbf{x} - \hat{\mathbf{x}}^*\|^2 + \|\mathbf{y} - \hat{\mathbf{y}}^*\|^2 \le \left(2 + \frac{2}{1 - 4\nu^2\beta^2}\right)(\|\mathbf{x}_0 - \hat{\mathbf{x}}^*\|^2 + \|\mathbf{y}_0 - \hat{\mathbf{y}}^*\|^2)\right\}. \tag{52}$$

*(b) for all $T \ge 1$, we have*

$$\sup_{\mathbf{y}:(\bar{\mathbf{x}}_T, \mathbf{y}) \in D} F_S(\bar{\mathbf{x}}_T, \mathbf{y}) - \inf_{\mathbf{x}:(\mathbf{x}, \bar{\mathbf{y}}_T) \in D} F_S(\mathbf{x}, \bar{\mathbf{y}}_T) \le \frac{2\beta(16 + \frac{33}{2(1-c^2)})(\|\mathbf{x}_0 - \hat{\mathbf{x}}^*\|^2 + \|\mathbf{y}_0 - \hat{\mathbf{y}}^*\|^2)}{T}.$$

Combined Lemma 7 and Lemma 15, we know that the argument stability bound of EG is

$$\|\bar{\mathbf{x}}_T^i - \bar{\mathbf{x}}_T\| + \|\bar{\mathbf{y}}_T^i - \bar{\mathbf{y}}_T\| \leq \frac{4L}{n\mu} + 4\sqrt{\frac{1}{\mu}}\sqrt{\triangle_S^s(\bar{\mathbf{x}}_T, \bar{\mathbf{y}}_T)}$$

$$\leq \frac{4L}{n\mu} + 4\sqrt{\frac{1}{\mu}}\sqrt{\frac{2\beta(16 + \frac{33}{2(1-c^2)})(\|\mathbf{x}_0 - \hat{\mathbf{x}}^*\|^2 + \|\mathbf{y}_0 - \hat{\mathbf{y}}^*\|^2)}{T}}. \tag{53}$$

Furthermore, for any $\mathbf{x} \in \mathcal{X}$, $\mathbf{y} \in \mathcal{Y}$ and $z \in \mathcal{Z}$,

$$f(\mathbf{x}, \mathbf{y}; z) \leq f(\hat{\mathbf{x}}^*, \hat{\mathbf{y}}^*; z) + L\|\mathbf{x} - \hat{\mathbf{x}}^*\| + L\|\mathbf{y} - \hat{\mathbf{y}}^*\|$$

$$\leq \sup_{z \in \mathcal{Z}} f(\hat{\mathbf{x}}^*, \hat{\mathbf{y}}^*; z) + \sqrt{2}L\sqrt{\|\mathbf{x} - \hat{\mathbf{x}}^*\|^2 + \|\mathbf{y} - \hat{\mathbf{y}}^*\|^2}$$

$$\leq \sup_{z \in \mathcal{Z}} f(\hat{\mathbf{x}}^*, \hat{\mathbf{y}}^*; z) + L\sqrt{\left(4 + \frac{4}{1 - 4\nu^2\beta^2}\right)(\|\mathbf{x}_0 - \hat{\mathbf{x}}^*\|^2 + \|\mathbf{y}_0 - \hat{\mathbf{y}}^*\|^2)}, \tag{54}$$

where the first inequality follows from Assumption 1, the second inequality follows from Caucy-Schwarz inequality and the last inequality follows from (52). Now, plugging (54), the stability bound in (53) and the strong PD empirical risk bound in Lemma 15 into Theorem 1, we obtain generalization bounds of EG. The main theorem is shown below.

**Theorem 8.** *Assume for all $z$, the function $(\mathbf{x}, \mathbf{y}) \mapsto f(\mathbf{x}, \mathbf{y}; z)$ is $\mu$-SC-SC. Let $\{\mathbf{x}_t, \mathbf{y}_t\}$ and $\{\mathbf{x}_{t+1/2}, \mathbf{y}_{t+1/2}\}$ be the iterates produced by (49)-(50). Assume the stepsize $\nu$ satisfies the condition $\nu = \frac{c}{2\beta}$ for any $c \in (0, 1)$. Denote $A_{\mathbf{x}}(S) = \bar{\mathbf{x}}_T$ and $A_{\mathbf{y}}(S) = \bar{\mathbf{y}}_T$ for $(\bar{\mathbf{x}}_T, \bar{\mathbf{y}}_T)$ in (51). Denote the ESP solution as $(\hat{\mathbf{x}}^*, \hat{\mathbf{y}}^*)$. Consider the compact convex set in (52). Fixed any $\eta > 0$. Let $M = \sup_{z \in \mathcal{Z}} f(\hat{\mathbf{x}}^*, \hat{\mathbf{y}}^*; z) + L\sqrt{\left(4 + \frac{4}{1-4\nu^2\beta^2}\right)(\|\mathbf{x}_0 - \hat{\mathbf{x}}^*\|^2 + \|\mathbf{y}_0 - \hat{\mathbf{y}}^*\|^2)}$. Let $B = \frac{2\beta(16+\frac{33}{2(1-c^2)})(\|\mathbf{x}_0-\hat{\mathbf{x}}^*\|^2+\|\mathbf{y}_0-\hat{\mathbf{y}}^*\|^2)}{T}$ and $E = \frac{4L}{n\mu} + 4\sqrt{\frac{1}{\mu}}\sqrt{B}$. There exists an absolute positive constant $C$.*

*(a) If Assumptions 1 and 2 hold, then for any $\delta > 0$, with probability at least $1 - \delta$, we have*

$$F(\bar{\mathbf{x}}_T, \bar{\mathbf{y}}_T) \leq (1 + \eta)F_S(\bar{\mathbf{x}}_T, \bar{\mathbf{y}}_T) + C\frac{1+\eta}{\eta}\left(\frac{M}{n}\log(1/\delta) + B\log_2 n \log(1/\delta)\right).$$

*(b) If Assumptions 1 and 2 hold, then for any $\delta > 0$, with probability at least $1 - \delta$, we have*

$$R(\bar{\mathbf{x}}_T) \leq (1 + \eta)R_S(\bar{\mathbf{x}}_T) + C\frac{1+\eta}{\eta}\left(\frac{M}{n}\log\frac{1}{\delta} + \left(\frac{\beta}{\mu} + 1\right)LB\log_2 n \log\frac{1}{\delta}\right).$$

*(c) If Assumptions 1 and 2 hold, then for any $\delta > 0$, with probability at least $1 - \delta$, we have*

$$\triangle^s(\bar{\mathbf{x}}_T, \bar{\mathbf{y}}_T) \leq (1 + \eta)E + C(1+\eta)\left(\frac{L^2(1+\eta)}{n\mu\eta} + \frac{M}{n} + \left(1 + \frac{\beta}{\mu}\right)BL\log_2 n\right)\log\left(\frac{1}{\delta}\right).$$

*(d) If Assumptions 1 and 2 hold, then for any $\delta > 0$, with probability at least $1 - \delta$, we have*

$$\triangle^s(\bar{\mathbf{x}}_T, \bar{\mathbf{y}}_T) - \triangle_S^s(\bar{\mathbf{x}}_T, \bar{\mathbf{y}}_T)$$

$$\leq \eta E + C(1+\eta)\left(\frac{L^2(1+\eta)}{n\mu\eta} + \frac{M}{n} + \left(1 + \frac{\beta}{\mu}\right)BL\log_2 n\right)\log\left(\frac{1}{\delta}\right).$$

*(e) If Assumptions 1 and 2 hold, then for any $\delta > 0$, with probability at least $1 - \delta$, we have*

$$R(\bar{\mathbf{x}}_T) \leq (1 + \eta)\inf_{\mathbf{x} \in \mathcal{X}} R(\mathbf{x}) + C\frac{2+\eta}{\eta}\left(\frac{M}{n}\log\frac{1}{\delta} + \left(\frac{\beta}{\mu} + 1\right)LB\log_2 n \log\frac{1}{\delta} + E\right).$$

**Remark 15.** When conditions in Theorem 8 hold, we obtain that (a) If Assumptions 1 and 2 hold and $F_S(\bar{\mathbf{x}}_T, \bar{\mathbf{y}}_T) = \mathcal{O}\left(\frac{1}{n}\right)$, then for any $\delta > 0$, with probability at least $1 - \delta$, the plain generalization

error of $(\bar{\mathbf{x}}_T, \bar{\mathbf{y}}_T)$ of EG is of the order $\mathcal{O}\left(\left(\frac{1}{n} + \sqrt{\frac{1}{T}}\right) \log_2 n \log(1/\delta)\right)$. (b) If Assumptions 1 and 2 hold and $R_S(\bar{\mathbf{x}}_T) = \mathcal{O}\left(\frac{1}{n}\right)$, then for any $\delta > 0$, with probability at least $1 - \delta$, the primal generalization error of $(\bar{\mathbf{x}}_T, \bar{\mathbf{y}}_T)$ of EG is of the order $\mathcal{O}\left(\left(\frac{1}{n} + \sqrt{\frac{1}{T}}\right) \log_2 n \log(1/\delta)\right)$. (c) If Assumptions 1 and 2 hold, then for any $\delta > 0$, with probability at least $1 - \delta$, the strong PD population risk and the strong PD generalization error of $(\bar{\mathbf{x}}_T, \bar{\mathbf{y}}_T)$ of EG are all of the order $\mathcal{O}\left(\left(\frac{1}{n} + \sqrt{\frac{1}{T}}\right) \log_2 n \log(1/\delta)\right)$. (d) If Assumptions 1 and 2 hold and $\inf_{\mathbf{x} \in \mathcal{X}} R(\mathbf{x}) = \mathcal{O}\left(\frac{1}{n}\right)$, then for any $\delta > 0$, with probability at least $1 - \delta$, the excess primal population risk of $(\bar{\mathbf{x}}_T, \bar{\mathbf{y}}_T)$ of EG is $\mathcal{O}\left(\left(\frac{1}{n} + \sqrt{\frac{1}{T}}\right) \log_2 n \log(1/\delta)\right)$. For the above bounds, we can take $T = \mathcal{O}(n^2)$ gradient evaluations to get bound of the order $\mathcal{O}\left(\frac{\log n}{n} \log(1/\delta)\right)$.

# G    OPTIMISTIC GRADIENT DESCENT ASCENT

OGDA is introduced by Popov (1980), as a variant of the EG method. We introduce some notations to state the result of OGDA. Given a stepsize parameter $\nu > 0$, OGDA do the following update for each $t \geq 0$

$$\begin{aligned}
\mathbf{x}_{t+1} &= \mathbf{x}_t - 2\nu \nabla_{\mathbf{x}} F_S(\mathbf{x}_t, \mathbf{y}_t) + \nu \nabla_{\mathbf{x}} F_S(\mathbf{x}_{t-1}, \mathbf{y}_{t-1}), \\
\mathbf{y}_{t+1} &= \mathbf{y}_t + 2\nu \nabla_{\mathbf{y}} F_S(\mathbf{x}_t, \mathbf{y}_t) - \nu \nabla_{\mathbf{y}} F_S(\mathbf{x}_{t-1}, \mathbf{y}_{t-1}).
\end{aligned} \tag{55}$$

Assume that the initial point satisfies $\mathbf{x}_0 = \mathbf{x}_{-1}$ and $\mathbf{y}_0 = \mathbf{y}_{-1}$. Consider the averaged iterate

$$\bar{\mathbf{x}}_T = \frac{1}{T} \sum_{t=1}^{T} \mathbf{x}_t \quad \text{and} \quad \bar{\mathbf{y}}_T = \frac{1}{T} \sum_{t=1}^{T} \mathbf{y}_t. \tag{56}$$

We first provide a lemma on the strong PD empirical risk of OGDA.

**Lemma 16.** *(Mokhtari et al., 2019) Let $\{\mathbf{x}_t, \mathbf{y}_t\}$ be the iterates generated by the OGDA updates in (55). Assume that the initial point satisfies $\mathbf{x}_0 = \mathbf{x}_{-1}$ and $\mathbf{y}_0 = \mathbf{y}_{-1}$. Suppose the ESP solution $(\hat{\mathbf{x}}^*, \hat{\mathbf{y}}^*)$ exists. Assume that $F_S(\mathbf{x}, \mathbf{y})$ is a convex function of $\mathbf{x}$ for any $\mathbf{y}$ and is a concave function of $\mathbf{y}$ for any $\mathbf{x}$. If Assumption 2 holds and the stepsize $\nu$ satisfies $0 < \nu \leq \frac{1}{4\beta}$, then:*

*(a) the iterates $\{\mathbf{x}_t, \mathbf{y}_t\}$ stay within the compact convex set*

$$D := \left\{ (\mathbf{x}, \mathbf{y}) | \|\mathbf{x} - \hat{\mathbf{x}}^*\|^2 + \|\mathbf{y} - \hat{\mathbf{y}}^*\|^2 \leq 2(\|\mathbf{x}_0 - \hat{\mathbf{x}}^*\|^2 + \|\mathbf{y}_0 - \hat{\mathbf{y}}^*\|^2) \right\}. \tag{57}$$

*(b) for all $T \geq 1$, we have*

$$\sup_{\mathbf{y}:(\bar{\mathbf{x}}_T, \mathbf{y}) \in D} F_S(\bar{\mathbf{x}}_T, \mathbf{y}) - \inf_{\mathbf{x}:(\mathbf{x}, \bar{\mathbf{y}}_T) \in D} F_S(\mathbf{x}, \bar{\mathbf{y}}_T) \leq \frac{(16\beta + \frac{1}{2\nu})(\|\mathbf{x}_0 - \hat{\mathbf{x}}^*\|^2 + \|\mathbf{y}_0 - \hat{\mathbf{y}}^*\|^2)}{T}.$$

Combined Lemma 7 and Lemma 16, we know that the argument stability bound of OGDA is

$$\begin{aligned}
\|\bar{\mathbf{x}}_T^i - \bar{\mathbf{x}}_T\| + \|\bar{\mathbf{y}}_T^i - \bar{\mathbf{y}}_T\| &\leq \frac{4L}{n\mu} + 4\sqrt{\frac{1}{\mu}}\sqrt{\triangle_S^s(\bar{\mathbf{x}}_T, \bar{\mathbf{y}}_T)} \\
&\leq \frac{4L}{n\mu} + 4\sqrt{\frac{1}{\mu}}\sqrt{\frac{(16\beta + \frac{1}{2\nu})(\|\mathbf{x}_0 - \hat{\mathbf{x}}^*\|^2 + \|\mathbf{y}_0 - \hat{\mathbf{y}}^*\|^2)}{T}}.
\end{aligned} \tag{58}$$

Moreover, similar to (54), we have

$$\begin{aligned}
f(\mathbf{x}, \mathbf{y}; z) &\leq f(\hat{\mathbf{x}}^*, \hat{\mathbf{y}}^*; z) + L\|\mathbf{x} - \hat{\mathbf{x}}^*\| + L\|\mathbf{y} - \hat{\mathbf{y}}^*\| \\
&\leq \sup_{z \in \mathcal{Z}} f(\hat{\mathbf{x}}^*, \hat{\mathbf{y}}^*; z) + 2L\sqrt{\|\mathbf{x}_0 - \hat{\mathbf{x}}^*\|^2 + \|\mathbf{y}_0 - \hat{\mathbf{y}}^*\|^2}.
\end{aligned} \tag{59}$$

Therefore, plugging (59), the stability bound in (58) and the strong PD empirical risk bound in Lemma 16 into Theorem 1, we obtain generalization bounds of OGDA.

**Theorem 9.** *Assume for all $z$, the function $(\mathbf{x}, \mathbf{y}) \mapsto f(\mathbf{x}, \mathbf{y}; z)$ is $\mu$-SC-SC. Let $\{\mathbf{x}_t, \mathbf{y}_t\}$ be produced by (55). Assume the stepsize parameter $\nu$ satisfies $0 < \nu \leq \frac{1}{4\beta}$. Denote $A_{\mathbf{x}}(S) = \bar{\mathbf{x}}_T$ and $A_{\mathbf{y}}(S) = \bar{\mathbf{y}}_T$ for $(\bar{\mathbf{x}}_T, \bar{\mathbf{y}}_T)$ in (56). Denote the ESP solution as $(\hat{\mathbf{x}}^*, \hat{\mathbf{y}}^*)$. Consider the compact convex set in (57). Fixed any $\eta > 0$. Let $M = \sup_{z \in \mathcal{Z}} f(\hat{\mathbf{x}}^*, \hat{\mathbf{y}}^*; z) + 2L\sqrt{\|\mathbf{x}_0 - \hat{\mathbf{x}}^*\|^2 + \|\mathbf{y}_0 - \hat{\mathbf{y}}^*\|^2}$. Define $B = \frac{(16\beta + \frac{1}{2\nu})(\|\mathbf{x}_0 - \hat{\mathbf{x}}^*\|^2 + \|\mathbf{y}_0 - \hat{\mathbf{y}}^*\|^2)}{T}$ and $E = \frac{4L}{n\mu} + 4\sqrt{\frac{1}{\mu}}\sqrt{B}$. There exists an absolute positive constant $C$.*

*(a) If Assumptions 1 and 2 hold, then for any $\delta > 0$, with probability at least $1 - \delta$, we have*

$$F(\bar{\mathbf{x}}_T, \bar{\mathbf{y}}_T) \leq (1 + \eta) F_S(\bar{\mathbf{x}}_T, \bar{\mathbf{y}}_T) + C \frac{1 + \eta}{\eta}\Big(\frac{M}{n}\log(1/\delta) + B \log_2 n \log(1/\delta)\Big).$$

*(b) If Assumptions 1 and 2 hold, then for any $\delta > 0$, with probability at least $1 - \delta$, we have*

$$R(\bar{\mathbf{x}}_T) \leq (1 + \eta) R_S(\bar{\mathbf{x}}_T) + C \frac{1 + \eta}{\eta}\Big(\frac{M}{n}\log\frac{1}{\delta} + \Big(\frac{\beta}{\mu} + 1\Big)LB\log_2 n \log\frac{1}{\delta}\Big).$$

*(c) If Assumptions 1 and 2 hold, then for any $\delta > 0$, with probability at least $1 - \delta$, we have*

$$\triangle^s(\bar{\mathbf{x}}_T, \bar{\mathbf{y}}_T) \leq (1 + \eta) E + C(1 + \eta)\Big(\frac{L^2(1 + \eta)}{n\mu\eta} + \frac{M}{n} + \Big(1 + \frac{\beta}{\mu}\Big)BL\log_2 n\Big)\log\Big(\frac{1}{\delta}\Big).$$

*(d) If Assumptions 1 and 2 hold, then for any $\delta > 0$, with probability at least $1 - \delta$, we have*

$$\triangle^s(\bar{\mathbf{x}}_T, \bar{\mathbf{y}}_T) - \triangle_S^s(\bar{\mathbf{x}}_T, \bar{\mathbf{y}}_T)$$
$$\leq \eta E + C(1 + \eta)\Big(\frac{L^2(1 + \eta)}{n\mu\eta} + \frac{M}{n} + \Big(1 + \frac{\beta}{\mu}\Big)BL\log_2 n\Big)\log\Big(\frac{1}{\delta}\Big).$$

*(e) If Assumptions 1 and 2 hold, then for any $\delta > 0$, with probability at least $1 - \delta$, we have*

$$R(\bar{\mathbf{x}}_T) \leq (1 + \eta)\inf_{\mathbf{x} \in \mathcal{X}} R(\mathbf{x}) + C \frac{2 + \eta}{\eta}\Big(\frac{M}{n}\log\frac{1}{\delta} + \Big(\frac{\beta}{\mu} + 1\Big)LB\log_2 n \log\frac{1}{\delta} + E\Big).$$

**Remark 16.** When conditions in Theorem 9 hold, we obtain that (a) If Assumptions 1 and 2 hold and $F_S(\bar{\mathbf{x}}_T, \bar{\mathbf{y}}_T) = \mathcal{O}\Big(\frac{1}{n}\Big)$, then for any $\delta > 0$, with probability at least $1 - \delta$, the plain generalization error of $(\bar{\mathbf{x}}_T, \bar{\mathbf{y}}_T)$ of OGDA is of the order $\mathcal{O}\Big(\Big(\frac{1}{n} + \sqrt{\frac{1}{T}}\Big)\log_2 n \log(1/\delta)\Big)$. (b) If Assumptions 1 and 2 hold and $R_S(\bar{\mathbf{x}}_T) = \mathcal{O}\Big(\frac{1}{n}\Big)$, then for any $\delta > 0$, with probability at least $1 - \delta$, the primal generalization error of $(\bar{\mathbf{x}}_T, \bar{\mathbf{y}}_T)$ of OGDA is of the order $\mathcal{O}\Big(\Big(\frac{1}{n} + \sqrt{\frac{1}{T}}\Big)\log_2 n \log(1/\delta)\Big)$. (c) If Assumptions 1 and 2 hold, then for any $\delta > 0$, with probability at least $1 - \delta$, the strong PD population risk and the strong PD generalization error of $(\bar{\mathbf{x}}_T, \bar{\mathbf{y}}_T)$ of OGDA are all of the order $\mathcal{O}\Big(\Big(\frac{1}{n} + \sqrt{\frac{1}{T}}\Big)\log_2 n \log(1/\delta)\Big)$. (d) If Assumptions 1 and 2 hold and $\inf_{\mathbf{x} \in \mathcal{X}} R(\mathbf{x}) = \mathcal{O}\Big(\frac{1}{n}\Big)$, then for any $\delta > 0$, with probability at least $1 - \delta$, the excess primal population risk of OGDA is of the order $\mathcal{O}\Big(\Big(\frac{1}{n} + \sqrt{\frac{1}{T}}\Big)\log_2 n \log(1/\delta)\Big)$. For the above bounds, we can take $T = \mathcal{O}(n^2)$ gradient evaluations to get bound of the order $\mathcal{O}\Big(\frac{\log n}{n}\log(1/\delta)\Big)$.

## H   AUXILIARY DESCRIPTIONS OF TABLE 1

In Table 1, Lip means Lipschitz continuity and S means smoothness. (R)-ESP means the (regularized)-empirical risk saddle point (Zhang et al., 2021a). C-SC means convex-$\mu$-strongly-concave, and NC-SC means nonconvex-$\mu$-strongly-concave. A function $f(\mathbf{x}, \mathbf{y})$ is called nonconvex-strongly-concave if $f(\mathbf{x}, \cdot)$ is strongly-concave for every $\mathbf{x}$. Moreover, a function $f(\mathbf{x}, \mathbf{y})$ is $\mu$-weakly-convex-weakly-concave (WC-WC) if $f + \frac{\mu}{2}\big(\|\mathbf{x}\|^2 + \|\mathbf{y}\|^2\big)$ is convex-concave. V-WC-WC is a variant of WC-WC, please refer to (Lei et al., 2021). PL means the two-sided PL condition, which relaxes the convex-concavity requirement of the objective function (Yang et al., 2020) and is usually used to guarantee the linear convergence rate (Karimi et al., 2016; Yang et al., 2020). AGDA algorithm is variant of GDA with alternating updates of the primal-dual variables. $c$ is a parameter in the step size, $\beta$ is a parameter in Assumption 2 and $k := \beta/\mu$.

# I  Numerical Experiments

In this section, we report preliminary experimental results to verify our theoretical results by performing numerical experiments on the simulated data. We study how the generalization error would behave along the number of samples. To this aim, we consider an isotropic Gaussian data vector $\mathbf{Z} \sim \mathcal{N}(\mathbf{0}, I_{d \times d})$ with zero mean and identity covariance. We will draw $n$ independent samples from the underlying Gaussian distribution to form a training dataset $S = \{\mathbf{z}_1, ..., \mathbf{z}_n\}$. We set the dimension $d$ of $\mathbf{Z}$ as $50$. Similar to the strongly-convex-strong-concave case of (Farnia & Ozdaglar, 2021), we consider the following minimax objective function

$$f(\mathbf{x}, \mathbf{y}; \mathbf{z}) = \mathbf{x}^T(\mathbf{z} - \mathbf{y}) + \frac{\mu}{2}(\|\mathbf{x}\|^2 - \|\mathbf{y}\|^2).$$

In the experiments, we set $\mu = 1$ and constrain optimization variables $\mathbf{x}$ and $\mathbf{y}$ to satisfy $\|\mathbf{x}\|, \|\mathbf{y}\| \le 100$ which we enforced by projection. For this minimax objective function, one can verify that

$$F(\mathbf{x}, \mathbf{y}) - F_S(\mathbf{x}, \mathbf{y}) = \mathbf{x}^T(\mathbb{E}[\mathbf{Z}] - \mathbb{E}_S[\mathbf{Z}]); \quad R(\mathbf{x}) - R_S(\mathbf{x}) = \mathbf{x}^T(\mathbb{E}[\mathbf{Z}] - \mathbb{E}_S[\mathbf{Z}]),$$

where $\mathbb{E}[\mathbf{Z}] = 0$ since the mean of the underlying Gaussian distribution is 0, and where $\mathbb{E}_S[\mathbf{Z}] = \frac{1}{n}\sum_{i=1}^{n} z_i$. For brevity, we call $|\mathbf{x}^T(\mathbb{E}[\mathbf{Z}] - \mathbb{E}_S[\mathbf{Z}])|$ the "generalization error".

We apply the above experimental settings to validate the theoretical results of GDA, SGDA, EG, and OGDA. We evaluate the generalization error $|\mathbf{x}^T(\mathbb{E}[\mathbf{Z}] - \mathbb{E}_S[\mathbf{Z}])|$ and apply these algorithms to $S$. For GDA and SGDA, we consider the stepsize parameter as $1/t$. We iterate GDA with $n^2$ times and SGDA with $n^4$ times. The generalization error of GDA and SGDA with different sizes of training data are reported in Figure 1. And for EG and OGDA, we select the stepsize parameter as $0.003$. We run EG and OGDA $n^2$ times. Similarly, the generalization error of EG and OGDA with different sizes of training data are given in Figure 2. From the two figures, we can see that the line of best fit for the generalization error is $\frac{\log^{3/2} n}{n^{0.98}}$ for GDA, $\frac{\log n}{n^{0.98}}$ for SGDA, $\frac{\log^{0.99} n}{n^{0.98}}$ for EG, and $\frac{\log n}{n^{1.02}}$ for OGDA. These results match the predictive rates of the plain generalization error and the primal generalization error in Table 1, i.e., $\frac{\log^{3/2} n}{n}$ for GDA, $\frac{\log n}{n}$ for SGDA, $\frac{\log n}{n}$ for EG, and $\frac{\log n}{n}$ for OGDA, which verifies our theoretical findings.

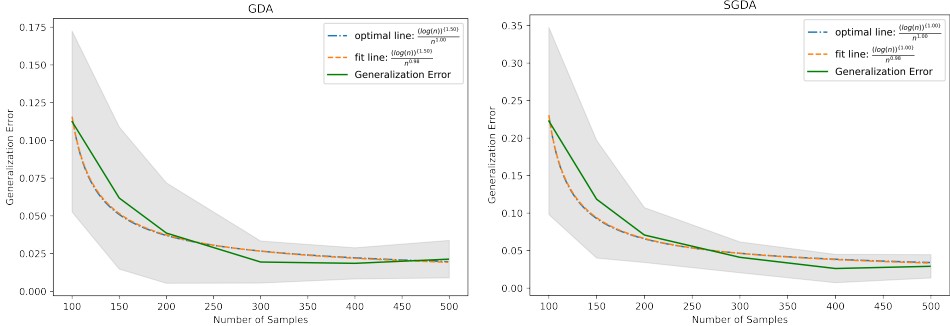

Figure 1: $|\mathbf{x}^T(\mathbb{E}[\mathbf{Z}] - \mathbb{E}_S[\mathbf{Z}])|$ versus the number of samples on GDA (left) and SGDA (right).

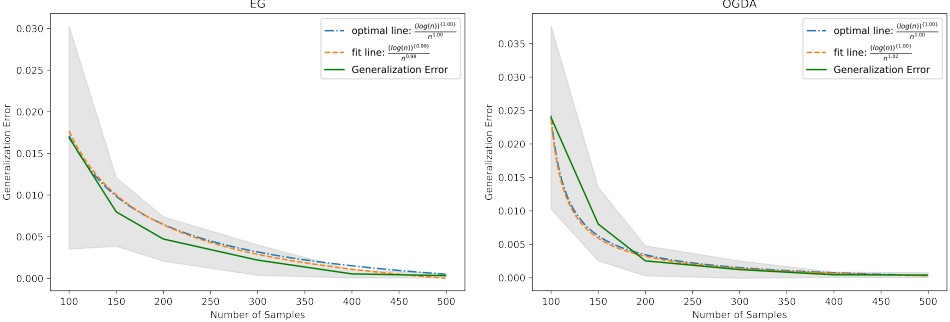

Figure 2: $|\mathbf{x}^T(\mathbb{E}[\mathbf{Z}] - \mathbb{E}_S[\mathbf{Z}])|$ versus the number of samples on EG (left) and OGDA (right).

