# OpenReview forum: "High Probability Generalization Bounds with Fast Rates for Minimax Problems"
_ICLR.cc/2022/Conference — ICLR 2022 Poster_

### Official Review · Reviewer_Wpqu · 2021-10-26

**Correctness:** 3
**Technical Novelty And Significance:** 3
**Empirical Novelty And Significance:** 3
**Recommendation:** 6
**Confidence:** 3

**Details Of Ethics Concerns:**

I did not find any ethics concerns.

**Main Review:**

The paper overall is written nicely and fairly clear. The problem setup is also interesting and has potential impact for other problems as well.

A concern I have for the results is the comparison of the generalization with respect to the empirical measures: is there a motivation for comparing these measures with a constant (1+eta) multiple of empirical measures? It seems that it would be more reasonable to have results for the case where eta approximates zero---when eta is a non-zero constant, then if we were to look at the difference between the generalization measure and the empirical measure, there would be a dependency on the empirical risk multiplied by eta, which may or may not have the desired dependency on order as expected.

Following the same concern as above, take for example the result Theorem 1-b., where the comparison of R(Ax(S)) is made with (1+eta)*R_S(Ax(S)). I took a look at the results of Farnia & Ozdaglar, and it appears that their paper compares the expected generalization risk E(R(Ax(S)-R_S(Ax(S))) which corresponds to eta->0.  Looking briefly at the referenced paper "Stability and generalization of stochastic gradient methods for minimax problems" by Lei et al., their results (Theorem 1) also compare the generalization measures directly with the empirical measures, without extra multiples such as (1+eta). Therefore it does seem that the fairness of the comparison can be discussed/explained in more detail.

Apart from the above concern, I agree with the authors that it would be interesting to see if assumptions can be relaxed. It would be helpful to have more discussions on the dependency of these assumptions.

**Summary Of The Paper:**

High-probability generalization bounds for minimax problems are proposed. The authors establish bounds for four quantities: plain generalization error, primal generalization error, strong primal-dual risk and strong primal-dual generalization error. The main contribution of the paper is in fast rates of the bounds compared to previous work for various applications to popular algorithms.

**Summary Of The Review:**

The paper is nicely written. There is a concern regarding the main theorem's dependency on the (1+eta) factor for the empirical measures. This also leads to a concern about the fairness of comparisons with related work.

---

> ### Author Response · Authors · 2021-11-18
> **Responses to Reviewer Wpqu**
>
> **Dear Reviewer Wpqu,** we sincerely appreciate your invaluable and constructive comments. We respond to your concerns one by one.
>
> **R1: A concern I have for the results is the comparison of the generalization with respect to the empirical measures......which may or may not have the desired dependency on order as expected.**
>
> **A**:  **Please refer to the public response to all Reviewers**. In this response, we have discussed the motivation for comparing the generalization measures with a constant $(1+\eta)$ multiple of empirical measures, the influence of $\eta$, and how our generalization bounds imply fast rates.
>
> **R2: Following the same concern as above, take for example the result Theorem 1-b......Therefore it does seem that the fairness of the comparison can be discussed/explained in more detail.**
>
> **A**:  We have carefully revised the comparison with the related work.
> **Please refer to the public response to all Reviewers**. In this response, we have clarified the vague statements, discussed the influence of $\eta$, and discussed that the comparison with the related work is fair in the new version. Also, please refer to Remarks 2-4, Table 1, and Section 5.1 in the new version for more detailed discussions. Thank you for your invaluable comments.
>
>
> **R3: Apart from the above concern, I agree with the authors that it would be interesting to see if assumptions can be relaxed. It would be helpful to have more discussions on the dependency of these assumptions.**
>
> **A**:  Generalization performance analysis of minimax problems is rarely studied. Due to the complexity of minimax problems, to establish fast $\mathcal{O}(1/n)$ rates, existing generalization bounds in the related work [13,14,15] almost all require strong assumptions. Even though [13,14,15] study the expected error, the Lipschitz continuity, smoothness, and strongly-convexity-strongly-concavity conditions are all needed in their papers (cf. Table 1 in our paper). We now discuss the direction which may relax these conditions. Recently, the Polyak-{\l}ojasiewicz (PL) condition is a popular alternative in relaxing the strong convexity assumption in (stochastic) optimization problems [16]. In the minimax optimization literature, [17] introduced the two-sided PL condition to relax the strongly-convexity-strongly-concavity condition. The smoothness condition is also relaxed to H\"older smoothness condition in [18]. We think that the conditions used in this paper may be relaxed in this direction.
>
>
> [13] Farzan Farnia and Asuman Ozdaglar. Train simultaneously, generalize better: Stability of gradient based minimax learners. ICML, 2021.
>
> [14] Yunwen Lei, Zhenhuan Yang, Tianbao Yang, and Yiming Ying. Stability and generalization of
> stochastic gradient methods for minimax problems. ICML, 2021.
>
> [15] Junyu Zhang, Mingyi Hong, Mengdi Wang, and Shuzhong Zhang. Generalization bounds for
> stochastic saddle point problems. AISTAS, 2021.
>
> [16] Hamed Karimi, Julie Nutini, and Mark Schmidt.
> Linear convergence of gradient and proximal-gradient methods under the polyak-{\l}ojasiewicz condition. Joint European Conference on Machine Learning and Knowledge Discovery in Databases, 2016.
>
> [17] Junchi Yang, Negar Kiyavash, and Niao He. Global Convergence and Variance Reduction for a Class of Nonconvex-Nonconcave Minimax Problems. NeurIPS, 2020.
>
> [18] Yunwen Lei, and Yiming Ying. Fine-grained analysis of stability and generalization for stochastic gradient descent. ICML, 2020.
>
>
> **R4:  The paper is nicely written. There is a concern regarding the main theorem's dependency on the (1+eta) factor for the empirical measures. This also leads to a concern about the fairness of comparisons with related work.**
>
> **A**: We have carefully revised the statements of this paper. We also have explained the impact of the $(1+\eta)$ factor in detail in the new version and made a fair comparison with the related work. Please refer to Remarks 2-4, Table 1, and Section 5.1 in the new version. We sincerely hope our responses will address your concerns well.

---

> > ### Comment · Reviewer_Wpqu · 2021-11-29
> > **Update after author response**
> >
> > Thank you for responding to my concerns. I have increased my score from 5 to 6. I agree with other reviewers that the descriptions of the effect of eta, comparisons against other work, and the claims of the paper deserves careful revision in future versions.

---

> > > ### Author Response · Authors · 2021-11-30
> > > **Thanks for the reply**
> > >
> > > Dear Reviewer Wpqu, we sincerely thank you for your reply and the recognition of this paper.
> > >
> > > In the final version, we will further carefully revise this paper and add more discussions.
> > >
> > >
> > > Thank you for raising your score. We feel very encouraged.

---

### Official Review · Reviewer_jPJ9 · 2021-10-31

**Correctness:** 3
**Technical Novelty And Significance:** 3
**Empirical Novelty And Significance:** Not applicable
**Recommendation:** 8
**Confidence:** 5

**Main Review:**

The results obtained in the paper are new and are very nice extension of the existing work on minimax optimization algorithms.  The related work is well cited and the paper is organized well.  I have the following comments and suggestions:

1.  it seems that the paper overly claimed the improvement over the existing work. In the abstract and introduction, the authors claim that " improved generalization analyses for almost all existing generalization measures of minimax problems which enables the minimax
problems to establish sharper bounds of order $O(1/n)$, significantly, with high probability".  This statement appeared several times in the paper.  From my perspective, it is not correct and misleading since the excess bounds are precisely given in the form of  $O(1/n +  \epsilon_{stab})$ plus extra terms (e.g.   extra $\eta F_S(A(S))$,  $\eta \bigtriangleup^s_S(A(s))$.  Indeed, the excess strong PD generalization bound is given by $O({1 \over \eta n} +  \epsilon_{stab}  +   \eta \bigtriangleup^s_S(A(s)) ).$   For the reason above, the comparison results in Table 1 in page 8 is extremely vague and not precise.  The bounds stated in Theorem 2 and 3 are also incorrect because the extra term (e.g. $\eta F_S(A(S))$ or $\\eta \bigtriangleup^s_S(A(s))$ or $\eta\inf_w R(w)$) is ignored there.

2. I never saw uniform stability for SGDA before:   uniform stability only holds true after taking expectation with respect to the internal randomness of the algorithm.   Indeed, the proof for Theorem 7 is not correct.  In this case, the uniform stability, i.e. inequality of (39) for SGDA holds with high probability.   But the whole of this paper is under the uniform stability assumption (NOT high probability form). Hence, you can not apply Theorem 1 to directly get the results in Theorem 7.

2.  While the results are new and interesting, the proof techniques are directly modified from Klochkov and Zhivotovskiy (2021).

3. Part (c) and Part (d) are duplicate as they are essentially the same results.




**Summary Of The Paper:**

This paper conducted a comprehensive study of high probability generalization bounds for minimax problems in terms of various forms of measuring generalization such as strong/weak PD generalization and primal generalization error.  Given uniform stability $\epsilon_{stab}$,  it proved sharper bounds of order $O(1/n + \epsilon_{stab})$  while sacrificing the empirical error (e.g.   extra $\eta F_S(A(S))$,  $\eta \bigtriangleup^s_S(A(s))$ on the bound). The results improves the previous excess generalization bound $O(1/\sqrt{n}+ \epsilon_{stab}))$ when  $F_S(A(S))$ or $\bigtriangleup^s_S(A(s))$ is very small.  The results extend the existing work by Lei et al. (2021) where most of the bounds were derived in expectation and high probability ones are given by f order $O(1/\sqrt{n} + \epsilon_{stab})$.  Then, specific bounds were given for GDA, PPM and SGDA for the strongly convex and strongly concave case as they are uniform stable.

**Summary Of The Review:**

Overall, the paper presents very nice high-probabilistic generalization bounds for minimax problems and improves the existing ones which are mostly given in expectation.  The main proof techniques are adapted from Klochkov and Zhivotovskiy (2021).  The paper is also well organized and related work is sufficient.  However, the claims about significant improvement over existing work and providing sharper bound $O(1/n)$ are not precise and need to be corrected.  The proof for Theorem 7 is problematic since SGDA is not uniform stability as far as I know:  it is uniform stable only after taking expectation w.r.t. the internal randomness of the algorithm.


**The authors have addressed main concerns on uniform stability of SGDA. It is a nice extension of the existing work. I only have minor suggestion on the final version of the paper about modifying the abstract in a modest tone and the caption to Table 1.   I will increase my score.**

---

> ### Author Response · Authors · 2021-11-18
> **Responses to Reviewer jPJ9 (Part 3/3)**
>
>
> The reason is that the supremum operator in $\inf_{\mathbf{x} \in \mathcal{X}} R(x)$ makes the Bernstein condition used in (Klochkov and Zhivotovskiy, 2021) not applicable for excess primal population risk. Additionally, (Klochkov and Zhivotovskiy, 2021) only study ERM and GD, while we study more optimization algorithms: SGDA, PPM, EG, and OGDA. In summary, we believe the techniques we develop to adapt to the minimax problems are novel and are not directly extended from Klochkov and Zhivotovskiy, (2021).
>
> **R4: Part (c) and Part (d) are duplicate as they are essentially the same results.**
>
> **A:** Part (c) and Part (d) have a similar upper bound. However, (c) measures the strong PD population risk (i.e., $\sup_{\mathbf{y}' \in \mathcal{Y}}F(\mathbf{x},\mathbf{y}') - \inf_{\mathbf{x}' \in \mathcal{X}} F(\mathbf{x}',\mathbf{y})$), while (d) measures the strong PD generalization error (i.e., $(\sup_{\mathbf{y}' \in \mathcal{Y}}F( \mathbf{x},\mathbf{y}' ) - \sup_{\mathbf{y}' \in \mathcal{Y}}F_S( \mathbf{x},\mathbf{y}' ) ) + ( \inf_{\mathbf{x}' \in \mathcal{X}}F_S (\mathbf{x}',\mathbf{y}) - \inf_{\mathbf{x}' \in \mathcal{X}}F (\mathbf{x}',\mathbf{y}) )$). Considering they are different generalization measures and our goal is to provide high probability bounds for almost all existing generalization measures of minimax problems, we thus put them all in Theorem 1. Indeed, the weak PD population risk (i.e., $\sup_{\mathbf{y}' \in \mathcal{Y}} \mathbb{E}[F(\mathbf{x},\mathbf{y}')] - \inf_{\mathbf{x}' \in \mathcal{X}} \mathbb{E}[F(\mathbf{x}',\mathbf{y})]$) and the weak PD generalization error (i.e., $(\sup_{\mathbf{y}' \in \mathcal{Y}}\mathbb{E}[F( \mathbf{x},\mathbf{y}' )] - \sup_{\mathbf{y}' \in \mathcal{Y}}\mathbb{E}[F_S( \mathbf{x},\mathbf{y}' )] ) + ( \inf_{\mathbf{x}' \in \mathcal{X}}\mathbb{E}[F_S (\mathbf{x}',\mathbf{y})] - \inf_{\mathbf{x}' \in \mathcal{X}} \mathbb{E}[F (\mathbf{x}',\mathbf{y})] )$), the weak versions of the strong PD population risk and strong PD generalization error, are also simultaneously studied for SGDA in [4]. We can remove one of (c) or (d) if it's necessary.
>
> [4] Yunwen Lei, Zhenhuan Yang, Tianbao Yang, and Yiming Ying. Stability and generalization of
> stochastic gradient methods for minimax problems. ICML, 2021.
>
> **R5: Overall, the paper presents very nice high-probabilistic generalization bounds......taking expectation w.r.t. the internal randomness of the algorithm.**
>
> **A:** We have carefully revised the statements of this paper. We sincerely hope our responses will address your concerns well. Thank you for your invaluable comments.

---

> ### Author Response · Authors · 2021-11-18
> **Responses to Reviewer jPJ9  (Part 2/3)**
>
>
> **R3: While the results are new and interesting, the proof techniques are directly modified from Klochkov and Zhivotovskiy (2021).**
>
> **A:** We discussed in Remark 4 (in our old version) that this work is inspired by the recent breakthrough work of Klochkov and Zhivotovskiy (2021). However, we would like to claim that the proof techniques of Klochkov and Zhivotovskiy (2021) can not be directly extended to minimax problems. The coupling construction between the minimization variables and the maximization variables makes the proofs of minimax problems more difficult than the minimization problem studied by Klochkov and Zhivotovskiy (2021). We must proceed with novel decompositions for the generalization measures. Note
> that different decompositions are required for different generalization measures. A pretty technical decomposition is exploited in the proof of Part (c). Specifically, to prove Part (c), we need the following decomposition
> $$
> \begin{align*}
> &\bigtriangleup^s(A\_{\mathbf{x}}(S),A\_{\mathbf{y}}(S))
> =F( A\_{\mathbf{x}}(S), \mathbf{y}^{\ast}\_S ) - \inf\_{\mathbf{x}' \in \mathcal{X}}F (\mathbf{x}',A\_{\mathbf{y}}(S)) \\\\
> = &F( A\_{\mathbf{x}}(S),\mathbf{y}^{\ast}\_S ) -  F\_S (A\_{\mathbf{x}}(S),\hat{\mathbf{y}}^{\ast}\_S) + \mathbb{E}\_{S'} F\_S ( A\_{\mathbf{x}}(S'),\mathbf{y}^{\ast}\_{S'}) - \mathbb{E}\_S F( A\_{\mathbf{x}}(S),\mathbf{y}^{\ast}\_{S}) \\\\
> &+\mathbb{E}\_S F(\mathbf{x}^{\ast}\_{S}, A\_{\mathbf{y}}(S)) - \mathbb{E}\_{S'} F\_S ( \mathbf{x}^{\ast}\_{S'},A\_{\mathbf{y}}(S')) + F\_S (\hat{\mathbf{x}}^{\ast}\_S,A\_{\mathbf{y}}(S)) - \inf\_{\mathbf{x}' \in \mathcal{X}}F (\mathbf{x}',A\_{\mathbf{y}}(S))  \\\\
> &-\mathbb{E}\_{S'} F\_S ( A\_{\mathbf{x}}(S'),\mathbf{y}^{\ast}\_{S'}) +\mathbb{E}\_S F( A\_{\mathbf{x}}(S),\mathbf{y}^{\ast}\_{S}) +  \mathbb{E}\_{S'} F\_S ( \mathbf{x}^{\ast}\_{S'},A\_{\mathbf{y}}(S')) -\mathbb{E}\_S F(\mathbf{x}^{\ast}\_{S}, A\_{\mathbf{y}}(S)) \\\\
>  &+F\_S (A\_{\mathbf{x}}(S),\hat{\mathbf{y}}^{\ast}\_S) - F\_S (\hat{\mathbf{x}}^{\ast}\_S,A\_{\mathbf{y}}(S)).
> \end{align*}
> $$
> The terms in the second line require the following decomposition
> $$
> \begin{align*}
> &F( A\_{\mathbf{x}}(S),\mathbf{y}^{\ast}\_S ) -  F\_S (A\_{\mathbf{x}}(S),\hat{\mathbf{y}}^{\ast}\_S) + \frac{1}{n}\sum\_{i=1}^n \mathbb{E}\_{z'\_i} \Big[ \mathbb{E}\_Z [f(A\_{\mathbf{x}}(S^{(i)}),\mathbf{y}^{\ast}\_{S^{(i)}}; Z)] - f(A\_{\mathbf{x}}(S^{(i)}),\mathbf{y}^{\ast}\_{S^{(i)}}; z\_i) \Big] \\\\
> &-\frac{1}{n}\sum\_{i=1}^n\mathbb{E}\_{z'\_i} \Big[ \mathbb{E}\_Z [f(A\_{\mathbf{x}}(S^{(i)}),\mathbf{y}^{\ast}\_{S^{(i)}}; Z)] - f(A\_{\mathbf{x}}(S^{(i)}),\mathbf{y}^{\ast}\_{S^{(i)}}; z\_i) \Big]  +\mathbb{E}\_{S'} F\_S ( A\_{\mathbf{x}}(S'),\mathbf{y}^{\ast}\_{S'}) -\mathbb{E}\_S F( A\_{\mathbf{x}}(S),\mathbf{y}^{\ast}\_{S}).
> \end{align*}
> $$
> The terms in the third line then require another different decomposition
> $$
> \begin{align*}
> & F\_S (\hat{\mathbf{x}}^{\ast}\_S,A\_{\mathbf{y}}(S))- \frac{1}{n}\sum\_{i=1}^n \mathbb{E}\_Z\left[  f(\mathbf{x}^{\ast}\_{S}, A\_{\mathbf{y}}(S),Z)  - \mathbb{E}\_{z'\_i} [ f(\mathbf{x}\_{S^{(i)}}^{\ast},A\_{\mathbf{y}}(S^{(i)});Z) ] \right]\\\\
> &+\frac{1}{n}\sum\_{i=1}^n \mathbb{E}\_{z'\_{i}} \left[ f(\mathbf{x}\_{S^{(i)}}^{\ast},A\_{\mathbf{y}}(S^{(i)});z\_i) -  \mathbb{E}\_Z[ f(\mathbf{x}\_{S^{(i)}}^{\ast},A\_{\mathbf{y}}(S^{(i)});Z) ] \right]\\\\
> &+\mathbb{E}\_S F(\mathbf{x}^{\ast}\_{S}, A\_{\mathbf{y}}(S)) - \mathbb{E}\_{S'} F\_S ( \mathbf{x}^{\ast}\_{S'},A\_{\mathbf{y}}(S')) - \frac{1}{n}\sum\_{i=1}^n \mathbb{E}\_{z'\_i}\left[ f(\mathbf{x}\_{S^{(i)}}^{\ast},A\_{\mathbf{y}}(S^{(i)});z\_i) \right].
> \end{align*}
> $$
> In contrast, in the minimization problem studied by Klochkov and Zhivotovskiy (2021), the decomposition is relatively simple:
> \begin{align*}
> F( A_{\mathbf{x}}(S)) - F ( A_{\mathbf{x}}^{\ast})
> = F( A_{\mathbf{x}}(S)) -F_S( A_{\mathbf{x}}(S))+F_S( A_{\mathbf{x}}(S))- F_S( A_{\mathbf{x}}^{\ast})+ F_S( A_{\mathbf{x}}^{\ast})-F ( A_{\mathbf{x}}^{\ast}).
> \end{align*}
> $F( A_{\mathbf{x}}(S)) -F_S( A_{\mathbf{x}}(S))$ is then needed to be decomposed as
> \begin{align*}
> F( A\_{\mathbf{x}}(S)) -F\_S( A\_{\mathbf{x}}(S))+\mathbb{E}\_{S'} F\_S ( A\_{\mathbf{x}}(S')) - \mathbb{E}\_S F( A\_{\mathbf{x}}(S))-\mathbb{E}\_{S'} F\_S ( A\_{\mathbf{x}}(S'))+ \mathbb{E}\_S F( A\_{\mathbf{x}}(S).
> \end{align*}
> By comparison, one can see that due to the minimax structure, the proof of minimax problems needs more refined analysis, where this requirement is scattered in many places. For instance, in proving the primal generalization error of Part (b), we need to quantify the fact that for different $A_{\mathbf{x}}(S)$, the optimal $\mathbf{y}$ is different in $R(A_{\mathbf{x}}(S))$. And the analysis of excess primal population risk in Part (e) is different from the excess risk analysis in (Klochkov and Zhivotovskiy, 2021).

---

> > ### Comment · Reviewer_jPJ9 · 2021-11-29
> > **thanks and further comments**
> >
> > Thank you.   My main concern on uniform stability of SGDA is well addressed.  I will support the acceptance of the paper as it is a nice extension and improvement of the existing results.   Two more concerns needs to be addressed in your final version:
> >
> > 1. In the abstract:  the statement"obtain sharper high probability generalization bounds for $\textbf{almost all existing generalization measures}$ of minimax problems" is misleading.  I strongly suggest you to modify this claim as, if t extra $\eta$ terms is large, then your bound doe not improve the existing ones.  It is not bad thing to use modest phrases.
> >
> > 2. Table 1 is misleading here:  your excess generalization bound does not lead to 1/n as there is extra $\eta$ terms. ( e.g., $\eta F_S$).   In the caption for Table 1, you may consider to add your assumptions.

---

> > > ### Author Response · Authors · 2021-11-29
> > > **Thanks for the reply**
> > >
> > > Dear Reviewer jPJ9, we sincerely thank you for your reply and the recognition of this paper.
> > >
> > > Regarding your new concerns:
> > >
> > > 1: we are deeply impressed by your comment "It is not bad thing to use modest phrases". We will definitely revise the statement you pointed out and accurately express the impact of $\eta$ in the abstract.
> > >
> > > 2: we added the "LN" condition (i.e., the empirical risk is small) in Table 1 of the previous version,  showing that our bounds can reach a fast rate under this condition. According to your comment "In the caption for Table 1, you may consider to add your assumptions", we will definitely add the assumptions to the caption of Table 1.
> > >
> > > Thanks again for your recognition of this paper. We feel very encouraged.

---

> ### Author Response · Authors · 2021-11-18
> **Responses to Reviewer jPJ9 (Part 1/3)**
>
> **Dear Reviewer jPJ9**, we sincerely appreciate your invaluable and constructive comments. We respond to your concerns one by one.
>
> **R1: it seems that the paper overly claimed the improvement over the existing work......because the extra term (e.g. $\eta F_S(A(S))$ or $\eta \bigtriangleup^s_S(A(S))$ or $\eta \inf_{\mathbf{x} \in \mathcal{X}} R(\mathbf{x})$ ) is ignored there.**
>
> **A:** We have carefully revised the statements of this paper. We have made a detailed discussion on the "extra terms", refer to Remarks 2-4 (new version). In Table 1, we have added the assumptions that $F_S (A_{\mathbf{x}}(S),A_{\mathbf{y}}(S))$, $R_S( A_{\mathbf{x}}(S))$ or $\inf_{\mathbf{x} \in \mathcal{X}} R(\mathbf{x})$ should be small. Due to the length limit, we put Theorem 2 and 3 to Remark 8 and Remark 12, where we also have added the above assumptions. And we would like to make more clarification of this paper, **please refer to the public response to all Reviewers**.
>
> **R2: I never saw uniform stability for SGDA before......Hence, you can not apply Theorem 1 to directly get the results in Theorem 7.**
>
> **A2:** High-probability upper bounds on uniform stability lead to high-probability upper bounds on generalization error [2]. Let's take the plain generalization error of Part (a) as an example. If $A$ has $\epsilon$-uniform stability with probability at least $1 - \delta'$ for some $\delta' \in (0,1)$ over the randomness of $A$, i.e.,
> \begin{align*}
> Pr_A \Big(\sup_{z} \left[f(A_{\mathbf{x}}(S),A_{\mathbf{y}}(S); z) - f(A_{\mathbf{x}}(S'),A_{\mathbf{y}}(S'); z) \right] \Big) \leq \epsilon.
> \end{align*}
> And if the randomness of $A$ is independent of the training set $S$. Then with probability at
> least **$1 - \delta - \delta'$** over $S$ and $A$,
> \begin{multline*}
> F( A_{\mathbf{x}}(S),A_{\mathbf{y}}(S) ) \leq (1+\eta)F_S (A_{\mathbf{x}}(S),A_{\mathbf{y}}(S))  +  C\frac{1+\eta}{\eta}\Big(\frac{M}{n}\log (1/\delta)+\epsilon  \log_2 n  \log \frac{1}{\delta}\Big).
> \end{multline*}
> The above inequalities mean that if the uniform stability holds with high probability, then the upper bound of Part (a) still holds, but with lower confidence **$1 - \delta - \delta'$**. Therefore, when we have established the high probability stability bounds of SGDA, we can provide high probability generalization bounds for SGDA by the above similar variant of Theorem 1. Indeed, as shown in Theorem 7 (in our old version), the generalization bounds of SGDA are provided with probability at least $1-2\delta$, while other deterministic algorithms are presented with probability at least $1-\delta$. To our best knowledge, existing high probability generalization bounds of SGD are almost proved by this method. Specifically, please refer to Theorem 4.5 in [1], Corollary A.5 in [2], Theorem 2.2 in [3], and the high probability generalization bounds of SGDA in [4] (cf. Theorem 4), etc. Moreover, we would like to emphasize that our generalization bounds' dependence on $\delta$ is logarithmic. Therefore, by replacing $\delta$ to $\delta'/2$, our generalization bounds for SGDA still have a high confidence $1-\delta'$. And note that there is little difference between $\log \frac{2}{\delta'}$ and $\log \frac{1}{\delta'}$ in the upper bound. The proof of SGDA maybe a little jumping. We have made a clarification in Theorem 5 (new version).
>
> [1] Vitaly Feldman and Jan Vondrak. High probability generalization bounds for uniformly stable algorithms with nearly optimal rate. ArXiv Preprint ArXiv:1902.10710, 2019. (refer to the Arxiv version)
>
> [2] Raef Bassily, Vitaly Feldman, Cristobal Guzman, and Kunal Talwar. Stability of stochastic gradient descent on nonsmooth convex losses. NeurIPS, 2020.
>
> [3] Zhenhuan Yang, Yunwen Lei, Siwei Lyu, Yiming Ying. Stability and Differential Privacy of Stochastic Gradient Descent for Pairwise Learning with Non-Smooth Loss. AISTATS, 2021.
>
> [4] Yunwen Lei, Zhenhuan Yang, Tianbao Yang, and Yiming Ying. Stability and generalization of
> stochastic gradient methods for minimax problems. ICML, 2021.

---

### Official Review · Reviewer_RFir · 2021-11-02

**Correctness:** 4
**Technical Novelty And Significance:** 4
**Empirical Novelty And Significance:** 4
**Recommendation:** 6
**Confidence:** 4

**Main Review:**

This paper generalizes the algorithmic stability notions to the minimax problems and provides high probability bounds for some specific instances of problem/algorithm pairings when the problem has bounded values. I think this is a good contribution.

There are some issues with presentation and I think some of the statements are a little vaguer than they need to be. Instead of saying "the plain generalization error is $O(1/n)$" perhaps one could provide a precise statement about what exactly is $O(1/n)$ as otherwise readers have to go a back and interpret the results. In particular, from what I can see in part (a) of the main theorem, it was shown:
$$F(x,y) \le (1+\eta)F_S(x,y) + \tilde O\left(\frac{M+\epsilon}{\eta n}\right)$$
Now, how is one to read an $O(1/n)$ rate out of this? The $\tilde O(M/\eta n)$ part suggests $\eta=\Theta(1)$, but then I have a $2F_S(x,y)$. Is it the case that $F_S(x,y)=0$? This seems unlikely to me, but perhaps I missed something.

In fact, ALL of the statements made about generalization error rates suffer from this issue: I intuitively expect the thing that we are measuring the "rate" of to be True Value - Empirical Value, or maybe "True value at point produced by algorithm - Optimal True value", but it appears in Theorem 1 that we are actually looking at True Value - $(1+\eta)\times$ Empirical value, which seems a bit weird. Now, I believe that in many of the cases for specific algorithms (e.g. using the empirical minimizer), it might be that the empirical value is $0$ or $O(1/n)$ and so we have do have a fast rate as realized in the later theorems, but I think this is asking for a bit too much jumping ahead by the reader. Instead perhaps it is just the case that Theorem 1 provides a "self-bounding'' style result that in concert with specific algorithms can be used to obtain a $O(1/n)$ rate?

These things would be clarified if the statements about generalization error were quantified correctly in equations rather than prose.

However, while I am rather dissatisfied with these aspects of the presentation, I believe they can be rectified and if so the contributions are enough to merit acceptance.


**Summary Of The Paper:**

This paper consider a stochastic minimax problem of the form $\min_x \max_y F(x,y)$ where $F(x,y) = \mathbb{E}[f(x,y,z)]$. The goal is to obtain ``fast'' rates of $O(1/n)$ with a training set of $n$ i.i.d. $z_1,\dots,z_n$ in high probability.
In this setting there are a variety of convergence measures one might consider, most notably perhaps the gap $\Delta(x,y) = \max_{\hat y} F(x,\hat y) - \min_{\hat x} F(\hat x, y)$. For these measures the fast rates are achieved for various specific problems such as strongly-convex-concave saddle-point detection. The main technique is to generalize classical results on algorithmic stability developed for stochastic minimization towards the stochastic minimax problem, and then to demonstrate that a variety of algorithms (mostly variants on gradient descent) are stable in particular settings.


**Summary Of The Review:**

The paper provides new results in high probability that are valuable confirmation that standard algorithms will work well. The presentation leaves something to be desired, but is likely fixable in revision.

---

> ### Author Response · Authors · 2021-11-18
> **Responses to Reviewer RFir**
>
> **Dear Reviewer RFir**, we sincerely appreciate your invaluable and constructive comments. We respond to your concerns one by one.
>
> **R1: There are some issues with presentation and I think some of the statements are a little vaguer than they need to be......Instead perhaps it is just the case that Theorem 1 provides a "self-bounding" style result that in concert with specific algorithms can be used to obtain a $\mathcal{O}(1/n)$ rate?**
>
> **A:** **Please refer to the public response to all Reviewers.** In this response, we have clarified the vague statements and discussed how our generalization bounds imply fast rates. We also have discussed the influence of $\eta$, why we provide the bounds of the form True Value $\leq (1+\eta) \times$ Empirical value $+ \mathcal{O}(\cdots) $, and how Part (b) and Part (c) provide precise $\mathcal{O}(1/n)$ rate. Also, please refer to Remark 2, Remark 3, Remark 4, Table 1, and Section 5.1 in the new version, where we have made more detailed discussions on your concerns.
>
> **R2: These things would be clarified if the statements about generalization error were quantified correctly in equations rather than prose.**
>
> **A**: We have carefully revised the statements about generalization errors. Please refer to Remarks 2-4, Table 1, and Section 5.1 in the new version for more details.
>
> **R3: However, while I am rather dissatisfied with these aspects of the presentation, I believe they can be rectified and if so the contributions are enough to merit acceptance.**
>
> **A**: We sincerely hope our responses will address your concerns well. Thank you for your invaluable comments.

---

### Official Review · Reviewer_LsE5 · 2021-11-04

**Correctness:** 3
**Technical Novelty And Significance:** 4
**Empirical Novelty And Significance:** Not applicable
**Recommendation:** 8
**Confidence:** 3

**Main Review:**

Overall, I think this is a timely solid work for the minimax learning problems which covers many interesting topics.

-------------------- pros

1. This paper is very well-written.
2. It is valuable to move expectation bound to high probability bound or improve high probability bound from $O(1/\sqrt{n})$ to $O(1/n)$.
3. It clearly discusses its relationships and differences with previous work.
4. The proofs are overall right although I cannot check the proofs line-by-line.

-------------------- cons

1. I have one concern is that how is the tightness of the achieved generalization upper bound,  and thus the lower bound might be discussed to illustrate it.
2. It might be better to add some synthetic empirical experiments to illustrate the tightness and effectiveness of these generalization upper bounds w.r.t. the true population risk.

Besides, I have some questions as follows.

1. Why to define the strong or weak measures (Def. 1) like these? Please give more intuitive explanation.
2. For Theorem 1, what is the differences between (c.) and (d.)? I think they are the same, do I miss something?
3. For Theorem 1, what is the effect of variable $\eta$? Please give more discussions.
4. In Remark 1, authors repeatedly claims that what if argument stability and the strong PD empirical risk are of the fast order $O(1/n)$. While the argument stability with $O(1/n)$ can be understood, how to understand the strong PD empirical risk has dependence on the sample size $n$?

--------------------- Minor comments

Typos:

1. Page 4. head: $|| z ||_p = (\mathbb{E} \| z \|^p)^{1/p}$.



**Summary Of The Paper:**

As a theory work, this paper considers the generalization of the minimax learning problems which has a wide applications in machine learning. In comparison with the results of the previous work, which either provide expectation bounds or high probability generalization bounds of $O(1/\sqrt{n})$, this paper gives improved high generalization bounds of $O(1/n)$. Besides, these bounds are applied in many popular optimization algorithms, including ESP, GDA, SGDA and so on.

**Summary Of The Review:**

While I think this is a solid theory work, it is better to add some discussions for the tightness of these generalization upper bounds, and some empirical validation experiments.

---

> ### Author Response · Authors · 2021-11-18
> **Responses to Reviewer LsE5 (Part 2/2)**
>
>
> Considering they are different generalization measures and our goal is to provide high probability bounds for almost all existing generalization measures of minimax problems, we thus put them all in Theorem 1. Indeed, the weak PD population risk (i.e., $\sup_{\mathbf{y}' \in \mathcal{Y}} \mathbb{E}[F(\mathbf{x},\mathbf{y}')] - \inf_{\mathbf{x}' \in \mathcal{X}} \mathbb{E}[F(\mathbf{x}',\mathbf{y})]$) and the weak PD generalization error (i.e., $(\sup_{\mathbf{y}' \in \mathcal{Y}}\mathbb{E}[F( \mathbf{x},\mathbf{y}' )] - \sup_{\mathbf{y}' \in \mathcal{Y}}\mathbb{E}[F_S( \mathbf{x},\mathbf{y}' )] ) + ( \inf_{\mathbf{x}' \in \mathcal{X}}\mathbb{E}[F_S (\mathbf{x}',\mathbf{y})] - \inf_{\mathbf{x}' \in \mathcal{X}} \mathbb{E}[F (\mathbf{x}',\mathbf{y})] )$), the weak versions of the strong PD population risk and strong PD generalization error, are also simultaneously studied for SGDA in [3]. We can remove one of (c) or (d) if it's necessary.
>
> [3] Yunwen Lei, Zhenhuan Yang, Tianbao Yang, and Yiming Ying. Stability and generalization of stochastic gradient methods for minimax problems. ICML, 2021.
>
> **R5: For Theorem 1, what is the effect of variable $\eta$?Please give more discussions.**
>
> **A**: Please refer to the public response to all Reviewers.
>
> **R6: In Remark 1, authors repeatedly claims that what if argument stability and the strong PD empirical risk are of the fast order $\mathcal{O}(1/n)$. While the argument stability with $\mathcal{O}(1/n)$ can be understood, how to understand the strong PD empirical risk has dependence on the sample size $n$?**
>
> **A**: We aim to express that if the strong PD empirical risk is small, the convergence rate of the generalization bounds will be of faster order. In our proof for the gradient-based optimization algorithms, the strong PD empirical risk mainly has a dependence on the iterative number $T$ (cf. Lemma 8 of GDA and Lemma 11 of SGDA in the new version, etc.). To obtain sharper generalization bounds, we require $T$ to be associated with $n$, such as $T= \mathcal{O}(n^2)$ for GDA, the strong PD empirical risk finally has a dependence on the sample size $n$. It may be a little jumping here. We have made a clarification in Remark 2 (new version).
>
> **R7: While I think this is a solid theory work, it is better to add some discussions for the tightness of these generalization upper bounds, and some empirical validation experiments.**
>
> **A**: We give an intuitive explanation in **R1** of the tightness. We add numerical experiments in **R2** to illustrate the tightness and effectiveness of the derived generalization bounds. We sincerely hope our responses will address your concerns well.

---

> ### Author Response · Authors · 2021-11-18
> **Responses to Reviewer LsE5 (Part 1/2)**
>
> **Dear Reviewer LsE5**, we sincerely appreciate your invaluable and constructive comments. We respond to your concerns one by one.
>
> **R1: I have one concern is that how is the tightness of the achieved generalization upper bound, and thus the lower bound might be discussed to illustrate it.**
>
> **A**: Systematically proving the lower bound of Theorem 1  is very important and interesting. We've been thinking about it, but we have not yet thought of a strict proof scheme. We provide an intuitive explanation for the tightness. When we consider the case that the maximization variable is a constant, then the minimax problem degrades to the minimization problem. The empirical saddle point (ESP) problem becomes the classical empirical risk minimizer (ERM) problem, and the strong PD population risk corresponds to the excess risk of the standard statistical learning theory setting. In this case, the strong PD population risk bound of Theorem 3 (in our new version) reveals that the excess risk bound of ERM is of order $\mathcal{O}\Big( \frac{\log_2 n}{n} \log(1/\delta) \Big)$. This upper bound is almost optimal (up to a $\log_2 n$ term) for ERM with strongly convex and Lipschitz losses (cf. Theorem 1.1 and Proposition 2.1 in [1]). From this perspective, the strong PD population risk upper bound may be almost optimal for the ESP problem. Thank you for your comments. We will continue to think about this problem and try to give strict proof of the lower bound.
>
> [1] Yegor Klochkov and Nikita Zhivotovskiy. Stability and deviation optimal risk bounds with convergence rate $O(1/n)$. NeurIPS, 2021.
>
> **R2: It might be better to add some synthetic empirical experiments to illustrate the tightness and effectiveness of these generalization upper bounds w.r.t. the true population risk.**
>
> **A**: We have added numerical experiments to support our theory. We report the convergence rates of the generalization error versus the number of samples on GDA, SGDA, EG, and OGDA. The experimental results match the predictive rates in Table 1, which verifies our theoretical findings. Please refer to Appendix I of the new version for details.
>
> **R3: Why to define the strong or weak measures (Def. 1) like these? Please give more intuitive explanation.**
>
> **A**: These generalization measures exist in the previous generalization studies of minimax problems. We now provide explanations for the four groups of measures in Definition 1. (1. Primal Measures:) In the context of GANs, the primal population risk $R(\mathbf{x})$ represents a divergence measure between the learned and true distributions, and in the context of adversarial training it represents the learner’s risk under adversarial perturbations [2]. One would be interested in the relationship between $R(\mathbf{x})$ and its corresponding empirical risk $R_S(\mathbf{x})$, and the relationship between $R(\mathbf{x})$ and its infimum $\inf_{\mathbf{x}' \in \mathcal{X}} R(\mathbf{x}')$. (2. Plain Measure:) This generalization measure is a direct extension of the standard generalization error in the minimization optimization. (3. Strong Measures:) $\bigtriangleup^s_S(\mathbf{x},\mathbf{y})$ is referred to as the primal-dual gap in the optimization literature. $\bigtriangleup^s(\mathbf{x},\mathbf{y})$ is the primal-dual gap of the population risk. $\bigtriangleup^s(\mathbf{x},\mathbf{y})  - \bigtriangleup^s_S(\mathbf{x},\mathbf{y})$ studies the difference between the population primal-dual gap and its empirical counterpart. (4. Weak Measures:) The difference between the strong and weak measures is that weak measures take the expectation over the randomness of the dataset and the algorithm, for instance, $\sup_{\mathbf{y}' \in \mathcal{Y}}F(\mathbf{x},\mathbf{y}') - \inf_{\mathbf{x}' \in \mathcal{X}} F(\mathbf{x}',\mathbf{y})$ in the strong measures and $\sup_{\mathbf{y}' \in \mathcal{Y}} \mathbb{E}[F(\mathbf{x},\mathbf{y}')] - \inf_{\mathbf{x}' \in \mathcal{X}} \mathbb{E}[F(\mathbf{x}',\mathbf{y})]$ in the weak measures. Therefore, the upper bounds of weak measures hold in expectation, while the upper bounds of strong measures hold uniformly for any dataset. We have made a clarification in Remark 1.
>
> [2] Farzan Farnia and Asuman Ozdaglar. Train simultaneously, generalize better: Stability of gradient based minimax learners. ICML, 2021.
>
> **R4: For Theorem 1, what is the differences between (c.) and (d.)? I think they are the same, do I miss something?**
>
> **A**: They have a similar upper bound. However, (c) measures the strong PD population risk (i.e., $\sup_{\mathbf{y}' \in \mathcal{Y}}F(\mathbf{x},\mathbf{y}') - \inf_{\mathbf{x}' \in \mathcal{X}} F(\mathbf{x}',\mathbf{y})$), while (d) measures the strong PD generalization error ( $(\sup_{\mathbf{y}' \in \mathcal{Y}}F( \mathbf{x},\mathbf{y}' ) - \sup_{\mathbf{y}' \in \mathcal{Y}}F_S( \mathbf{x},\mathbf{y}' ) ) + ( \inf_{\mathbf{x}' \in \mathcal{X}}F_S (\mathbf{x}',\mathbf{y}) - \inf_{\mathbf{x}' \in \mathcal{X}}F (\mathbf{x}',\mathbf{y}) )$).

---

### Author Response · Authors · 2021-11-18
**The effect of variable $\eta$ (Part 3/3).**


**(5.)** This section clarifies the comparison with the related work. We have carefully revised the comparison in the new version. In Remarks 2-4 and Table 1, we have added the assumptions that $F( A_{\mathbf{x}}(S),A_{\mathbf{y}}(S) ) $, $R( A_{\mathbf{x}}(S) ) $, and $\inf_{\mathbf{x} \in \mathcal{X}} R(\mathbf{x})$ should be of order $\mathcal{O}(1/n)$. Moreover, we would like to claim that our comparison with the related work is fair in the new version. Specifically, in Part (a), Part (b), and Part (e), we study the upper bounds of $F( A_{\mathbf{x}}(S),A_{\mathbf{y}}(S) ) $, $R( A_{\mathbf{x}}(S) ) $ (w.r.t. $ R_S( A_{\mathbf{x}}(S))$), and $R( A_{\mathbf{x}}(S) ) $ (w.r.t. $\inf_{\mathbf{x} \in \mathcal{X}} R(\mathbf{x})$), respectively, while [2,8] study the upper bounds of $F( A_{\mathbf{x}}(S),A_{\mathbf{y}}(S) ) - F_S (A_{\mathbf{x}}(S),A_{\mathbf{y}}(S))$, $R( A_{\mathbf{x}}(S) ) -R_S( A_{\mathbf{x}}(S)) $, and $R( A_{\mathbf{x}}(S) ) -\inf_{\mathbf{x} \in \mathcal{X}} R(\mathbf{x})$ (or their expected form).  In the comparison with the related work, for our learning bounds and the results in [2,8], we all take the right side of the generalization bound inequalities to compare, which is fair since our bounds can be written as $F( A_{\mathbf{x}}(S),A_{\mathbf{y}}(S) )-F_S (A_{\mathbf{x}}(S),A_{\mathbf{y}}(S)) \leq \eta F_S (A_{\mathbf{x}}(S),A_{\mathbf{y}}(S))+C\frac{1+\eta}{\eta}(\frac{M}{n}\log (1/\delta)+\epsilon  \log_2 n  \log \frac{1}{\delta})$, etc. Please note that the comparison involving Part (c) and Part (d) has always been fair.


**(6.)** Please refer to Remark 2, Remark 3, Remark 4, Table 1, and Section 5.1 in the new version for more detailed discussions. Thank all the Reviewers again for your careful reading and constructive comments.

[1] Yegor Klochkov and Nikita Zhivotovskiy. Stability and deviation optimal risk bounds with convergence rate $O(1/n)$. NeurIPS, 2021.

[2] Farzan Farnia and Asuman Ozdaglar. Train simultaneously, generalize better: Stability of gradient based minimax learners. ICML, 2021.

[3] Peter L. Bartlett, Olivier Bousquet, and Shahar Mendelson. “Local Rademacher Complexities”. The Annals
of Statistics 33.4 (2005), pp. 1497–1537.

[4] Olivier Catoni. PAC-Bayesian Supervised Classification: The Thermodynamics of Statistical
Learning. Vol. 56. Lecture Notes – Monograph Series. Institute of Mathematical Statistics,
2007.

[5] Guy Lever, François Laviolette, and John Shawe-Taylor. “Tighter PAC-Bayes bounds through
distribution-dependent priors”. Theoretical Computer Science 473 (2013), pp. 4–28.

[6] Jun Yang, Shengyang Sun, and Daniel M. Roy. Fast-rate pac-bayes generalization bounds via shifted
rademacher processes. NeurIPS, 2019.

[7] Corinna Cortes, Mehryar Mohri, and Ananda Theertha Suresh. Relative Deviation Margin Bounds. ICML, 2021.

[8] Yunwen Lei, Zhenhuan Yang, Tianbao Yang, and Yiming Ying. Stability and generalization of
stochastic gradient methods for minimax problems. ICML, 2021.

[9]  Yunwen Lei and Yiming Ying. Sharper generalization bounds for learning with
gradient-dominated objective functions. ICLR, 2021.

[10] Lijun Zhang, Tianbao Yang, and Rong Jin. Empirical risk minimization for
stochastic convex optimization: $O(1/n)$- and $O(1/n^2
)$-type of risk bounds. COLT, 2017.

[11] Nathan Srebro, Karthik Sridharan, and Ambuj Tewari. Optimistic rates for
learning with a smooth loss. arXiv preprint arXiv:1009.3896, 2010.

[12] Lijun Zhang and Zhi-Hua Zhou. Stochastic approximation of smooth
and strongly convex functions: Beyond the $O(1/t)$ convergence
rate. COLT, 2019.

[13] Junyu Zhang, Mingyi Hong, Mengdi Wang, and Shuzhong Zhang. Generalization bounds for
stochastic saddle point problems. AISTAS, 2021.

---

### Author Response · Authors · 2021-11-18
**The effect of variable $\eta$ (Part 2/3).**


It has completely removed the $\mathcal{O}(1/\sqrt{n})$ term and enables $\mathcal{O}(\frac{1}{n})$ rate when the empirical risk is small. It is also discussed in [1,3,5,6,7] that this type of generalization error bound can obtain a fast rate when the empirical risk is small. For Part (b), considering that we assume the function $f$ is well-behaved, i.e., Lipschitz continuity, smoothness, and the strong-concavity of its population risk $F$, and that $R_S( A_{\mathbf{x}}(S))$ is data-dependent, thus it is reasonable to assume $R_S( A_{\mathbf{x}}(S))$ is small for a well-trained model $A_{\mathbf{x}}(S)$. Meanwhile, in the standard learning theory, assuming the optimal population risk $F^{\ast}$ is small or even zero, i.e., $F^{\ast} \leq \mathcal{O}(1/n)$, can be found in [9-12]. (Note that the optimal population risk $F^{\ast} = O(1/n)$ just to show that the improved bound can be got under low noise conditions. $F^{\ast}$ should be independent of $n$). Similar to the assumption on $F^{\ast}$ and considering that we assume the function $f$ is well-behaved, it will also be reasonable to assume $\inf_{\mathbf{x} \in \mathcal{X}} R(\mathbf{x})$ in Part (e) is small. For the above reasons, similar to the analysis of Part (a), we say that Part (b) and Part (e) also imply fast rates of convergence. In summary, the above analyses support our claim that Part (a), Part (b), and Part (e) provide sharper high probability generalization bounds.

**(3.)** This section emphasizes that Part (c) and Part (d) can provide precise $\mathcal{O}(\frac{1}{n})$ rate. For instance, Part (c) shows that $$
\begin{align*}
\bigtriangleup^s(A\_{\mathbf{x}}(S),A\_{\mathbf{y}}(S))  \leq \bigtriangleup^s\_S(A\_{\mathbf{x}}(S),A\_{\mathbf{y}}(S)) + \eta \mathbb{E}\_S \bigtriangleup^s\_S(A\_{\mathbf{x}}(S),A\_{\mathbf{y}}(S)) \\\\+ C (1+\eta) \Big( \frac{L^2 (1+\eta)}{n\mu \eta} + \frac{M}{n} + \Big( 1+ \frac{\beta}{\mu} \Big) \epsilon L  \log_2 n  \Big)\log\Big( \frac{1}{\delta}\Big).
\end{align*}
$$ When applying Theorem 1 to the later applications, we will establish $\mathcal{O}(1/n)$ order bounds for two terms: stability measures and strong PD empirical risk (i.e., $\bigtriangleup^s_S(A_{\mathbf{x}}(S),A_{\mathbf{y}}(S))$). Hence, we can choose a proper constant for $\eta$, and the strong PD population risk (Part (c) ) and the strong PD generalization error (Part (d)) will be of the fast order $\mathcal{O}(1/n)$ when applying Part (c) and Part (d) of Theorem 1 to these applications. These bounds are clearly of order $\mathcal{O}(1/n)$ and sharper than the results in [2,8,13].
Part (c) and Part (d)  provide precise $\mathcal{O}(\frac{1}{n})$ rate.

**(4.)** This section clarifies the vague statements.  For Part (a), Part (b), and Part (e), to obtain $\mathcal{O}(1/n)$ order bounds for the later applications, we need to assume the corresponding terms $F( A_{\mathbf{x}}(S),A_{\mathbf{y}}(S) ) $, $R( A_{\mathbf{x}}(S) ) $, and $\inf\_{\mathbf{x} \in \mathcal{X}} R(\mathbf{x})$ are of order $\mathcal{O}(1/n)$, respectively. For instance, for the ESP solution $(\hat{\mathbf{x}}^{\ast}\_S,\hat{\mathbf{y}}^{\ast}\_S)$, we need to assume $F(\hat{\mathbf{x}}^{\ast}\_S,\hat{\mathbf{y}}^{\ast}\_S)$, $R( \hat{\mathbf{x}}^{\ast}\_S )$, and $\inf\_{\mathbf{x} \in \mathcal{X}} R(\mathbf{x})$ are of order $\mathcal{O}(1/n)$ for Part (a), Part (b), and Part (e), respectively. The clear motivation is that in modern practice, learning algorithms achieve a small or even zero empirical risk, as discussed in **(2)**. Note that Part (c) and Part (d) don't require these assumptions. In the old version, we omitted these assumptions. We are very grateful to the Reviewers for their reminders. We have carefully revised these assumptions and the vague statements. Please refer to Remarks 2-4 and Table 1 for details in the new version.

---

### Author Response · Authors · 2021-11-18
**The effect of variable $\eta$ (Part 1/3).**

Dear Reviewers, we find that the influence of $\eta$ is a common concern. We thus respond to this concern separately here.

**(1.)** This section shows that our generalization bounds of the form "True Value $\leq (1+\eta) \times$ Empirical value" is not weird and discusses the motivation why we provide such bounds. When establishing sharper generalization error bound (i.e., $Pf - P_n f$), the existence of $\eta$ is usual in the standard statistical learning theory. Specifically, in the uniform localized convergence theory, the generalization error bound in [3] is of the form $Pf \leq \frac{\eta}{\eta-1} P_n f  + \mathcal{O}(\eta r^{\ast} + \frac{\eta \log(1/\delta)}{n})$ with $\eta > 1$ (cf. Theorem 3.3 and Theorem 4.1). In the PAC-Bayesian theory, the generalization bounds in [4] (cf. Theorem 1.2.6), [5] (cf. Theorem 6), [6] (cf. Proposition 3.1 and Theorem 4.3), etc., also have $\eta$. For instance, the Catoni's bound is of the form $PQ \leq \frac{1}{1-e^{-\eta}}(\eta P_n Q  + \mathcal{O}( \frac{KL(Q\|Prior)+ \log(1/\delta) }{n}))$ with $\eta > 0$ [4]. In the algorithmic stability theory, Theorem 1.2 in [1] is of the form  $P f \leq (1+ \eta) P_n f + \frac{1+\eta}{\eta}\mathcal{O}( (\epsilon \log n + \frac{1}{n})\log(1/\delta) )$ with $\eta > 0$. In the recent Cortes's deviation margin bounds [7], they also imply a multiplier $\eta$. The above bounds can be transformed into the form of empirical risk multiplied by $1+\eta$, similar to our results in Part (a), Part (b), and Part (e). Note that Part (e) also involves generalization error bounds due to the decomposition, see (31) in the new version. Hence, the above generalization error analysis holds for Part (e). Meanwhile, note that in practice, we are often directly interested in the population risk (i.e., $Pf$), i.e., how the learned models from the training samples behave on the testing data, such as $F( A_{\mathbf{x}}(S), A_{\mathbf{y}}(S) )$, instead of the error between the population risk and the empirical risk (i.e., $Pf - P_n f$). The above is the motivation why we directly provide bounds for $P f$.

**(2.)** This section discusses why $\eta$ appears and how Part (a), Part (b), and Part (e) imply fast rates.  We take Part (a) of Theorem 1 as an example. In (10) of Appendix A (the new version), we show that $F( A_{\mathbf{x}}(S),A_{\mathbf{y}}(S) ) - F_S (A_{\mathbf{x}}(S),A_{\mathbf{y}}(S)) \nonumber \leq \mathcal{O}\Big(\sqrt{\frac{ MF(A_{\mathbf{x}}(S),A_{\mathbf{y}}(S)) \log(1/\delta)}{n}} + \frac{\log(1/\delta)}{n} + \epsilon   \log (1/\delta) \Big)$, where $M$ is the upper bound of $f(\mathbf{x},\mathbf{y};z)$, that is $|f(\mathbf{x},\mathbf{y};z)|\leq M, \forall \mathbf{x},\mathbf{y},z$. Using the elementary inequality $\sqrt{ab} \leq \eta a + \frac{1}{\eta} b$ for any $a,b,\eta>0$ and by some rearrangements, $(1+\eta) F_S (A_{\mathbf{x}}(S),A_{\mathbf{y}}(S))$ appars. The reason for the existence of $\eta$ in Part (b) and Part (e) is similar. The following analyzes how Part (a), Part (b), and Part (e) imply fast rates. We first compare the above bound with [8], where the authors show that $F( A_{\mathbf{x}}(S),A_{\mathbf{y}}(S) ) - F_S (A_{\mathbf{x}}(S),A_{\mathbf{y}}(S)) \leq \mathcal{O}\big( \epsilon \log n \log(1/\delta) + M n^{-\frac{1}{2}} \sqrt{\log(1/\delta)}\big)$. Focusing on the dominated terms $\sqrt{\frac{ MF(A_{\mathbf{x}}(S),A_{\mathbf{y}}(S)) \log(1/\delta)}{n}}$ (ours) and $M n^{-\frac{1}{2}} \sqrt{\log(1/\delta)}$, it is clear that $F(A_{\mathbf{x}}(S),A_{\mathbf{y}}(S))\ll M$ since $F(A_{\mathbf{x}}(S),A_{\mathbf{y}}(S))$ is data-dependent, which implies that our plain generalization error bound of Part (a) is sharper. Similar analysis holds for Part (b) and Part (e). Then, we further show that our bounds enable the fast $\mathcal{O}(1/n)$ rate, while [8] only allows $n^{-\frac{1}{2}}$ rate. Part (a) of Theorem 1 is of the form $F( A_{\mathbf{x}}(S),A_{\mathbf{y}}(S) ) \leq (1+ \eta) F_S (A_{\mathbf{x}}(S),A_{\mathbf{y}}(S)) + \frac{1}{\eta}\mathcal{O}(\frac{\log (1/\delta)}{n} + \epsilon \log n \log(1/\delta))$. This inequality holds for any $\eta>0$. One can select the optimal $\eta$ to make the bound tighter. Moreover, usually for a well-trained model $(A_{\mathbf{x}}(S),A_{\mathbf{y}}(S))$ over training set, the empirical risk $F_S (A_{\mathbf{x}}(S),A_{\mathbf{y}}(S))$ is small or even zero. Suppose the empirical risk is of order $\mathcal{O}(\frac{1}{n})$, this generalization upper bound will be dominated by the later term $\mathcal{O}(\frac{\log (1/\delta)}{n} + \epsilon \log n \log(1/\delta))$. If the stability bound is also of order $\mathcal{O}(\frac{1}{n})$,  our bound has a $\mathcal{O}(\frac{1}{n})$ convergence rate. Note that the stability bounds are all established with $\mathcal{O}(\frac{1}{n})$ rates in the later applications. For this reason, we say that Part (a) implies a fast rate of convergence.

---

### Author Response · Authors · 2021-11-18
**Paper Revision**

We have carefully revised the manuscript based on the initial reviews. The following is a summary of major changes:

-  In Remark 1, we provide explanations for the generalization measures of Definition 1.

- In Remark 2, Remark 3, and Remark 4, we carefully revise the statements, including adding assumptions to generalization errors, explaining the motivation of $1+ \eta$, discussing the effect of $\eta$, discussing the fairness of comparison, and indicating that we have provided sharper generalization bounds.

- In Section 5, we revise Table 1 and put ESP and GDA in the Appendix.

- In Theorem 5, we provide a variant of Theorem 1, which is applicable for randomized algorithms.

- In Appendix I, we add numerical experiments.

We sincerely appreciate all reviewers for their invaluable and constructive comments. Hopefully, reviewers will be satisfied with these responses.

---

### Author Response · Authors · 2021-11-27
**Looking forward to further feedbacks**

Dear ACs and Reviewers,

Thank you again for the great efforts and the valuable comments. We have responded carefully and in detail to the main concerns. We hope you will be satisfied with these responses. As the discussion phase is about to close, we are very much looking forward to hearing from you about any further feedback. We will be delighted to clarify any further concerns (if any).

Thank you very much for reading this letter in such a busy schedule.

Best,

Authors

---

### Decision · Program_Chairs · 2022-01-20

**Decision:**

Accept (Poster)

**Comment:**

This paper establishes high probability generalization bounds of the order O(1/n) for a range of stochastic minimax problems. The reviewers agreed that results are of broad interest and the techniques are non-trivial. I recommend acceptance.